# Neutralizing and binding antibodies are a correlate of risk of COVID-19 in the CoVPN 3008 study in people with HIV

People with HIV (PWH) are understudied in COVID-19 vaccine trials, leaving knowledge gaps on whether the identified immune correlates of protection also hold in PWH. CoVPN 3008 (NCT05168813) enrolled predominantly PWH and reported lower COVID-19 incidence for a Hybrid vs. Vaccine Group (baseline SARS-CoV-2-positive and one mRNA-1273 dose vs. negative and two doses). Using case-cohort sampling, antibody markers at enrolment (M0) and four weeks post-final vaccination (Peak) are assessed as immune correlates of COVID-19. For the Hybrid Group [$n = 287$ (195 PWH)], all M0 markers inversely correlate with COVID-19 through 230 days post-Peak, with 50% inhibitory dilution BA.4/5 neutralizing antibody titer (nAb-ID50 BA.4/5) the strongest and only independent correlate (HR per 10-fold increase=0.46, 95% CI 0.28, 0.75; $P = 0.002$). For the Vaccine Group [$n = 115$ (86 PWH)], Peak nAb-ID50 BA.4/5 correlates with reduced COVID-19 risk (1.9%, 1.1%, and 0.3% at titers 10, 100, and 1000 AU/ml) through 92, but not 165, days post-Peak. Using multivariable Cox analysis of binding and nAb, nAb titers predict COVID-19 in PWH. Two doses of a 100-μg Ancestral strain mRNA vaccine in baseline-SARS-CoV-2-negative individuals elicit sufficient cross-reacting Omicron antibodies to reduce COVID-19 incidence for 90 days post-Peak, but viral evolution and waning antibodies abrogate this protection thereafter.

The CoVPN 3008 study investigated the effect of hybrid vs. mRNA-vaccine-alone immunity on the risk of COVID-19 and severe COVID-19 in a large population of East and Southern Africans enriched (83%) for people with HIV (PWH). Adults aged ≥18 years were enrolled in Botswana, eSwatini, Kenya, Malawi, South Africa, Uganda, and Zambia from December 2, 2021 to September 9, 2022, and assigned to groups receiving one or two doses of 100 μg of mRNA-1273 vaccine depending on testing SARS-CoV-2 anti-Spike positive or negative. The two analysis groups were defined by receiving one dose and all the diagnostic tests indicating prior infection (Hybrid Group, $N = 9613$) vs. receiving two doses and all diagnostic tests indicating naïve (Vaccine Group, $N = 2867$). Garrett et al.[1] previously reported the results of the primary safety and relative risk (Hybrid vs. Vaccine Group) objectives of CoVPN 3008, with a major finding being that the Hybrid Group had lower risk

of symptomatic COVID-19 [Centers for Disease Control and Prevention (CDC) criteria; hereafter referred to as "COVID-19"] and of severe COVID-19 between 1 day and 6 months post-enrolment compared to the Vaccine Group. Specifically, for PWH the hazard ratio (HR) for COVID-19 was 0.58 (95% CI 0.44, 0.77; $p < 0.001$) and for severe COVID-19 was 0.27 (0.07, 1.04; $p = 0.056$), based on 358 total COVID-19 endpoints (309 among PWH) and 15 total severe COVID-19 endpoints (11 among PWH)[1].

PWH are a heterogenous population in terms of level of immune suppression, systemic inflammation, and viral control[2]. Hence, immune responses in PWH to SARS-CoV-2 acquisition and to COVID-19 vaccination may vary widely, often with clear distinctions between untreated PWH vs. PWH on antiretroviral therapy[2] (such distinctions have also been shown for many non-COVID-19 vaccines[3,4]). We

✉ e-mail: gdt@duke.edu; pgilbert@fredhutch.org

previously reported trends of lower binding, ACE2 receptor blocking, and antibody-dependent cellular phagocytosis responses in unvaccinated PWH compared to unvaccinated PWoH who acquired Ancestral SARS-CoV-2[5]. Given that the study did not restrict to PWH with well-controlled HIV-1, this finding is consistent with previous conclusions that reduced immune responses to SARS-CoV-2 acquisition and to COVID-19 vaccination are seen in PWH without well-controlled HIV-1[6].

This analysis investigates antibody markers as correlates of risk (CoRs) of COVID-19, seeking insights into hybrid and vaccine immunity and whether antibody correlates previously defined in immunocompetent persons are qualitatively and quantitatively similar among PWH. A CoR refers to an immune marker that is associated with an infectious disease outcome in a cohort such as a vaccine arm or pooled set of vaccine arms. When this association is inverse, i.e., higher immune marker levels are associated with lower risk of the infectious disease outcome, the correlate is referred to as an "inverse CoR." On the other hand, an immune correlate of protection (CoP) refers to an immune marker that can be used to reliably predict a vaccine's level of protection against an infectious disease outcome[7–9]. Immune CoPs are important for understanding basic immunology, providing endpoints for immunobridging regulatory approval of new vaccine formulations or extending indications to new populations, and modeling of vaccine efficacy and effectiveness. For example, CoPs have facilitated regulatory approval of modified (e.g., variant-adapted) vaccines, as well as approval for use in a different population (e.g., pediatric) other than the one in which the original phase 3 efficacy trial(s) was conducted, for pathogens including influenza virus[10] and SARS-CoV-2[11].

A large body of work has studied immune CoRs and CoPs against COVID-19[11–13]. US Government (USG) sponsored research[14,15] in partnership with vaccine manufacturers has evaluated IgG binding antibodies (bAb) and neutralizing antibodies (nAb) as CoRs and CoPs against COVID-19 in four randomized, placebo-controlled vaccine efficacy trials[16–21], consistently showing that both markers are CoRs of COVID-19 and CoPs of vaccine protection against COVID-19[22–29], which has also been supported by other individual-level[30–32] and population-level[33–38] immune correlates analyses. While these first-wave studies evaluated antibody marker correlates prior to the emergence of the Omicron lineage in SARS-CoV-2 naïve individuals, subsequent correlates studies supported these markers as CoR and CoP for Omicron COVID-19, among both SARS-CoV-2 naïve and experienced populations[39,40]. With this study, we provide results on bAb and nAb correlates for PWH, as well as analyze the correlates in both Hybrid and Vaccine Groups and hence explore how prior infection impacts the immune correlates.

## Results

### Sets of participants included in immunogenicity and immune correlates analyses selected through a case-cohort sampling design

Following the design of previous CoVPN immune correlates studies[22–26,41], case-cohort sampling was used to determine the set of participants for measurement of immune responses at M0 (day of enrolment; before receiving any study vaccination) and Peak (one month after the last study vaccination) for inclusion in analyses (Supplementary Fig. 1). The per-protocol Serum Immunogenicity Analysis Set (Serum-IAS) was based on sera stored from 11,697 participants in the per-protocol serum correlates cohort. According to the case-cohort sampling design, M0 and Peak marker levels were measured in all participants randomly sampled into the Serum Random Immunogenicity Subset (Serum-RIS), as well as in all evaluable breakthrough COVID-19 endpoints outside the Serum-RIS, together forming the Serum-IAS. For evaluable COVID-19 endpoints (CDC criteria), COVID-19 was defined as in Garrett et al.[1], i.e., one Nucleic Acid Amplification Test (NAAT) positive nasal swab within 14 days of the onset of at least one systemic symptom or at least one respiratory sign/symptom, or

clinical or radiographical evidence of pneumonia) occurred between 7 to 230 days post Peak. The Serum-IAS included 152 evaluable breakthrough COVID-19 endpoints in the Hybrid Group (130 in PWH) and 54 in the Vaccine Group (49 in PWH). Based on this data availability, antibody marker correlates were assessed for each of the Hybrid and Vaccine Groups (pooling across PWH and PWoH to maximize statistical information). A subset of antibody marker correlates analyses were also done separately in PWH vs. PWoH, to evaluate whether the correlates differ in these groups.

### Participant demographics

Supplementary Table 1 summarizes demographic variables in the Serum-RIS, which represent the Hybrid [$N = 49$ males (14 PWH, 35 PWoH), mean age 40.8 years and $N = 89$ females (52 PWH, 37 PWoH), mean age 39.3 years] and Vaccine [$N = 36$ males (16 PWH, 20 PWoH), mean age 40.0 years and $N = 25$ females (21 PWH, 4 PWoH), mean age 42.5 years] Group populations for immunogenicity assessment. Within each Group (Hybrid or Vaccine) in Supplementary Table 1, demographic variables are reported for the PWH and PWoH subsets.

Comparing the PWH Hybrid vs. Vaccine groups, 21% vs. 43% were male, 48% vs. 57% were age > 40, 17% vs. 11% had a history of TB, 94% vs. 78% had a CD4 + T-cell count ≥ 200 cells/mm$^3$, and 79% vs. 65% had a viral load controlled below 50 copies/ml, suggesting the Hybrid Group may have had slightly better controlled HIV. For the Hybrid Group comparing PWH vs. PWoH, 21% vs. 49% were male, 48% vs. 49% were age > 40, 21% vs. 29% of males had BMI > 25, and 73% vs. 86% of females had BMI > 25. For the Vaccine Group comparing PWH vs. PWoH, 43% vs. 83% were male, 57% vs. 46% were age > 40, 31% vs. 5% of males had BMI > 25, and 48% vs. 50% of females had BMI > 25. Notable imbalances are greater proportions of female participants among PWH for both immunity groups, and greater proportions of males with BMI > 25 among PWH for the Vaccine but not Hybrid group.

### Immune response markers at M0 and Peak differ between the Hybrid and Vaccine Groups and are similar between people with and without HIV

In order to assess potential humoral immunogenicity differences at enrolment and at Peak, we plotted the distributions of the antibody response markers at M0 and Peak with stratification into four groups defined by (Hybrid, Vaccine) cross-classified with (PWH, PWoH) (Fig. 1). At M0, all antibody responses are higher in the Hybrid Group compared to the Vaccine Group ($p < 0.001$; Fig. 1a, c, e and Supplementary Table 2). At Peak, antibody responses are also higher in the Hybrid Group ($p < 0.001$; Fig. 1d, f and Supplementary Table 2). In the Hybrid Group, antibody responses are similar between PWH and PWoH at both time points ($p > 0.27$; Fig. 1e, f and Supplementary Table 2), whereas in the Vaccine Group, Peak responses are higher in PWH than PWoH, with a 0.23-log difference (95% CI 0.08, 0.36) for IgG Spike BA.5 and 0.48-log difference (0.17, 0.81) for 50% inhibitory serum dilution neutralizing antibody (nAb-ID50) BA.4/5 titer (Fig. 1f and Supplementary Table 2). Within PWH, Peak nAb-ID50 BA.4/5 titer appears to trend higher in participants in the Hybrid Group with higher baseline CD4 + T-cell count; this finding is less apparent in the Vaccine Group (Supplementary Fig. 2). In a post hoc linear regression analysis comparing Peak nAb-ID50 BA.4/5 titers among Serum-RIS PWH participants with baseline CD4 + T-cell count below vs. above 500 cells/mm$^3$ and adjusting for baseline nAb-ID50 BA.4/5 titer and Hybrid vs. Vaccine Group, participants with baseline CD4 + T-cell count <500 cells/mm$^3$ had 0.23-log lower Peak nAb-ID50 BA.4/5 titer, although this finding is not significant ($p = 0.27$).

### Correlations of immune response markers at M0, at Peak, and across time points

To understand the level of independent information provided by each antibody marker, which aids in planning and interpretation of the

subsequent correlates analyses, we investigated antibody marker intercorrelations based on the Serum-RIS. For the Hybrid Group, at M0, IgG binding antibody responses to different Spike antigens at M0 are highly correlated (e.g., Spearman ρ = 0.95 for IgG Spike BA.4/5 and IgG Spike Index, Supplementary Fig. 3A). The other antibody markers at M0 generally provide more independent information: IgG levels to BA.4/5 Spike and N Index have ρ = 0.61, and nAb-ID50 BA.4/5 titer and N Index have ρ = 0.55 (Supplementary Fig. 4). M0 IgG Spike BA.4/5 and M0 nAb-ID50 BA.4/5 were more highly correlated, with ρ = 0.80 (Supplementary Fig. 4).

For the Hybrid Group, the correlations among the antibody markers at Peak decreased strongly compared to at M0, e.g., ρ = 0.26 for nAb-ID50 BA.4/5 titer with IgG Spike BA.4/5 (Supplementary Fig. 5). After two doses (for the Vaccine Group) this correlation was also lower, but to a lesser extent (ρ = 0.57) (Supplementary Fig. 6). Lower correlations at Peak could potentially be explained by the observation that vaccination appeared to narrow the range of inter-individual variation in IgG Spike BA.4/5 levels (compare the Hybrid Group plots in Fig. 1c vs. those in Fig. 1d), and the fact that the majority of the participants in the Hybrid Group had post first vaccination values > 100,000 AU/ml vs. M0 values between 100 and 100,000 AU/ml.

Examining the correlation of each marker between the two time points M0 and Peak for the Hybrid Group, nAb-ID50 BA.4/5 titer and IgG Spike BA.4/5 have low cross time point correlation (ρ = 0.19, 0.36), and IgG N Index has high correlation (ρ = 0.95) (Supplementary Fig. 7). The latter result is expected (for a validated correctly performing assay) given that the mRNA-1273 vaccine did not include the N protein.

For the antibody markers at Peak studied as correlates, Supplementary Fig. 8 shows spaghetti plots connecting individuals' anti-Spike responses from M0 to Peak for the Hybrid and Vaccine groups.

Vaccination boosts anti-Spike responses in the majority of study participants in both the Hybrid and Vaccine groups.

## Higher nAb-ID50 BA.4/5 titers at enrolment are an independent inverse CoR of post-vaccination COVID-19 in the Hybrid Group

To investigate whether antibody marker levels differed between COVID-19 cases [including vaccination-proximal COVID-19 cases (in the first 3 months post Peak) and vaccination-distal cases (in months 3 to 7.5 post Peak)] and non-cases, we compared antibody response levels at M0 and at Peak across these groups (Fig. 2). We first show results for the Hybrid Group. At M0, the three antibody markers all have higher levels for non-cases compared to both sets of cases. The 95% CIs about the ratios of geometric mean (GM) values (Non-cases/All cases) surpass one (Table 1). Antibody markers at Peak are generally higher in non-cases than cases, more so against vaccination-proximal than -distal cases, although with 95% CIs about the GM ratios including one (Table 1, Fig. 2). The difference in antibody marker levels between non-cases and cases is smaller at Peak compared to at M0.

Supplementary Fig. 9 shows the distributions of the two fold-rise (M0 to Peak) antibody markers for IgG Spike BA.4/5 and nAb-ID50 BA.4/5, in cases and in non-cases in the Hybrid Group. Among cases and among non-cases, the median fold-rise was higher for nAb-ID50 BA.4/5 than for IgG Spike BA.4/5, and for each antibody marker the fold-rise was higher in cases than in non-cases. Specifically, the median fold-rise of IgG Spike BA.4/5 on the $\log_{10}$ scale was 1.35 (22-fold increase) for cases and 0.97 (9-fold increase) for non-cases, and for nAb-ID50 BA.4/5 it was 1.54 (35-fold increase) for cases and 1.08 (12-fold increase) for non-cases. This result is consistent with the observation above that the difference in antibody marker levels between non-cases and cases is smaller at Peak compared to at M0.

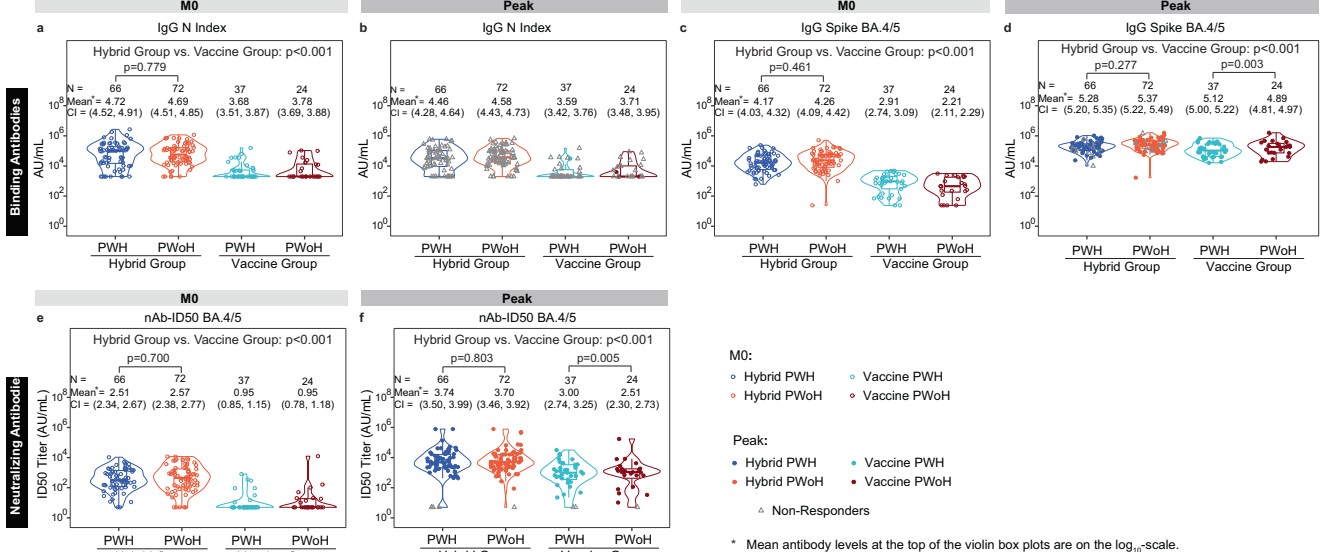

**Fig. 1 | Distribution of antibody response markers at M0 and at Peak in the four participant groups defined by (Hybrid Group, Vaccine Group) cross-classified with (PWH, PWoH).** The cohorts are random samples from the four groups, comprising the Serum-RIS. **a**, **b** IgG N Index; **c**, **d** IgG Spike BA.4/5; **e**, **f** nAb-ID50 BA.4/5. Violin plots contain interior box plots with upper and lower horizontal edges representing the 25th and 75th percentiles of antibody level and middle line representing the 50th percentile. Vertical bars represent the distance from the 25th (or 75th) percentile of antibody level and the minimum (or maximum) antibody level within the 25th (or 75th) percentile of antibody level minus (or plus) 1.5 times the interquartile range. Each side shows a rotated probability density of the data. Covariate-adjusted, mean titer level (in the $\log_{10}$-scale) with 95% confidence intervals and two-sided P-values based on non-parametric bootstrap (see Methods for details of how the bootstrapped p-values were obtained based on g-computation) are given above violin boxplots. Analyses adjusted for HIV status at baseline, age, sex, and BMI (for M0 comparisons of Hybrid Group vs. Vaccine Group and for Peak comparisons of Hybrid Group vs. Vaccine Group); age, sex, and BMI (for M0 comparisons of Hybrid PWH vs. Hybrid PWoH, Peak comparisons of Hybrid PWH vs. Hybrid PWoH, and Peak comparisons of Vaccine PWH vs. Vaccine PWoH). M0 Vaccine PWH and Vaccine PWoH GMVs and 95% CIs shown were also adjusted by age, sex, and BMI. No adjustment was made for multiple comparisons. Peak, 4 weeks post last vaccine dose (M1, Hybrid Group; M2, Vaccine Group). N Nucleocapsid protein, nAb-ID50 50% inhibitory serum dilution neutralizing antibody, PWH people with HIV, PWoH people without HIV, RIS random immunogenicity subset.

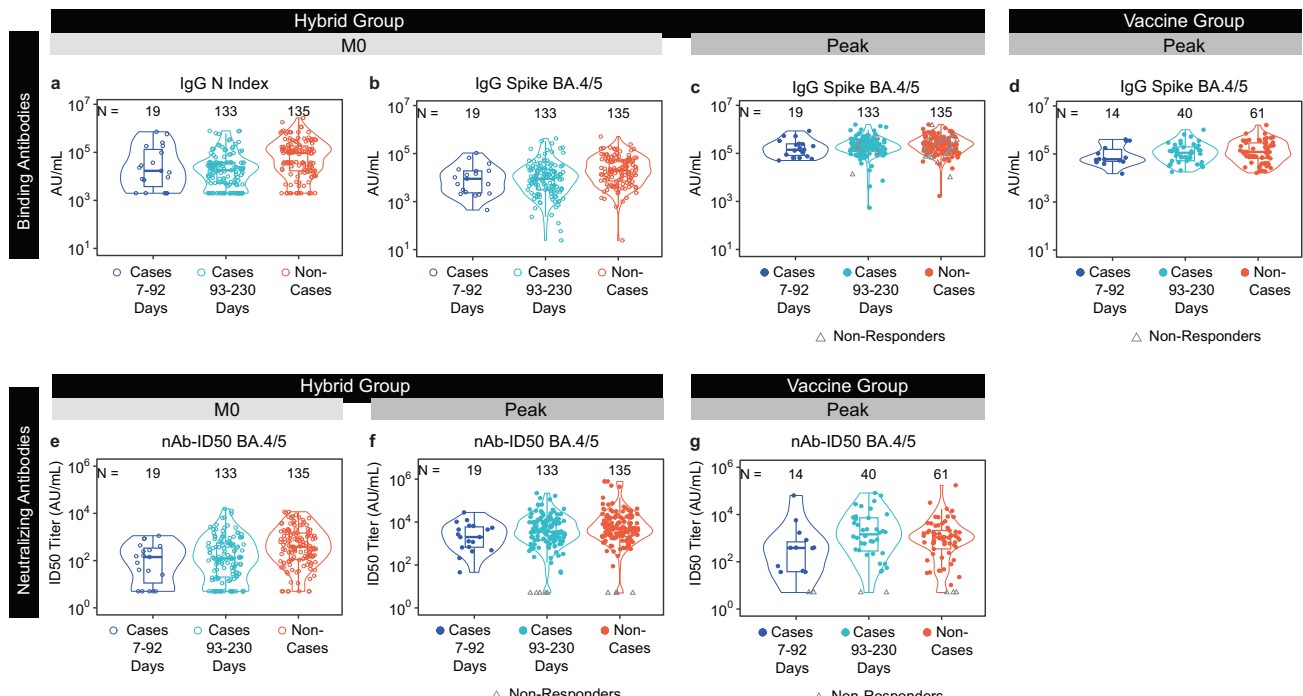

**Fig. 2 | Antibody response levels at M0 or at Peak by COVID-19 outcome status.** Data points are from eligible COVID-19 cases and non-cases in the Per-protocol Serum Immunogenicity Analysis Set. The plots show immune markers for which correlates of risk are assessed. Plots are shown for **a–c**, **e**, **f** Hybrid Group or **d**, **g** Vaccine Group levels of **a** IgG N Index at M0; **b** IgG Spike BA.4/5 at M0; **c**, **d** IgG Spike BA.4/5 at Peak; **e** nAb-ID50 BA.4/5 at M0; and **f**, **g** nAb-ID50 BA.4/5 at Peak. The violin plots contain interior box plots with upper and lower horizontal edges representing the 25th and 75th percentiles of antibody level and middle line representing the 50th percentile. The vertical bars represent the distance from the 25th (or 75th) percentile of antibody level and the minimum (or maximum) antibody level within the 25th (or 75th) percentile of antibody level minus (or plus) 1.5 times the interquartile range. Each side shows a rotated probability density (estimated by a kernel density estimator with a default Gaussian kernel) of the data. Positive response at M0 and vaccine-response at Peak for each antibody marker is defined in Table 1. Cases acquired a COVID-19 endpoint 7 days post Peak through 92 days post Peak or 93 days post Peak through 230 days post Peak, as designated in the key at the bottom of each panel. Non-cases did not have a positive RT-PCR result at the Peak visit and did not acquire a COVID-19 endpoint after M0 up to the date by which the last enrolled participant reached 230 days post Peak (March 31, 2023). Peak, 4 weeks post last vaccine dose (M1 for Hybrid Group, M2 for Vaccine Group). The frequency of COVID-19 endpoints through 230 days post Peak that were BA.4 or BA.5 was 46.6% for Hybrid and 30.0% for Vaccine (Fig. 4a in ref. 1). N, Nucleocapsid protein; nAb-ID50, 50% inhibitory serum dilution neutralizing antibody; PWH, people with HIV. PWoH, people without HIV.

Figure 3 shows the results of two types of correlate of risk analyses in the Hybrid Group. First, based on proportional hazards models adjusting for region, PWH status, TB status, enrolment period, and baseline risk score, IgG Spike BA.4/5 and nAb-ID50 BA.4/5 levels at M0 inversely correlated with risk of COVID-19 over 92 days follow-up post Peak ($p = 0.017$, $p = 0.003$). IgG N at M0 trended toward inverse correlation, but this association was not significant ($p = 0.079$) (Fig. 3a). The inverse correlations of M0 antibody levels with COVID-19 also hold over a longer follow-up, 230 days post Peak ($p = 0.032$, 0.002, and 0.064, respectively) (Fig. 3B). However, the markers at Peak do not correlate with COVID-19 ($p > 0.50$ for the 230 day follow-up) (Fig. 3b). These results are recapitulated by the fold-rise in antibody levels being significantly directly correlated with COVID-19 (Fig. 3b), as the fold-rise inversely correlated with the antibody level at M0 (Supplementary Fig. 12).

To address the question of whether the identified antibody correlates still associate with COVID-19 risk in the Hybrid Group even after controlling for the other antibody correlates that were identified, i.e., which antibody markers are independent correlates, multivariable Cox models including pairs of antibody markers were analyzed. Results from a Cox model including IgG Spike BA.4/5 at both M0 and Peak (Table 2, model 4) and from a Cox model including nAb-ID50 BA.4/5 at both M0 and Peak (Table 2, model 5) showed that the M0 markers are inverse CoRs of COVID-19 through 230 days post Peak (HRs = 0.44, 0.46; $p$-values 0.029, 0.002), while the Peak markers are not (HRs = 1.86, 1.13; $p$-values 0.21, 0.45). Models with any pair of the three antibody markers at M0 (Table 2, models 1, 2, and 3) showed that nAb-ID50

BA.4/5 is the independent correlate (models 2 and 3: HRs = 0.50, 0.36; $p$-values 0.018, 0.004), i.e., is still statistically associated with COVID-19 risk even after the model controls for any of the other markers considered in the analysis. Moreover, comparing to the univariable Cox model results (HR = 0.46, $p$-value 0.002, Fig. 3b), the strength of the inverse correlation of M0 nAb-ID50 BA.4/5 with COVID-19 risk remains similar whether or not M0 IgG Spike or M0 N levels are accounted for. Given that N levels are a proxy for recency or for the severity of prior illness, these results support robustness of the neutralization correlate to this aspect of hybrid immunity heterogeneity.

We systematically compared the goodness-of-fit of various Cox models with different sets of covariates. Adding M0 nAb-ID50 to a model containing only baseline covariates significantly improved model fit ($p = 0.004$; Table 3). However, further inclusion of M0 or Peak Ab markers (e.g., M0 IgG N Index or Peak nAb-ID50) did not lead to significant improvements (Table 3).

Figure 3c–g shows the results of the controlled-risk analyses that estimate the probability of COVID-19 by 165 days post Peak visit under hypothetical assignments of all participants to the Hybrid Group and a given value of a marker at M0 or at Peak. COVID-19 risk decreases with increasing antibody level at M0 (Fig. 3c–e), but remains generally flat with increasing antibody level at Peak (Fig. 3f–g), with a decrease in risk only observed at increases at the highest antibody levels.

For the same markers studied in controlled-risk analyses, covariate-adjusted cumulative incidence of COVID-19 through 160 days post Peak in subgroups defined by M0 or Peak marker value above a specified threshold was estimated, across the spectrum of

possible thresholds (Supplementary Fig. 13). Interestingly there is a consistent and gradual drop in COVID-19 risk with increasing thresholded M0 anti-N Index concentration, whereas for Peak nAb-ID50 BA.4/5, COVID-19 risk is constant across titers except with a precipitous drop at high titers, suggesting a threshold effect. For example, the probability of COVID-19 by 160 days post Peak is 0.047 (0.040, 0.053) for all Hybrid Group participants and precipitously drops to 0.0077 (0.0017, 0.014) for the subset of Hybrid Group participants with Peak nAb-ID50 BA.4/5 titer exceeding 79,200 AU/ml (6.1-fold reduction in risk).

**Vaccine Group Peak antibody levels inversely correlate with risk of COVID-19, but only in early vaccination-proximal follow-up**

In the Vaccine Group, Peak antibody marker levels are higher in non-cases than Peak antibody marker levels in vaccination-proximal cases, whereas Peak antibody marker levels are similar in non-cases compared to vaccination-distal cases (Table 4, Fig. 2). Peak levels are considerably lower for the Vaccine Group compared to the Hybrid Group (GM nAb-ID50 BA.4/5 titer 757 and 4700 AU/ml for non-cases, respectively) (Tables 4 and 1, respectively).

Figure 4 shows results from the same correlate of risk analyses as in Fig. 3, but for the Vaccine Group. Based on covariate-adjusted proportional hazards models, nAb-ID50 BA.4/5 Peak titer inversely correlated with COVID-19 over 92 days follow-up post Peak ($p = 0.015$) (Fig. 4a). However, this correlate does not hold over the longer follow-up period of 230 days follow-up post Peak ($p = 0.959$) (Fig. 4b). Peak IgG Spike BA.4/5 concentration trends toward an inverse CoR over the 92 day follow-up period (HR per 10-fold increase of 0.56, 95% CI: 0.14, 2.21), but with a wide confidence interval and a $p$-value of 0.41 (Fig. 4a). No correlation is seen over the longer 230 day follow-up.

Similar results were seen in the controlled risk analyses for the Vaccine Group, with COVID-19 risk through 92 days post Peak decreasing with both Peak antibody markers (Fig. 4c, d). Covariate-adjusted COVID-19 risk over 92 days post Peak equals 1.9% at a Peak nAb-ID50 BA.4/5 titer of 10 AU/ml, 1.1% at 100 AU/ml, and 0.3% at 1000 AU/ml (Fig. 4d). In contrast, the curve of COVID-19 risk over a longer follow-up period, 165 days post Peak, is flat (Fig. 4f). Similar attenuation of the correlate over the longer follow-up period is seen for Peak IgG Spike BA.4/5 (Fig. 4e). As a sensitivity analysis, Supplementary Fig. 14 repeats the analyses through 130 days post Peak and shows similarly flat curves of COVID-19 risk for both Peak antibody markers.

For the controlled risk analyses of thresholded antibody marker levels in the Vaccine Group, covariate-adjusted cumulative incidence of COVID-19 through 92 days post Peak in subgroups defined by Peak marker value above a specified threshold shows a decrease with increasing Spike IgG BA.4/5 and nAb-ID50 BA.4/5 thresholds, with a greater degree of decrease for the Spike IgG marker (Supplementary Fig. 15A, B). In contrast, COVID-19 risk through a longer follow-up, 160 days post Peak, is flat with increasing Spike IgG and nAb-ID50 thresholds (Supplementary Fig. 15C, D).

**Antibody marker correlates of risk are similar for people with and without HIV**

Since prior studies demonstrated that PWH have lower antibody responses than PWoH to SARS-CoV-2 acquisition[5], and the current study suggests antibody responses in the Vaccine Group may be higher in PWH, we next assessed whether the correlates of risk would differ in PWH. Supplementary Fig. 16 repeats the Cox model antibody marker correlates analyses restricting to PWH, showing similar results to the overall analyses. Tests for whether HIV status modifies the correlate provided no evidence of effect modification, with one possible exception: M0 IgG N (vaccine-matched) had $p$-value 0.011 with HR = 0.52 (0.31, 0.86) for PWH and HR = 1.54 (0.40, 5.94) for PWoH (Supplementary Table 3); this result did not pass multiplicity correction (multiplicity-adjusted $p = 0.077$).

**Correlates of protection analyses support that the Hybrid Group advantage over the Vaccine Group was partially mediated through the antibody markers**

To further test the role of prior immunity through neutralizing antibodies, we estimated the controlled relative risk of COVID-19 through 165 days post Peak for the Hybrid Group, which is the controlled risk by M0 antibody level for the Hybrid Group divided by the overall risk for the Vaccine Group (Supplementary Fig. 17). This analysis shows controlled relative risks greater than one (i.e., higher risk of COVID-19) at low antibody levels for the Hybrid Group and approaching 0 (i.e., lowest risk of COVID-19) at high antibody levels for the Hybrid Group.

To test whether there were specific thresholds of antibody levels marking high efficacy of hybrid vs. vaccine immunity, Fig. 5 shows thresholded relative risk by 160 days post Peak for Hybrid Group participants with M0 antibody marker level above a threshold vs. overall risk for the Vaccine Group. The relative risk was 0.46 (0.28, 0.64) when thresholding Hybrid Group participants by M0 nAb-ID50 BA.4/5 titer above 10 AU/ml (5th percentile) and decreased to 0.23 (0.10, 0.35) at titer above 1000 AU/ml (70th percentile).

**Lower antibody responses against the vaccine strain compared to against the BA.5 circulating strain**

The correlates analyses focused on responses measured against the circulating strain BA.4/5. Pre-vaccination antibody levels against BA.4/5 in the Hybrid Group were about 6-fold lower than against Reference (Supplementary Fig. 10, where for non-cases GM nAb-ID50 against Reference and BA.4/5 was 2233 and 368 AU/ml, respectively). Peak antibody levels in both the Hybrid and Vaccine groups were also about 6-fold lower against BA.4/5 (Supplementary Fig. 11, where for Vaccine Group non-cases GM nAb-ID50 against Reference and BA.4/5 was 5279 and 757 AU/ml, respectively).

**Quantitative comparison of CoVPN 3008 nAb-ID50 correlates results to those of previous mRNA-1273 vaccine immune correlates studies**

To interpret the CoVPN 3008 results, we derived head-to-head comparisons of the relationship between Peak neutralization titers and COVID-19 risk for CoVPN 3008 vs. two recent immune correlates studies of the same mRNA-1273 vaccine: COVE 3-dose mRNA-1273[39] and COVAIL second-booster mRNA-1273[42] (hereafter "COVE" and "COVAIL"). All COVE correlates study participants received three mRNA-1273 doses (Prototype) and all COVAIL correlates study participants received one dose of an mRNA-1273 vaccine (Prototype, Beta+Omicron BA.1, Delta + Omicron BA.1, Omicron BA.1, or Omicron BA.1+Prototype mRNA-1273) as their second booster dose. Both studies were conducted in the United States, predominantly in PWoH. Comparisons were made possible by assay concordance studies that place the nAb-ID50 titers on the same scale (see Methods). Figure 6 shows the controlled-risk by Peak nAb-ID50 titer curves, for each trial using nAb-ID50 titers measured against the predominant circulating variant during the trial follow-up for evaluating correlates. The results are shown separately for 92 days and 165 days of follow-up post Peak, as well as separately for the Hybrid and Vaccine groups, where Hybrid vs. Vaccine is defined by SARS-CoV-2 positive vs. negative at the time of the first dose (CoVPN 3008), second boost post-primary series (COVAIL), or third dose (COVE). Figure 6 also compares Peak nAb-ID50 titers across the trials, which we interpret first for contextualizing the correlates comparison. For the Hybrid Group, titers were similar in COVAIL and CoVPN 3008 and lowest in COVE (Fig. 6e, g, i), which may be explained by the fact that the COVE correlates study took place about 6 months earlier, such that many previous infections were with pre-Omicron viruses. For the Vaccine Group, titers were also similar in COVAIL and CoVPN 3008 and lowest in COVE (Fig. 6f, h, j), where the lower titers in COVE vs. COVAIL may be explained by a 4th vs. 3rd vaccination, and the reasons for lower titers in COVE vs. CoVPN 3008 are not clear given CoVPN 3008 participants received only 2 vaccinations.

**Table 1 | In the Hybrid Group (baseline SARS-CoV-2 positive and received 1 mRNA-1273 dose), vaccine-response frequencies at Peak by COVID-19 outcome status, geometric means (GMs) by COVID-19 outcome status, and their comparisons between non-cases and cases**

| Marker | COVID-19 Cases[a] | | | Non-Cases[b] | | | Comparison | |
|---|---|---|---|---|---|---|---|---|
| | N | Proportion with Response over Baseline (95% CI)[c] | Geometric Mean (GM) (95% CI) | N | Proportion with Response over Baseline (95% CI)[c] | Geometric Mean (GM) (95% CI) | Response Frequency Difference (Non-Cases – Cases) | Ratio of GM (Non-Cases/Cases) |
| M0 IgG N Index (AU/ml) | 152 | N/A | 17622 (13546, 22925) | 135 | N/A | 55948 (41404, 75602) | N/A | 3.17 (2.14, 4.72) |
| M0 IgG Spike BA.4/5 (AU/ml) | 152 | N/A | 7900 (6103, 10227) | 135 | N/A | 18477 (14457, 23614) | N/A | 2.34 (1.64, 3.34) |
| M0 nAb ID50 BA.4/5 (AU/ml) | 152 | N/A | 109 (77.8, 152.7) | 135 | N/A | 367.8 (267, 507) | N/A | 3.37 (2.12, 5.38) |
| Peak IgG N Index (AU/ml) | 152 | 0.7% (0.1%, 3.7%) | 15110 (11958, 19093) | 135 | 0.0% (0.0%, 2.8%) | 33963 (25959, 44434) | −0.7% (−2.6%, 1.3%) | 2.25 (1.58, 3.2) |
| Peak IgG Spike BA.4/5 (AU/ml) | 152 | 92.7% (87.4%, 95.9%) | 178506 (152321, 209192) | 135 | 77.8% (70.0%, 84.0%) | 213245 (181560, 250459) | −14.9% (−23.8%, −6.2%) | 1.19 (0.95, 1.5) |
| Peak nAb ID50 BA.4/5 (AU/ml) | 152 | 96.7% (92.5%, 98.6%) | 3022 (2212, 4130) | 135 | 96.3% (91.6%, 98.4%) | 4700 (3306, 6682) | −0.4% (−5.1%, 4.3%) | 1.55 (0.98, 2.48) |
| Peak IgG Spike BA.4/5 (AU/ml) vs. D7-92 cases[d] | 19 | 100% (83.2%, 100.0%) | 161629 (109200, 239232) | 135 | 77.8% (70.0%, 84.0%) | 213245 (181560, 250459) | −22.2% (15.2%, 29.2%) | 1.32 (0.84, 2.07) |
| Peak nAb ID50 BA.4/5 (AU/ml) vs. D7-92 cases[d] | 19 | 100% (83.2%, 100.0%) | 1924 (899, 4116) | 135 | 96.3% (91.6%, 98.4%) | 4700 (3306, 6682) | −3.7% (0.52%, 6.89%) | 2.44 (0.92, 6.48) |

[a]Cases acquired a COVID-19 endpoint 7 days post Peak through 230 days post Peak.

[b]Non-cases did not have a positive RT-PCR result at the Peak visit and did not acquire a COVID-19 endpoint after M0 up to the date by which the last enrolled participant reached 230 days post Peak (March 31, 2023).

[c]At Peak, a vaccine response (sero-response) for IgG N Index or IgG Spike BA.5 is defined as participants with value above the antigen-specific positivity cut-off (if the M0 readout is below the positivity cut-off) or is at least 4-fold above the M0 value (if the M0 readout is above the positivity cut-off) (positivity cut-off = 3970 AU/ml for N Index and 48.6 AU/ml for Spike BA.5); positive sero-response for nAb ID50 BA.4/5 is defined as participants with value above the LOD (if the M0 readout is below the LOD) or is at least 4-fold above the M0 value (if the M0 readout is above the LOD).

[d]For these rows, cases restrict to those who acquired a COVID-19 endpoint 7 days post Peak through 92 days post Peak.

Peak, 4 weeks post last vaccine dose (M1 for Hybrid Group); N, Nucleocapsid protein; nAb-ID50, 50% inhibitory serum dilution neutralizing antibody titer.

Analysis based on eligible COVID-19 cases[a] and non-cases[b] from the Per-protocol Serum Immunogenicity Analysis Set.

COVID-19 risk was considerably greater in the two United States studies compared to CoVPN 3008 (Fig. 6a–d), especially for the first 92 days of follow-up. In addition, COVID-19 cases occurred later after vaccination in CoVPN 3008 than in COVAIL and COVE, with 16%, 60%, and 94% of the COVID-19 endpoints by 92 days post Peak in the three studies, respectively, and 75% of the CoVPN 3008 cases after 110 days. The controlled risk curves show consistency of the correlates results across the trials over the 92-day follow-up period (Fig. 6a, b) in that over the range of overlapping titers, COVID-19 risk decreases with increasing Peak nAb-ID50 titer, in both the Hybrid and Vaccine groups. [This finding is consistent with the previously reported HRs (95% CIs) of COVID-19 per 10-fold increase in Peak nAb-ID50 BA.1 titer: 0.31 (0.10, 0.96) in the naïve (Vaccine) cohort and 0.28 (0.07, 1.08) in the non-naïve (Hybrid) cohort over 92 days post-dost 3 in the COVE booster study (Fig. 2e in Zhang et al.[39]), as well as the HR of COVID-19 per 10-fold increase in Peak nAb-ID50 BA.4/5 titer in the non-naïve (Hybrid) cohort over 92 days post-second booster for one-dose mRNA-1273 second boost recipients in COVAIL study, 0.35 (0.20, 0.62). The corresponding HR for the naïve (Vaccine) cohort was 0.88 (0.65, 1.19)].

However, over the 165-day follow-up period (for which the analysis could be done for COVAIL and CoVPN 3008 but not COVE), the decrease in COVID-19 risk with increasing Peak nAb-ID50 titer is still observed in COVAIL (Fig. 6c, d), but in CoVPN 3008, the relationship between risk and Peak nAb-ID50 titer appears flat in both the Hybrid (Fig. 6c) and Vaccine (Fig. 6d) Groups. This attenuation of the correlate in CoVPN 3008 over the longer follow-up period can be explained by the fact that the COVID-19 cases in CoVPN 3008 occurred much later after vaccination than in COVAIL. Weakening of the nAb-ID50 correlate over longer-term follow-up is consistent with the previous correlates results in non-naive (Hybrid) COVAIL one-dose mRNA-1273 second boost recipients (the same 5 vaccine arms referred to above)[42]: The reported HR of COVID-19 from 22 through 203 days post-second boost per 10-fold increase in Peak nAb-ID50 BA.4/5 titer was 0.44 (0.26, 0.73) (Fig. 5c in ref. 42). When breaking this follow-up period into vaccination-proximal and vaccination distal periods, the HR (95% CI) of COVID-19 per 10-fold increase in Peak nAb-ID50 BA.1 titer was 0.87 (0.27, 2.79) for 107 through 203 days post-second boost (showing apparent attenuation of the correlate at time periods farther from second boost receipt), compared to 0.35 (0.20, 0.62) for 22 through 106 days post-second boost. For naïve (Vaccine) COVAIL one-dose mRNA-1273 second boost recipients, evidence supported weak inverse correlates at best over the 22 to 106- day and over the 22 to 203-day follow up period [HRs per 10-fold increase in Peak nAb-ID50 BA.4/5 titer of 0.88 (0.65, 1.19) and 0.79 (0.62, 1.01) (with the latter shown in Fig. 5a in ref. 42), respectively].

**Circulating strains have similar Spike sequence antigenic distances to the mRNA-1273 vaccine strain in CoVPN 3008 compared to the COVE and COVAIL correlates studies**

To provide context for interpreting the results above, we compared how well the vaccine strain matched the circulating strains in the three studies, given that CoVPN 3008 was conducted over a different period of the COVID-19 pandemic compared to COVE and COVAIL. Supplementary Fig. 18 shows distributions of Spike sequence physicochemical weighted Hamming distances (scaled as numbers of mutations) and RBD antigenic escape distances to the Ancestral mRNA-1273 vaccine strain of the SARS-CoV-2 sequences estimated to be circulating in the CoVPN 3008 correlates study compared to those in the COVE and COVAIL correlates studies. The antigenic escape distances are computed using the Bloom lab's antibody escape calculator[43]. The mean Spike weighted Hamming distance for CoVPN 3008, COVE, and COVAIL was 36.7, 37.3, and 35.9, respectively, with about 60% of viral distances ~2 mutations farther from the vaccine for COVE than CoVPN 3008, and about 20% of distances ~3-4 mutations farther from the vaccine for CoVPN 3008 than COVE. The mean RBD antigenic escape distance was 3.00, 2.58, and 2.73, respectively, with about 20% of viral distances showing substantially more escape from the vaccine for CoVPN 3008 compared to those for COVE. We draw two conclusions: (1) mismatch of the vaccine strain to the circulating strains may partly explain the CoVPN 3008 correlate in the Vaccine Group only being

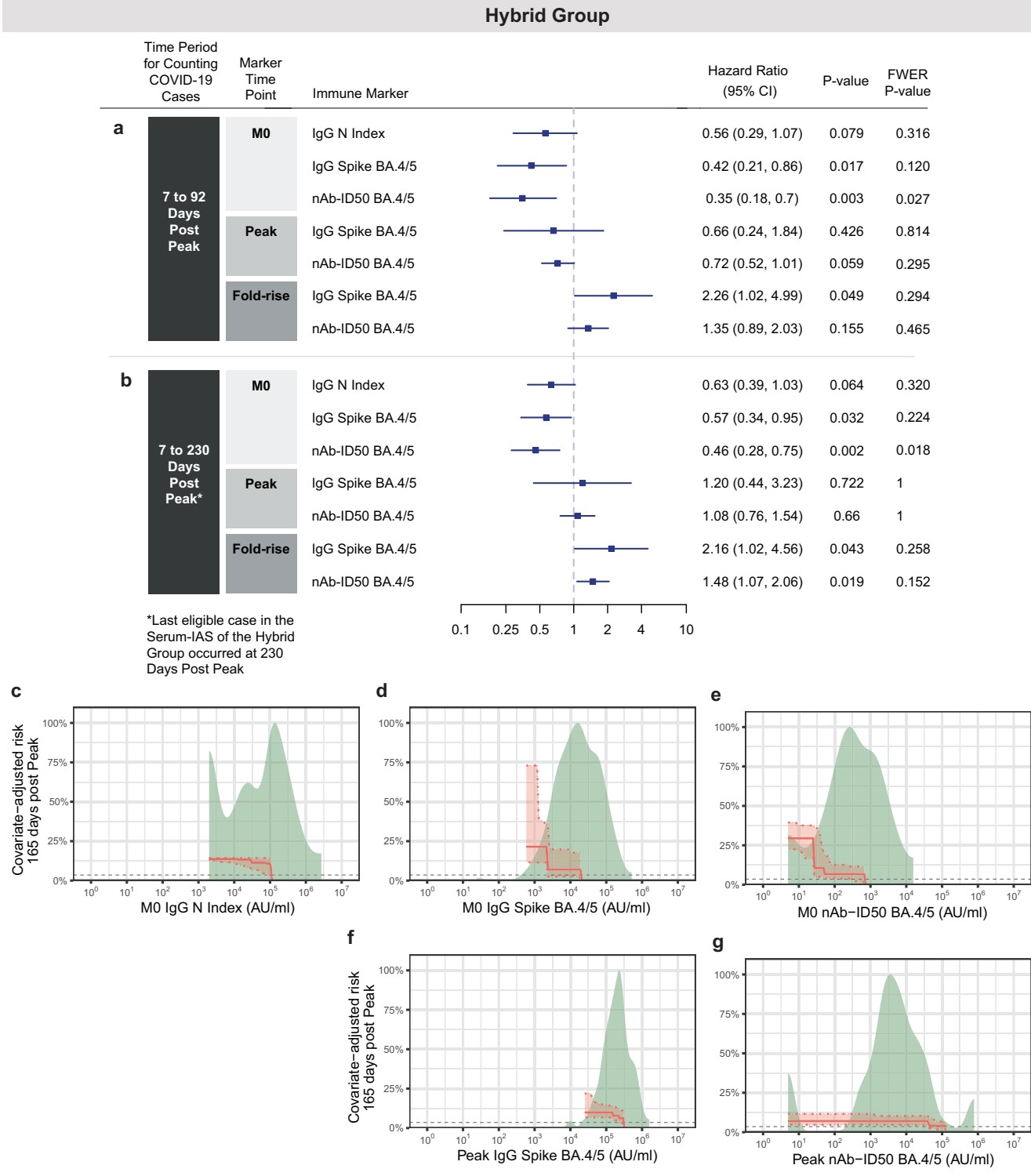

attained for vaccination-proximal follow-up; and (2) these results support a generally similar distribution of genetic and antigenic distances of circulating exposing strains across the three correlates studies.

## Discussion

This study supplies insights for mRNA-1273 vaccination in indicating: (1) no diminution in IgG Spike binding antibody and neutralizing antibody levels in PWH with good virologic control compared to PWoH in East and Southern Africa; (2) the antibody immune correlates of COVID-19 developed through many studies with only small numbers of

PWH continue to hold in an East and Southern African population with HIV; and (3) these antibody immune correlates can become weaker or shorter-lived in a naïve vaccine immunity population compared to a hybrid immunity population, either due to quantitative aspects of the responses like lower antibody levels and/or to the qualitative aspects that are more difficult to analyze like aspects related to B-cell memory, with both aspects relevant and connected to protection. We next discuss each of these conclusions in detail.

Our first major conclusion is no diminution of SARS-CoV-2-specific IgG Spike and neutralization responses to mRNA-1273 vaccination in a PWH population. This conclusion is supported by the

**Fig. 3 | Correlate of risk analyses in the Hybrid Group.** Analyses used eligible COVID-19 cases and non-cases in the Per-protocol Serum Immunogenicity Analysis Set in the Hybrid Group (*N* = 287). **a**, **b** Covariate-adjusted hazard ratios (HRs) of COVID-19 [follow-up: **a** 7 to 92 days post-Peak; **b** 7 to 230 days post-Peak] per 10-fold increase in immune marker at M0, at Peak, or for Peak/M0 fold-rise. Blue squares are point estimates and horizontal lines are 95% confidence intervals. *P*-values are from a two-sided Wald test. Holm-Bonferroni family-wise error rate (FWER) adjusted two-sided *p*-values are also shown. Cases acquired a COVID-19 endpoint **a** 7 through 92 days post-Peak or **b** 7 through 230 days post-Peak. HRs were estimated using inverse probability sampling weighted Cox regression models; 95% confidence intervals (CIs) and Wald-based p-values are shown. Analyses adjusted for: **a** HIV status and baseline risk score; **b** whether enrolled in South Africa, HIV status, TB status, enrolment period, and baseline risk score. **c**–**g** Controlled risk plots for Hybrid Group (*N* = 287) antibody markers at M0 and Peak. Covariate-adjusted probability of COVID-19 over 165 days post-Peak is estimated

under hypothetical assignments of all participants to the Hybrid Group and the marker value on the x-axis. **c** M0 IgG N Index concentration, **d** M0 IgG Spike BA.4/5 concentration, **e** M0 nAb ID50 BA.4/5 titer, **f** Peak IgG Spike BA.4/5 concentration, **g** Peak nAb ID50 BA.4/5 titer. Controlled risk was estimated using a monotone-constrained nonparametric method with covariate adjustment and restricted to the middle 95% quantiles of available data. Solid lines indicate point estimates; dotted lines and shading indicate pointwise 95% CIs. Horizontal gray lines: overall cumulative incidence of COVID-19 from 7 to 165 days post-Peak in the Hybrid Group. Background kernel density plots show marker distributions at M0 or at Peak. Analyses adjusted for whether enrolled in South Africa, HIV status, TB status, enrolment period, and baseline risk score. AU, arbitrary units; FWER, family-wise error rate adjusted *p*-value; N, Nucleocapsid protein; nAb-ID50, 50% inhibitory serum dilution neutralizing antibody titer; Peak, 4 weeks post-last dose (M1 for Hybrid Group).

### Table 2 | Cox proportional hazards models of 2 antibody markers with baseline covariate adjustment for the Hybrid Group[a]

| Model[b] | Variables | Hazard Ratio (95% CI) | P-value[c] | Overall P-value[d] |
|---|---|---|---|---|
| 1 | M0 IgG N Index | 0.74 (0.40, 1.35) | 0.32 | 0.043 |
|  | M0 IgG Spike BA.4/5 | 0.66 (0.34, 1.27) | 0.21 |  |
| 2 | M0 IgG N Index | 0.87 (0.48, 1.55) | 0.63 | 0.006 |
|  | M0 nAb-ID50 BA.4/5 | 0.50 (0.28, 0.89) | 0.018 |  |
| 3 | M0 IgG Spike BA.4/5 | 1.48 (0.66, 3.29) | 0.34 | 0.005 |
|  | M0 nAb-ID50 BA.4/5 | 0.36 (0.18, 0.73) | 0.004 |  |
| 4 | M0 IgG Spike BA.4/5 | 0.44 (0.21, 0.92) | 0.029 | 0.086 |
|  | Peak IgG Spike BA.4/5 | 1.86 (0.71, 4.86) | 0.21 |  |
| 5 | M0 nAb-ID50 BA.4/5 | 0.46 (0.28, 0.74) | 0.002 | 0.007 |
|  | Peak nAb-ID50 BA.4/5 | 1.13 (0.82, 1.57) | 0.45 |  |

nAb-ID50, 50% inhibitory serum dilution neutralizing antibody titer. Peak, 4 weeks post last vaccine dose (M1 for Hybrid Group).
[a]Data points are from eligible COVID-19 cases (7 to 230 days post Peak) and non-cases in the Per-protocol Serum Immunogenicity Analysis Set.
[b]Adjusted for the baseline covariates whether enrolled in South Africa, PWH status, TB status, enrolment period (< 3 months, 3–6 months, > 6 months post first person enrolled), and baseline risk score.
[c]P-values are from a two-sided Wald test and are associated with the hazard ratio in the third column.
[d]The Overall P-value is from a generalized two-sided Wald test of the null hypothesis that both hazard ratios from the model are unity.

### Table 3 | Likelihood ratio tests assessing if adding additional variables significantly improves the Cox model fit

| Variables | Added Variables | P-value[a] |
|---|---|---|
| Baseline covariates | M0 nAb-ID50 BA.4/5 | 0.004 |
| Baseline covariates + M0 nAb-ID50 BA.4/5 | M0 IgG N Index | 0.66 |
| Baseline covariates + M0 nAb-ID50 BA.4/5 | M0 IgG Spike BA.4/5 | 0.32 |
| Baseline covariates + M0 nAb-ID50 BA.4/5 | M0 IgG N Index &M0 IgG Spike BA.4/5 | 0.53 |
| Baseline covariates + M0 nAb-ID50 BA.4/5 | Peak nAb-ID50 BA.4/5 | 0.45 |
| Baseline covariates + M0 nAb-ID50 BA.4/5 | Peak IgG Spike BA.4/5 | 0.29 |
| Baseline covariates + M0 nAb-ID50 BA.4/5 | Peak nAb-ID50 BA.4/5 &Peak IgG Spike BA.4/5 | 0.47 |

Variables: predictors included in the restricted model. Added variables: predictors further added to the restricted model.
nAb-ID50, 50% inhibitory serum dilution neutralizing antibody titer.
[a]Two-sided *P*-values are based on the partial-likelihood ratio test[64] implemented by the function regTermTest in the `survey` package. The *P*-values tested the null hypothesis that the coefficients of added variables (one or more) all equal 0.

finding that in both the Vaccine Group and the Hybrid Group, neither Peak pseudovirus neutralizing antibody titers nor Peak IgG binding antibodies against Spike were diminished in PWH compared to PWoH. However, this is accompanied by the fact that the PWH cohort in this study had well-controlled HIV (majority on antiretroviral therapy and with HIV viral load <50 copies/ml); thus, our conclusion may not generalize to all PWH. Surprisingly, in the Vaccine Group, Peak marker levels were significantly higher in PWH. We conjecture that this result may be caused by unmeasured confounding (such as differential antiretroviral therapy use, or differential rates of coinfections) that led to apparently higher responses in PWH in the Vaccine Group.

Our second major conclusion is that the antibody immune correlates of COVID-19 risk identified in previous studies (which enrolled no PWH or only small numbers of PWH) also hold in the studied East and Southern African population with HIV. In support of this conclusion, statistical tests for whether the inverse CoR of COVID-19 differed in PWH vs. PWoH did not find evidence for a difference in these two populations, with none of the interaction tests for a differential correlate passing multiplicity correction.

Our third major conclusion is that the antibody immune correlates can become weaker or shorter-lived in a naïve vaccine immunity population compared to a hybrid immunity population, with evidence

**Table 4 | In the Vaccine Group (SARS-CoV-2 baseline negative and received 2 mRNA-1273 doses), vaccine-response frequencies at Peak by COVID-19 outcome status, geometric means (GMs) by COVID-19 outcome status, and their comparisons between non-cases and cases**

| Marker | COVID-19 Cases[a] | | | Non-Cases[b] | | | Comparison | |
| | N | Proportion with Response over Baseline (95% CI)[c] | Geometric Mean | N | Proportion with Response over Baseline (95% CI)[c] | Geometric Mean | Response Frequency Difference (Non-Cases – Cases) | Ratio of GM (Non-Cases/Cases) |
|---|---|---|---|---|---|---|---|---|
| M0 IgG N Index (AU/ml) | 54 | N/A | 4837 (3661, 6391) | 61 | N/A | 4831 (3481, 6703) | N/A | 1.0 (0.65, 1.54) |
| M0 IgG Spike BA.4/5 (AU/ml) | 54 | N/A | 586 (376, 913) | 61 | N/A | 538 (359, 806) | N/A | 0.92 (0.51, 1.66) |
| M0 nAb ID50 BA.4/5 (AU/ml) | 54 | N/A | 13.8 (8.1, 23.3) | 61 | N/A | 10.5 (7.1, 15.7) | N/A | 0.76 (0.4, 1.45) |
| Peak IgG N Index (AU/ml) | 54 | 3.9% (1.1%, 13.2%) | 4342 (3382, 5575) | 61 | 1.8% (0.3%, 9.3%) | 4342 (3172, 5945) | −2.1% (−9.8%, 5.6%) | 1 (0.67, 1.5) |
| Peak IgG Spike BA.4/5 (AU/ml) | 54 | 100.0% (93.4%, 100.0%) | 102310 (78986, 132523) | 61 | 100.0% (94.0%, 100.0%) | 120017 (91598, 157253) | 0% (0%, 0%) | 1.17 (0.81, 1.7) |
| Peak nAb ID50 BA.4/5 (AU/ml) | 54 | 92.6% (82.5%, 97.1%) | 788 (396, 1566) | 61 | 95.1% (86.5%, 98.3%) | 757 (439, 1306) | 2.5% (−8.1%, 13.1%) | 0.96 (0.41, 2.27) |
| Peak IgG Spike BA.4/5 (AU/ml) vs. D7-92 cases[d] | 14 | 100% (78.5%, 100.0%) | 84534 (48463, 147451) | 61 | 100.0% (94.0%, 100.0%) | 120017 (91598, 157253) | 0% (−1.55%, 4.83%) | 1.42 (0.77, 2.62) |
| Peak nAb ID50 BA.4/5 (AU/ml) vs. D7-92 cases[d] | 14 | 85.7% (60.1%, 96.0%) | 230 (51, 1045) | 61 | 95.1% (86.5%, 98.3%) | 757 (439, 1306) | 9.4% (−28.48%, 9.75%) | 3.29 (0.89, 12.26) |

Analysis based on eligible COVID-19 cases[a] and non-cases[b] from the Per-protocol Serum Immunogenicity Analysis Set.

Peak, 4 weeks post last vaccine dose (M2 for Vaccine Group); N Nucleocapsid protein, nAb-ID50, 50% inhibitory serum dilution neutralizing antibody titer.

[a]Cases acquired a COVID-19 endpoint 7 days post Peak through 230 days post Peak.

[b]Non-cases did not have a positive RT-PCR result at the Peak visit and did not acquire a COVID-19 endpoint after M0 up to the date by which the last enrolled participant reached 230 days post Peak (March 31, 2023).

[c]At Peak, a vaccine response (sero-response) for IgG N Index or IgG Spike BA.5 is defined as participants with value above the antigen-specific positivity cut-off (if the M0 readout is below the positivity cut-off) or is at least 4-fold above the M0 value (if the M0 readout is above the positivity cut-off) (positivity cut-off = 3970 AU/ml for N Index and 48.6 AU/ml for Spike BA.5); positive sero-response for nAb ID50 BA.4/5 is defined as participants with value above the LOD (if the M0 readout is below the LOD) or is at least 4-fold above the M0 value (if the M0 readout is above the LOD).

[d]For these rows cases restrict to those who acquired a COVID-19 endpoint 7 days post Peak through 92 days post Peak.

that this observation also holds in PWH. In the Hybrid Group, pre-vaccination nAb-ID50 BA.4/5, Spike IgG Index, and Spike IgG BA.4/5 levels all inversely correlated with COVID-19 through the full follow-up period, 230 days post Peak. In contrast, in the Vaccine Group, neither Spike IgG BA.4/5 or nAb-ID50 BA.4/5 levels at Peak correlated with COVID-19 when including endpoints through 230 days post Peak. However, when restricting to COVID-19 endpoints through 92 days post Peak, both markers were inverse CoRs. The finding that Peak nAb-ID50 BA.4/5 titer inversely correlated with risk of COVID-19 in the Vaccine Group through early vaccination-proximal follow-up, coupled with the statistical interaction test result finding no evidence that HIV status modifies this inverse correlation, together support that among PWH, those with high post vaccination BA.4/5 neutralization titers—especially those with > 1000 AU/ml—develop significant short-term protection after receiving 2 doses of ancestral strain vaccine, but that this protection wanes precipitously over time.

In interpreting the different result in the Vaccine Group for vaccination-proximal (by 92 days) vs. vaccination-distal (93 to 230 days) COVID-19, note that 79% of Vaccine Group cases were vaccination-distal, which is relevant because the relationship between risk and marker level may vary over time. Moreover, all COVID-19 endpoints by 92 days were symptom-triggered given the lack of nasal swabbing for virus detection, such that endpoint occurrence was spread across 7 to 92 days post Peak. In contrast, most of the vaccination-distal endpoints were detected by a positive nasal swab at the month 6 visit, indicating a late event ~150–230 days post Peak. Therefore, antibody levels had waned to lower levels at the times of SARS-CoV-2 exposure events for the vaccination-distal period compared to the vaccination-proximal period, where essentially all Vaccine Group participants had antibody levels that were too low ~150–230 days post Peak for an association with reduced COVID-19 risk to be observed. This may explain CoVPN 3008's finding that the Peak inverse CoRs in the Vaccine Group only attained for COVID-19

endpoints through 4 months post vaccination. Two other possible contributing explanations are the continued evolution of the Omicron epidemic from BA.4/5 to Omicron-descendent isolates[44,45] and that other protective components of the immune response not captured by neutralizing antibody titers may have waned by 4 months post vaccination. One implication of this waning utility of Peak titer as a correlate is that titers measured after Peak may better inform decisions around booster timing.

In the Hybrid Group, the markers measured at enrolment/M0 were clearly correlates whereas as the evidence was less clear for the markers measured at Peak. From the Cox regression models, none of the Peak marker HRs had a P-value below the chosen threshold for defining statistical significance, 0.05. However, Peak nAb-ID50 BA.4/5 trended towards an inverse correlate, with this conclusion based on the Cox model result (HR 0.72; 95% CI: 0.52, 1.01; P = 0.059) combined with the marker-thresholded correlates analysis that showed a precipitous drop in COVID-19 risk through 160 days post Peak at high Peak nAb-ID50 BA.4/5 titer. The apparent superiority of the nAb-ID50 titer correlate when measured pre- vs. post-vaccination suggests that BA.4/5 neutralization titer captures immunity differently at the two time points. High pre-vaccination titer may be associated with unmeasured protective immune functions induced by infection (e.g., in the B and T cell repertoires), whereas post-vaccination titer skews the post-vaccination immunity to Spike neutralizing antibodies and appears to have a diminished connection to the underlying protective functions.

Antibody markers measuring fold-rise from M0 to Peak were also assessed as a correlate of COVID-19, where we found that both the fold-rise in IgG Spike concentration and in nAb-ID50 titer were direct correlates of COVID-19. Because the pre-vaccination/M0 values—but not the Peak values—were direct correlates of COVID-19, we would argue that the marker at M0 may be more practical for applications than the markers defined as fold-rise, given that fewer measurements would be

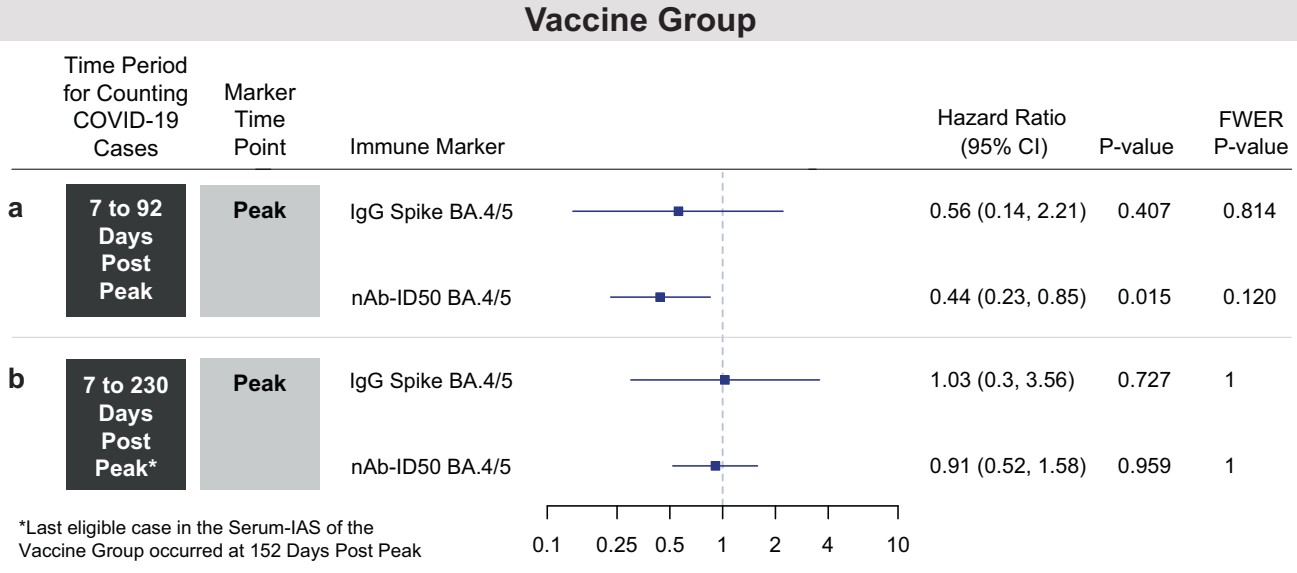

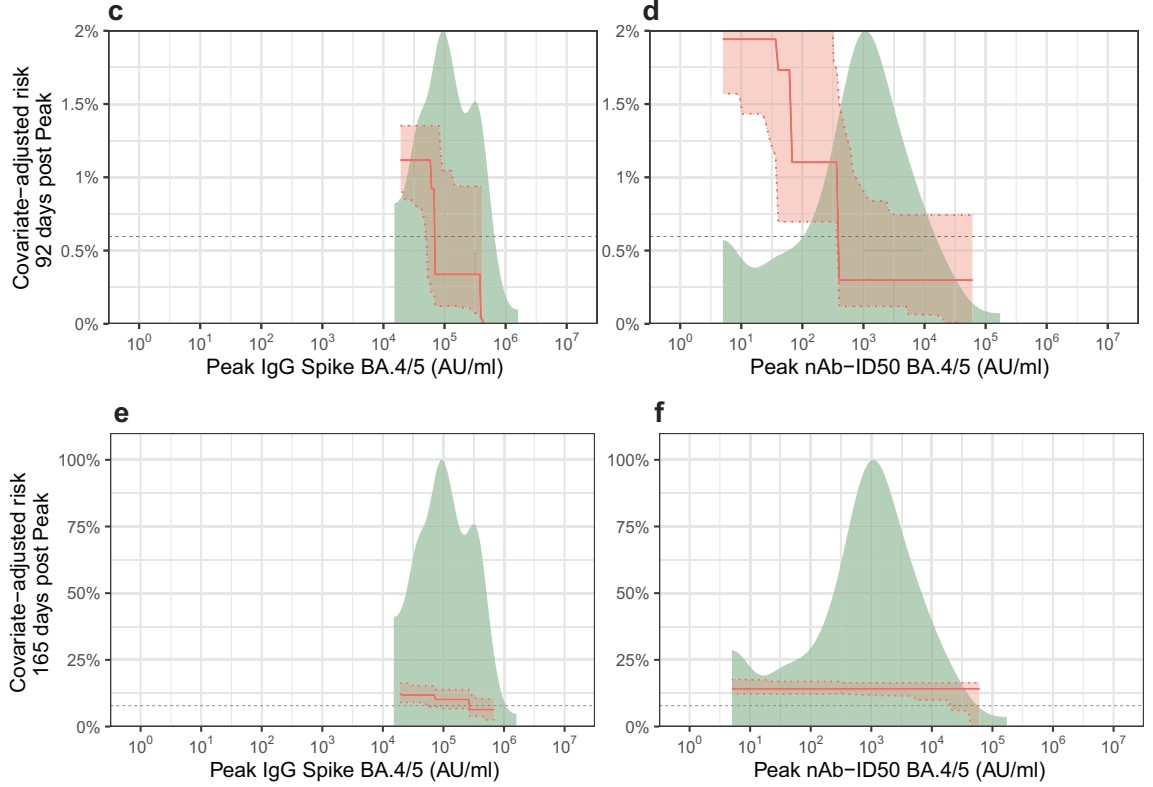

**Fig. 4 | Correlate of risk analyses in the Vaccine Group for antibodies measured at Peak.** Analyses used eligible COVID-19 cases and non-cases in the Per-protocol Serum Immunogenicity Analysis Set in the Vaccine Group (N = 115). **a, b** Covariate-adjusted hazard ratios (HRs) of COVID-19 [follow-up: **a** 7 to 92 days post-Peak; **b** 7 to 230 days-post Peak] per 10-fold Peak immune marker increase. Blue squares are point estimates and horizontal lines are 95% confidence intervals. *P*-values are from a two-sided Wald test. Holm-Bonferroni family-wise error rate (FWER) adjusted two-sided *p*-values are also shown. Cases acquired a COVID-19 endpoint **a** 7 through 92 days post Peak or **b** 7 through 230 days post Peak. HRs were estimated using inverse probability sampling weighted Cox regression models; 95% confidence intervals (CIs) and two-sided Wald-based p-values are shown. Analyses adjusted for: **a** HIV status and baseline risk score; **b** whether enrolled in South Africa, HIV status, TB status, enrolment period, and baseline risk score. **c–f** Controlled risk plots for Vaccine Group (N = 115) antibody markers at Peak. Covariate-adjusted probability of

COVID-19 by **c**, **d** 92 days post Peak or **e**, **f** 165 days post-Peak is estimated under hypothetical assignments of all participants to the Vaccine Group and the marker value on the x-axis. **c, e** Peak IgG Spike BA.4/5 concentration; **d, f** Peak nAb ID50 BA.4/5 titer. Controlled risk was estimated using a monotone-constrained non-parametric method with covariate adjustment and restricted to the middle 95% quantiles of the available marker data. Solid lines indicate point estimates; dotted lines and shading indicate pointwise 95% CIs. Horizontal gray lines: overall cumulative incidence of COVID-19 from 7 to 92 (or 165) days post-Peak in the Vaccine Group. Background kernel density plots show marker distributions at M0 or at Peak. Analyses adjusted for whether enrolled in South Africa, HIV status, TB status, enrolment period, and baseline risk score. AU, arbitrary units; FWER, family-wise error rate adjusted p-value; N, Nucleocapsid protein; nAb-ID50, 50% inhibitory serum dilution neutralizing antibody titer; Peak, 4 weeks post-last vaccine dose (M2 for Vaccine Group).

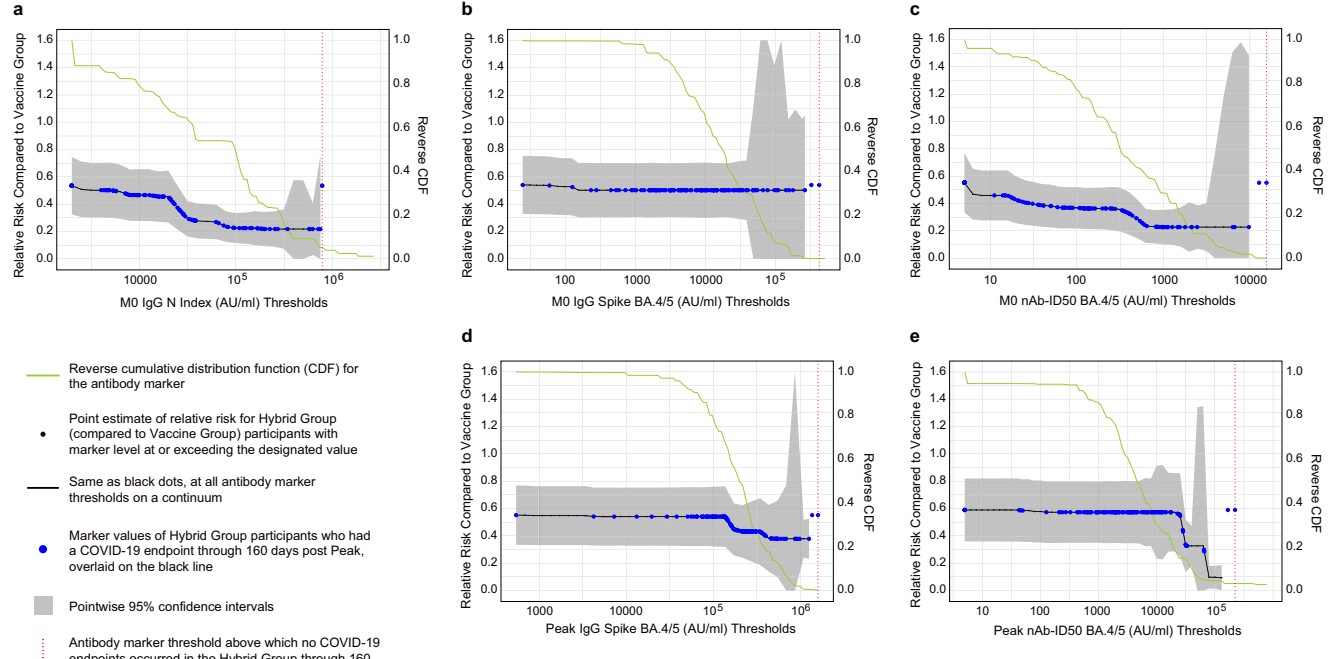

**Fig. 5 | Stochastic-thresholded relative risk CoP analysis.** Plots show relative risk of COVID-19 through 160 days post-Peak for the Hybrid Group with thresholded antibody marker **a** M0 IgG N Index concentration, **b** M0 IgG Spike BA.5 concentration, **c** M0 nAb ID50 BA.4/5 titer, **d** Peak IgG Spike BA.4/5 concentration, **e** Peak nAb ID50 BA.4/5 titer vs. overall risk of the Vaccine Group. Relative risk is the covariate-adjusted thresholded cumulative incidence in Hybrid Group participants divided by the covariate-adjusted cumulative incidence in the Vaccine Group. Each black dot (threshold value) represents a point estimate of relative risk (compared to the entire Vaccine Group) of COVID-19 through 160 days post-Peak for Hybrid Group participants if their marker levels were as high as or higher than that given threshold value. The grid of thresholds was created by segmenting the marker values at COVID-19 endpoints into increments of 0.1. This grid spans from the minimum marker value to the highest value for which there are at least 3 COVID-19 endpoints with a marker value at or above that value. The solid black lines linearly interpolate the grid points. The blue dots represent the marker values of Hybrid Group participants who had a COVID-19 endpoint through 160 days post-Peak, overlaid on the black line. The vertical red dashed line is the antibody marker threshold above which no COVID-19 endpoints occurred in the Hybrid Group through 160 days post-Peak. The gray shaded area indicates pointwise 95% CIs. The estimates and CIs were adjusted using the assumption that the true threshold-response relative risk is non-decreasing. The upper boundary of the green shaded area is the estimate of the reverse cumulative distribution function (CDF) of the marker in Hybrid Group participants. Analyses adjusted for whether enrolled in South Africa, HIV status, and baseline risk score. Data points are from eligible COVID-19 cases and non-cases in the Per-protocol Serum Immunogenicity Analysis Set. nAb-ID50, 50% inhibitory serum dilution neutralizing antibody titer. Peak, 4 weeks post last vaccine dose (M1 for Hybrid Group). CI, confidence interval; CoP, correlate of protection.

required. Moreover, the interpretation of the fold-rise depends on the M0 level, making its use more complicated compared to the absolute M0 level.

Pre-vaccination antibody markers capture only B-cell responses generated by prior SARS-CoV-2 infection, mostly with Omicron viruses in the present study (enrolment December 2, 2021 to September 9, 2022), which may be correlated with the salient memory/recall B-cell responses[46,47] that appear to be critical for protection against future exposures to Omicron viruses[48,49]. In contrast, antibodies elicited by vaccination to the Ancestral vaccine strain have limited cross-reactivity against Omicron viruses (~ 6-fold reduced neutralization titer, Supplementary Fig. 11 in the present work) and are less connected to the salient memory/recall responses. Hence, higher levels of these antibodies may be needed for an inverse association with risk to be observed. Further studies to evaluate antibody form, specificity, and function (e.g., subclass, isotype, epitope mapping, Fc function, etc.) may improve precision for detecting vaccine-elicited antibodies that correspond with protection, if they are present.

For the Hybrid Group, our correlates results align with literature consistently supporting the pseudovirus neutralization assay's strong performance as an immune correlate[11,22-26,30,35,38-41]. In the Moderna COVE trial, IgG Spike binding levels and neutralizing antibodies against BA.1 measured 4 weeks after a third mRNA-1273 dose were inverse CoRs of BA.1 COVID-19 and also supported as correlates of protection (vs. 2-dose recipients) in non-naïve individuals in the United States

followed from December 2021 to April 2022[39]. Similarly, in the COVAIL study in the United States with follow-up from March 2022 to May 2023, nAb-ID50 BA.1 titer was a strong inverse CoR of Omicron COVID-19 in non-naïve recipients of a second mRNA booster for each of several insert sequences considered, including the mRNA-1273 Ancestral strain[42]. A prospective cohort study in the United Kingdom studied IgG N and Spike levels as correlates of SARS-CoV-2 re-infection in 9–13 year-olds during 2022, finding that anti-N levels were a stronger correlate of reinfection than anti-Spike levels[50], which we also found in CoVPN 3008 by the thresholding correlates analyses (Fig. 5, Supplementary Fig. 13).

For the Vaccine Group, the finding that Peak markers were inverse CoRs of vaccination-proximal (by 92 days) but not vaccination-distal (93 to 230 days) COVID-19 is consistent with the results in COVE and COVAIL that Peak nAb-ID50 markers in both studies were inverse CoRs in naïve participants. The former study had follow-up over about 4 months (consistent with CoVPN 3008) and the latter had follow-up over 6 months, with results showing evidence that the correlate was stronger over the first 3 months. In addition, consistent with CoVPN 3008 results, COVAIL supported that the neutralization correlate was stronger for non-naïve individuals, with leading explanation that more previously infected participants had antibody levels in a high enough range to be relevant for marking protection. A significant factor in comparing results is whether circulating strains during follow-up had different genetic or antigenic distances to the mRNA-1273 vaccine

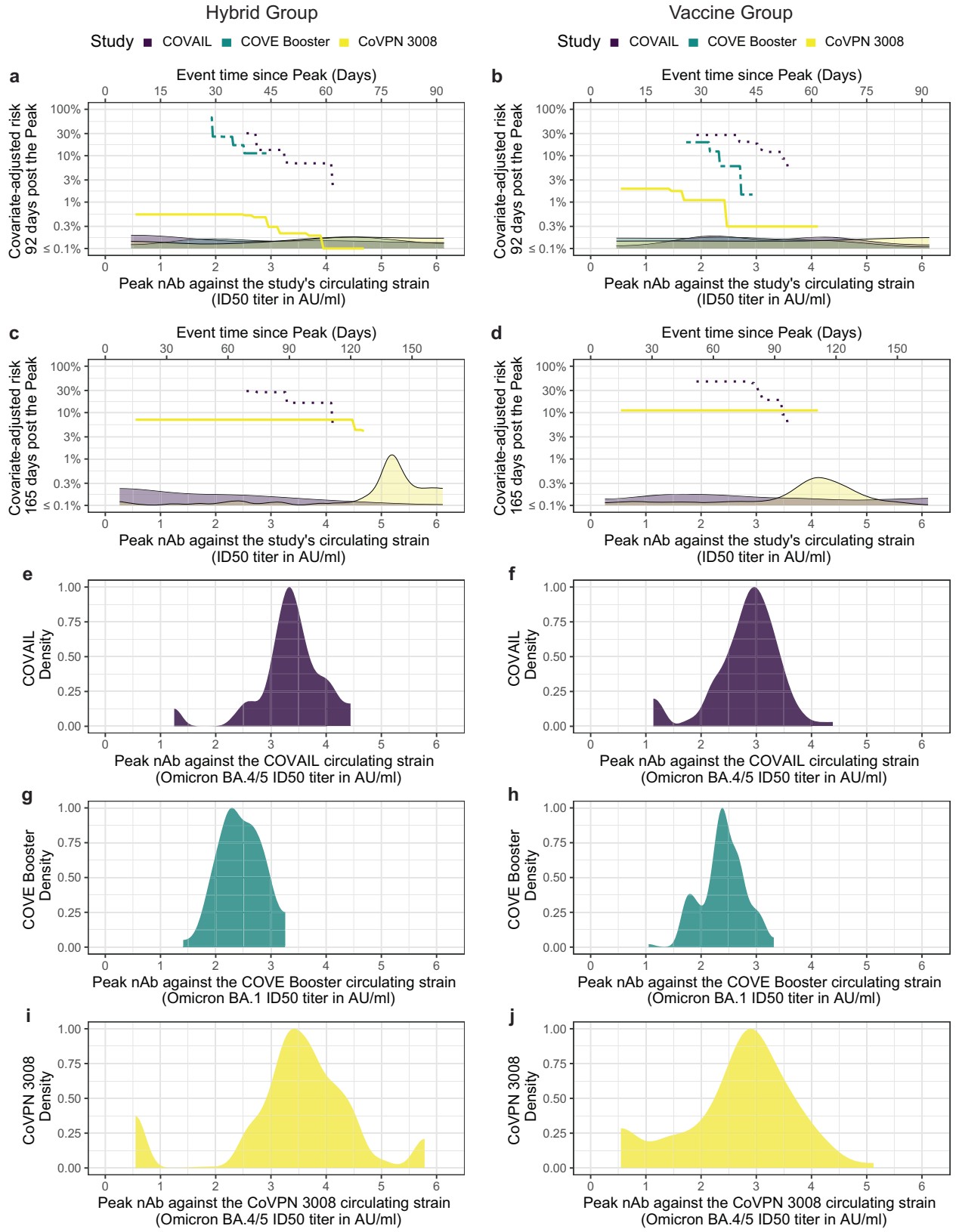

strain—our bioinformatics analysis supported fairly similar distributions of distances across the three trials. While it is challenging to fully compare results between the trials, given the different numbers of vaccine doses and the different overall COVID-19 rates, our interpretation is no evidence that the IgG Spike or neutralization titer correlates operate differently in the East and Southern Africa PWH population compared to the United States PWoH population.

A strength of this analysis is that the relatively large numbers of evaluable breakthrough COVID-19 endpoints (152 in people with prior natural immunity from COVID-19 acquisition, i.e., Hybrid Group, and

**Fig. 6 | Comparison of controlled risk curves between the CoVPN 3008 study (recipients of one or two doses of mRNA-1273, depending on history of prior infection), the COVAIL study (data shown here are restricted to recipients of a second one-dose mRNA-1273 boost**[42]**), and the COVE study (recipients of a third mRNA-1273 dose**[39]**), separately in the Hybrid and Vaccine Groups. a–d** Controlled risk of COVID-19 through **a**, **b** 92 days post-Peak or **c**, **d** 165 days post-Peak for **a**, **c** Hybrid Group and **b**, **d** Vaccine Group participants, by Peak neutralizing antibody ID50 titer (in AU/ml) against the dominantly circulating variant (CoVPN 3008: BA.4/5, COVAIL BA.4/5, COVE Booster: BA.1). For each trial, Peak is: CoVPN 3008 – 4 weeks post-last dose (M1 for Hybrid Group, M2 for Vaccine Group); COVAIL – 2 weeks post-second boost; COVE – 28 days post-third dose. Hybrid vs. Vaccine is defined by SARS-CoV-2 positive vs. negative at first dose (CoVPN 3008), second boost post-primary series (COVAIL), or third dose (COVE). For COVE, all participants received three mRNA-1273 doses; for COVAIL, all participants received mRNA-1273 as their second booster dose. Controlled risk was estimated using a monotone-constrained nonparametric method with covariate adjustment and restricted to the middle 90% quantiles of the available marker data. Background kernel density plots are estimates of the distribution of the time-to-COVID-19 in days since Peak (**a**, **b** 7 to 92 days; **c**, **d** 7 to 165 days). **e–j** Estimates (kernel density plots) of Peak neutralizing antibody distribution against the circulating strain in the respective analysis population. Analyses adjusted for: COVAIL, force of infection score and baseline risk score; COVE, baseline risk score, at risk status, and community of color status; CoVPN 3008, whether enrolled in South Africa, HIV status, TB status, enrolment period, and baseline risk score. Study endpoint lineages and timeframes were: COVE, BA.1, December 1, 2021 to April 5, 2022; COVAIL, several Omicron sub-lineages, May 30, 2022 to November 28, 2022; CoVPN 3008, several Omicron sub-lineages, April 26, 2022 to February 7, 2023 (pooling over Hybrid and Vaccine groups). AU/ml, arbitrary units/ml; nAb, neutralizing antibody; ID50, 50% inhibitory serum dilution titer.

54 in people without prior natural immunity, i.e., Vaccine Group) provided ample precision for assessing antibody marker immunogenicity and immune correlates in both groups. A limitation of this analysis is the lack of an unvaccinated/placebo arm in the CoVPN 3008 study, as such an arm would have enabled the direct study of the effect of mRNA-1273 vaccination on COVID-19 and of immune correlates of protection that causally link antibody markers to vaccine efficacy. Nevertheless, our immune correlates of risk findings support that mRNA-1273 vaccine protection was likely modest and short-term, given that if efficacy were high then the post-vaccination nAb-ID50 titers would be expected to inversely correlate with COVID-19 in the Vaccine Group for a longer duration of follow-up. Peak nAb-ID50 titers were higher in the Hybrid vs. Vaccine Group, such that for Hybrid the range of nAb-ID50 titers across participants remained high enough to provide protection for 8 months. In contrast, for Vaccine, almost all participants had nAb-ID50 titers too low to be able to provide protection for more than 4 months. The one randomized phase 3 trial of an Ancestral-strain vaccine vs. placebo against Omicron circulating strains supported very low or absent efficacy in a vaccine group naïve to SARS-CoV-2[20]. Another limitation is the inability to fully account for immunity-relevant factors that are heterogeneous across Hybrid Group participants such as the lineage and timing of the prior SARS-CoV-2 infection. However, the M0 IgG Spike and nAb-ID50 correlates were the same with and without adjustment for M0 IgG N Index levels that are a proxy for recency of previous infection. In addition, the analysis studied anti-BA.4/5 immune correlates of overall COVID-19 without accounting for the lineage of breakthrough viruses. We also note that the study adopted a case-cohort design when sampling the immune markers. While the cohort was randomly selected, exclusion criteria were used to arrive at the final per-protocol serum correlates cohort. Estimates could be biased if the final per-protocol serum correlates cohort, after applying the exclusion criteria, did not remain a stratified random sample of the entire per-protocol cohort. Finally, the present analysis assessed IgG binding antibody concentrations without considering fine specificity and/or avidity and isotype/subclasses, where future analyses could assess whether the correlates would be more accurate predictors if considering such characteristics.

## Methods

We have complied with all relevant ethical regulations. The trial was registered on ClinicalTrials.gov (NCT05168813) and conducted according to International Council for Harmonisation of Technical Requirements for Pharmaceuticals for Human Use, Good Clinical Practice guidelines. Written informed consent was obtained from all participants prior to enrolment. Each site followed its standard, approved procedures to determine reimbursement amounts for participants to cover travel, time, and inconvenience costs associated with study participation.

The CoVPN 3008 study protocol was approved by the following Research Ethics Committees (RECs) [listed as Clinical Research Site (CRS)/Institution Name − REC(s); Regulatory Body(ies)]: Gaborone CRS − Harvard Institutional Review Board (IRB) and Health Research and Development Committee; Botswana Medicines Regulatory Authority. Eswatini Prevention Center CRS − Columbia Human Research Protection Office IRB, Eswatini Health and Human Research Review Board; Medicine Regulatory Unit. Moi University Clinical Research Center (CRC) − Moi University CRC: Institutional Research and Ethics Committee (IREC); Ministry of Health Pharmacy and Poisons Board and National Commission for Science, Technology and Innovation (NACOSTI). Kisumu CRS − Kenya Medical Research Institute (KEMRI) Scientific Ethics Review Unit; NACOSTI, Jaramogi Oginga Odinga Teaching & Referral Hospital (JOOTRH), County Administry of Health. Kombewa CRC − KEMRI Scientific Ethical Review Unit & Walter Reed Army Institute of Research; NACOSTI, County Administry of Health. Blantyre CRS − College of Medicine Research Ethics Committee, University of Malawi College of Medicine, Johns Hopkins School of Public Health IRB. Malawi CRS − University of North Carolina (UNC) Chapel Hill Office of Human Research Ethics, National Health Science Research Committee. UVRI-IAVI HIV Vaccine Program LTD. CRS − UVRI (Uganda Virus Research Institute Research and Ethics Committee) Ethics; Uganda National Council for Science and Technology (UNCST). Baylor-Uganda CRS − UVRI REC/IRC; UNCST. Joint Clinical Research Center − UVRI REC; UNCST. MU-JHU Research Collaboration CRS − UVRI Ethics; UNCST. CFHRZ CRS − University of Zambia Biomedical Research Ethics Committee (UNZABREC); Zambia Medicines Regulatory Authority; National Health Research Authority. Matero Reference Clinic CRS / Center for Infectious Disease Research in Zambia (CIDRZ) − UNZABREC; Zambia Medicines Regulatory Authority; National Health Research Authority. UNC Global Projects / Kamwala District Health Center − UNZABREC; Zambia Medicines Regulatory Authority; National Health Research Authority. Zambia Emory HIV Research Project – Ndola CRS − UNZABREC; Zambia Medicines Regulatory Authority; National Health Research Authority. Zambia Emory HIV Research Project – Ndola CRS − UNZABREC; Zambia Medicines Regulatory Authority; National Health Research Authority. PHOENIX Pharma (Pty) Ltd − Pharma Ethics; South African Health Products Regulatory Authority (SAHPRA). Groote SchuurHIV CRS − University of Cape Town Human Research Ethics Committee (UCT HREC); SAHPRA. Task Central − Pharma Ethics; SAHPRA. Soweto - Kliptown CRS − Wits HREC; SAHPRA. FAM-CRU (Family Clinical Research Unit) − University of Stellenbosch Ethics Committee; SAHPRA. Josha Research CRS − Wits HREC; SAHPRA. Tembisa Clinic 4 − Wits HREC; SAHPRA. Qhakaza Mbokodo Research Clinic (QMRC) − Wits HREC; SAHPRA. Aurum Institute Klerksdorp CRS − Wits HREC; SAHPRA. Rustenburg CRS − Wits HREC; SAHPRA. Emavundleni CRS − UCT HREC; SAHPRA. Clinical HIV Research Unit (CHRU)/ Helen Joseph CRS − Wits HREC; SAHPRA. CAPRISA eThekwini

Clinic − University of KwaZulu-Natal Biomedical Research Ethics Committee (UKZN BREC); SAHPRA. Masiphumelele Clincial Research Site (Masi) CRS − UCT HREC; SAHPRA. Soweto - Bara CRS − Wits HREC; SAHPRA. Tongaat CRS − South African Medical Research Council (SAMRC); SAHPRA. CAPRISA Vulindlela CRS − UKZN BREC; SAHPRA. Nelson Mandela Academic Clinical Research Unit CRS − Wits HREC; SAHPRA. MeCRU CRS − Sefako Makgatho University Research Ethics Committee (SMUREC); SAHPRA. University of Cape Town Lung CRS Institute − UCT HREC; SAHPRA. PHRU Matlosana CRS − Wits HREC; SAHPRA. Synergy Biomed Research Institute − Pharma Ethics; SAHPRA. Newtown Clinical Research − Pharma Ethics; SAHPRA. TASK Eden − Pharma Ethics; SAHPRA. Wits RHI Ward 21 CRS − Wits HREC; SAHPRA. Isipingo CRS − SAMRC; SAHPRA.

### Participants and procedures

CoVPN 3008 enrolled adults with HIV and/or with other comorbidities associated with severe COVID-19 based on CDC criteria[51]; see Garrett et al.[1] for details. At baseline, all participants had an HIV antibody test, a point-of-care[52] SARS-CoV-2 anti-Spike serology test (Assure Ecotest, Assure Tech, Hangzhou, China), a central laboratory anti-nucleoprotein SARS-CoV-2 serology test (anti-N) (Abbott SARS-CoV-2 IgG, Abbott, Chicago, IL, USA), and a nasal swab SARS-CoV-2 nucleic acid amplification test (NAAT). All participants received mRNA-1273 vaccination (monovalent mRNA vaccine encoding the Spike protein of SARS-CoV-2 strain WA1) at enrolment (M0). The baseline point-of-care anti-Spike serology test result was used to determine whether participants were given a second mRNA-1273 dose at Month 1, where only participants with a negative result were given a second mRNA-1273 dose at Month 1. The populations of interest for immune correlates assessment are per-protocol (received all planned vaccinations with no major protocol violations) and qualify for either the Hybrid Group (baseline positive by any of the three tests anti-N, anti-Spike, and NAAT) or the Vaccine Group (baseline negative by all three tests). Garrett et al.[1] describes surveillance for occurrence of COVID-19, which includes nasal swabs for SARS-CoV-2 NAAT before each vaccination and at month 6, self-monitoring for prespecified COVID-19 symptoms at home, and site personnel contacting participants every two weeks to assess for symptoms. COVID-19 symptoms triggered a clinic visit for further assessment and nasal swab testing.

### COVID-19 endpoint definition

The same COVID-19 endpoint evaluated in Garrett et al.[1] was used, based on CDC symptom criteria, except COVID-19 endpoints began to be counted only at 7 days after the Peak visit up until 230 days post Peak (or up until 92 days post Peak, as noted), including endpoints up to March 31, 2023.

### Reporting on sex and gender

Information on sex was determined based on self-report. Participants could choose between the following options: Male, Female. Sex was used as one of the baseline input variables considered in building the baseline risk score [see the Statistical Analysis Plan (SAP)], and sex was adjusted for in analyses comparing antibody levels between the Hybrid and Vaccine Groups and between the PWH and PWoH groups. Disaggregated sex data are not provided in the source data because consent has not been obtained for sharing of individual-level data.

The present study did not prespecify any sex-based analyses as immunogenicity of the mRNA-1273 vaccine did not vary with sex in our previous analyses of this vaccine[53] and there has not been evidence that sex modifies correlates of risk of COVID-19 for this vaccine[39].

### Meso scale discovery-electrochemiluminescence assay (MSD-ECL) for defining Spike and N IgG binding antibody markers

SARS-CoV-2 specific IgG antibody responses were evaluated using the Mesoscale Discovery validated COVID-19 V-Plex serology kit. The V-Plex SARS-CoV-2 variant spike panel 1 kit quantitatively measures antibodies to antigens related to SARS-CoV-2, including variants of the virus. 96-well plates are provided with wells containing spots corresponding to specific antigens. Following the validated kit protocol, antibodies in the sample bind to antigens on the spots and are detected using anti-human IgG antibodies conjugated with MSD SULFO-TAG (SULFO-TAG Anti-Human IgG Antibody, MSD, Catalog Number: D21ADF-3) diluted (1X) in MSD Diluent 100 solution. Each serum sample was tested over three dilutions (1:500, 1:10,000, and 1:200,000) in duplicate, including 1 dilution factor per assay plate. Each assay plate included 36 serum samples, incorporating both serology controls and a reference standard.

The plate is read on a MESO SECTOR S 600 instrument, measuring binding magnitude as electro chemiluminescence (ECL) signal. Methodical Mind Instrument Software was used for raw data acquisition; data were then imported to and further analyzed using MSD Discover Workbench 4.0 Analysis Software. Antibody concentration is measured in Arbitrary Units per milliliter (AU/mL), which is calculated by backfitting to a 7-point calibrator curve. Results are reported in AU/mL. Readouts of the Index D614 strain (Spike) and Index D614 strain (N) in AU/ml units may be multiplied by 0.009 to convert them to Binding Antibody Units (BAU/ml), for comparison with previous correlates analyses[22,26]. Readouts to the other, non-Index strains cannot be converted to BAU/ml because the anti-SARS-CoV-2 immunoglobulin international standard has not been assayed against these strains for conversion.

The assay included the vaccine strain for Nucleocapsid and Spike (Index D614) and eight Spike variants: Alpha (B.1.1.7), Beta (B.1.351), Delta (B.1.617.2; AY.4), Omicron BA.1 (BA.1.1.529), Omicron BA.2, Omicron BA.2.12.1, Omicron BA.2.75, Omicron BA.5. The assay's qualification defined lower and upper limits of quantitation for each antigen, as well as positivity cut-offs (Supplementary Table 5A). Given the focus on Omicron COVID-19, the selected anti-Spike marker for correlates analyses is IgG Spike BA.5. With the majority of participants in the Hybrid Group, the IgG N Index marker is also selected. IgG values below the antigen-specific positivity cut-off were assigned value positivity cut-off/2. Values below the antigen-specific positivity cut-off at M0 were considered negative; otherwise positive. For each antigen, an IgG sero-response at Peak was defined as IgG above the antigen-specific positivity cut-off (when M0 IgG ≤ antigen-specific positivity cut-off) or at least 4-fold above the M0 value (when M0 IgG > antigen-specific positivity cut-off).

### Pseudovirus neutralization assay for defining the nAb-ID50 BA.4/5 marker

Neutralizing antibody responses against SARS-CoV-2 were tested using a vesicular stomatitis virus (VSV)-based neutralization assay. In the VSV assay, heat-inactivated serum samples underwent five-fold serial dilutions in 96-well plates (Corning Life Sciences) before addition of SARS-CoV-2 pseudovirus. Each serum sample was tested over eight dilutions (plus the initial dilution, 1:10) in duplicate on each assay plate; one assay plate was run for five serum samples. The serum-virus complexes were mixed with Vero E6 cells (American Type Culture Collection, Catalog Number: CRL-1586) and incubated for 20−24 h. nAb responses were measured as a function of a reduction in luciferase reporter gene expression after a single round of infection with molecularly cloned SARS-CoV-2 pseudovirus. SARS-CoV-2 BA.4/5 pseudotyped virus was purchased from and titered by Nexelis Laboratories Canada, Inc. and used at a dilution of 1:27. Data were acquired using Perkin Elmer 2030 workstation version 4 and analyzed on SCHARP's Atlas portal (https://atlas.scharp.org/project/home/begin.view). Further details are in refs. 54,55. Supplementary Table 5B provides the nAb-ID50 assay limits. The limit of detection (LOD) was 10 AU/ml, and values below the LOD were assigned LOD/2. Values greater than the antigen-specific upper limit of

quantitation (ULOQ) were assigned the ULOQ. A nAb-ID50 positive sero-response at Peak was defined as ID50 > LOD (when M0 ID50 ≤ LOD) or at least 4-fold above the M0 value (when M0 ID50 > LOD).

## Statistical methods
Data analyses were performed as specified in the SAP (Supplementary Material).

**Comparison of antibody markers between key demographic groups.** Linear regression is used to provide covariate-adjusted point and 95% confidence interval estimation of the mean log-scale antibody marker levels at M0 and Peak for the four subgroups defined by (Hybrid, Vaccine) × (PWH, PWoH). Two-sided bootstrap p-values are reported for the test of whether marginalized mean $\log_{10}$-scale M0 levels significantly differ by Hybrid vs. Vaccine (combining across PWH and PWoH) and by Hybrid PWH vs. Hybrid PWoH, and for whether marginalized mean Peak levels significantly differ by Hybrid vs. Vaccine, by Hybrid PWH vs. Hybrid PWoH, and by Vaccine PWH vs. Vaccine PWoH. The marginalized mean outcome was calculated using g-computation. The bootstrap p-value was obtained by comparing the observed difference in the marginalized mean outcome in each group to its bootstrapped distribution.

**Covariate adjustment.** Analyses comparing antibody levels between the Hybrid and Vaccine Groups and between the PWH and PWoH groups (within each of the Hybrid and Vaccine groups) adjusted for whether age exceeded 40 and the four categories (male, female) × (BMI < = 25, BMI > 25); where the Hybrid vs. Vaccine group analyses also adjusted for PWH status. All correlates analyses pooling over PWH and PWoH adjusted for whether enrolled in South Africa (versus any other African country), PWH status, TB status, enrolment period (< 3 months, 3-6 months, > 6 months post first person enrolled), and baseline risk score.

**Correlates of risk assessment.** All antibody markers at M0 were studied as correlates in the Hybrid Group but not in the Vaccine Group, given the low variability of marker values across Vaccine Group participants that are all SARS-CoV-2 negative. All anti-Spike markers at Peak were studied as correlates in both the Hybrid and Vaccine Groups; anti-N markers at Peak were excluded because the mRNA-1273 vaccine did not contain the N protein. In addition, for the Hybrid Group, fold-rise of the anti-Spike markers from M0 to Peak were studied as an immune correlate. As all markers are analyzed on the log10 scale, "fold-rise" is the additive difference in Peak value minus M0 value.

For each antibody marker assessed at M0 or Peak, the baseline factor-adjusted hazard ratio of COVID-19 was estimated using inverse probability sampling weighted Cox regression models with 95% CIs and Wald-based *p*-values, fit using the R survey package[56]. Nonparametric monotone-constrained methodology was applied to flexibly estimate baseline-factor adjusted marker-conditional cumulative incidence of COVID-19 through 92 days or 165 days post Peak[57]. Analyses were done using the default implementation in the R package *vaccine* available at CRAN[57]. Point and 95% CI estimates about baseline-factor adjusted marker-thresholded conditional cumulative incidence through 92 days or through 160 days post Peak were computed by nonparametric targeted minimum loss-based regression, ranging over a grid of all possible thresholds[58]. The landmark times 160 and 165 days were chosen as the latest times supporting stable estimation.

Bivariate Cox models were also fit for five pairs of antibody marker values at M0 and/or Peak, to assess which assay and which time point provided the strongest and independent association with COVID-19. Partial likelihood ratio tests were used to systematically compare the goodness-of-fit of Cox models with baseline covariates and M0 nAb-

ID50 BA.4/5 titer plus each possible augmentation with 1 or 2 markers at M0 and/or Peak.

**Hypothesis testing and multiple testing adjustment.** For each group (Hybrid, Vaccine) and each eligible M0 and Peak antibody marker, a Wald hypothesis test from the Cox model was conducted for whether the marker correlates with risk. For the bivariate marker models, a generalized Wald test was used for any association of the two markers with risk.

All p-values are two-sided. For the nine Cox-model based antibody marker correlates of risk analyses reported in Fig. 3, Holm-Bonferroni family-wise error rate (FWER) adjusted p-values are calculated; see SAP Section 11.5 for methods details. In addition, Holm-Bonferroni FWER-adjusted are reported for the seven interaction tests reported in Supplementary Table 3 for whether PWH status modifies the correlate of risk.

## Correlates of protection assessment
**Thresholded stochastic risk correlates of protection.** Thresholded relative risk through 92 or 160 days post Peak was estimated as one minus baseline-factor adjusted marker-thresholded conditional cumulative incidence of Hybrid Group participants (described above) divided by baseline-factor adjusted cumulative incidence of the Vaccine Group (causal interpretation in van der Laan, Zhang, and Gilbert[58]), ranging over the same thresholds studied in the CoR analysis.

**Controlled risk correlates of protection.** The baseline-factor adjusted marker-conditional marginalized cumulative incidence parameters noted above equate to controlled-risk correlates of protection parameters under causal assumptions[59]. We focus on reporting results from the nonparametric monotone-constrained methodology, given the flexible estimation that can potentially detect thresholds of low risk[57]. Point estimates and 95% CIs of controlled-risk curves are shown for each group (Hybrid, Vaccine) and each eligible M0 and Peak antibody marker.

**Pseudovirus neutralization titer conversion from (NICD) AU/ml to Duke ID50/ml.** A concordance study for the VSV-based neutralization assay performed at the National Institute for Communicable Diseases (NICD) compared to the lentivirus-based ACE-2 neutralization assay performed at the Duke lab[60] evaluated $n = 76$ samples from $n = 38$ participants from CoVPN 3008. The results showed high concordance between the assays against both the D614G and BA.4/5 variants, in PWoH for which both assays are valid. A comprehensive summary of the concordance results can be found in the Supplementary Material. Based on the concordance study, we derived the following relationships, each based on a simple linear regression.

Between NICD AU/ml and Duke ID50 titers (for BA.4/5):

$$1.008 \times \log_{10}(NICD\,AU/ml) - 0.157 = \log_{10}(Duke\,ID50)$$

Between NICD AU/ml and Duke WHO International Units of IU50/ml (for D614G/Reference):

$$0.833 \times \log_{10}(NICD\,AU/ml) + 0.017 = \log_{10}(Duke\,IU50/ml)$$

The adjusted R-squared equals 0.91 in both cases, suggesting good fitting.

**Pseudovirus neutralization titer conversion from Monogram ID50 titers to Duke ID50 titers.** A concordance study of pseudovirus neutralization ID50 antibody titers measured by the Monogram vs. Duke assay evaluated 169 participants from the COVID-19 Variant Immunologic Landscape (COVAIL) Trial. The study evaluated the concordance

for variants BA.1, BA.2.12.1, BA.4/5, Beta, D614G, and Delta at two timepoints: baseline (D1) and peak (D15). For the predominant circulating variant BA.4/5 we derived the following relationship between Monogram ID50 titers and Duke ID50 titers based on simple linear regression:

$$0.954 \times \log_{10}(\mathrm{MonogramID50titer}) - 0.105 = \log_{10}(\mathrm{DukeID50titer}).$$

The adjusted R-squared equals 0.84, suggesting good fitting.

### Retrieval of SARS-CoV-2 sequences for analysis
For comparison of the vaccine strain to the circulating SARS-CoV-2 strains in this and in previous studies, SARS-CoV-2 Spike genomic sequences representing the estimated circulating strains were retrieved from GISAID[61].

**Software and data quality assurance.** The analysis was implemented in R version 4.2.1 and 4.4.2[62].

### Reporting summary
Further information on research design is available in the Nature Portfolio Reporting Summary linked to this article.

## Data availability
In accordance with the data sharing policies of the National Institute of Allergy and Infectious Diseases, permission to access data will have to be requested from the HVTN and Statistical Center for HIV/AIDS Research & Prevention (SCHARP). The trial dataset with data dictionary will be made available to appropriate academic parties with input from the investigator group subject to submission of a suitable study protocol and analysis plan. Requests should be directed to Peter Gilbert (pgilbert@fredhutch.org) and will be responded to within one month. Some of the findings of this study are based on Spike amino acid sequences obtained from 3,000 individual SARS-CoV-2 genome sequences available on GISAID up to December 20, 2024 (EPI_SET_241220es), and accessible at https://doi.org/10.55876/gis8.241220es (Supplementary Table 4). Source Data pertaining to Figs. 3, 4 and 6 are provided with this paper. Source data are provided with this paper.

## Code availability
All analyses were done reproducibly based on publicly available R scripts hosted on the GitHub collaborative platform at https://github.com/CoVPN/CoVPN3008/tree/master/Antibody_manucript[63].

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

## Acknowledgements

The authors thank the participants and study staff of the CoVPN 3008 trial for their dedication and contributions to the trial. The members of the CoVPN 3008 study team are listed in the Supplement. This work was supported by the National Institute of Allergy and Infectious Diseases (NIAID) of the National Institutes of Health (NIH) through awards UM1 AI068614-14 (HVTN/CoVPN LOC to L.C., G.E.G., and G.D.T.), UM1 AI068635 (HVTN/CoVPN SDMC to Y.H. and P.B.G.), 3UM1AI068618-15S1 (HVTN/CoVPN LC to M.J.M.), 3 UM1AI068619-17SI, 3 UM1AI068619-15S2, T32AI007044-45 (A.T.), R37AI054165 (P.B.G.) and by the Intramural Research Program of the NIAID Scientific Computing Infrastructure at Fred Hutch, under ORIP grant S10OD028685. The sponsor had no role in data collection and analysis or the decision to publish. National Institutes of Health co-authors reviewed and approved the manuscript. The content is solely the responsibility of the authors and does not necessarily represent the official views of the National Institutes of Health. This manuscript is the result of funding in whole or in part by the National Institutes of Health (NIH). It is subject to the NIH Public Access Policy. Through acceptance of this federal funding, NIH has been given a right to make this manuscript publicly available in PubMed Central upon the Official Date of Publication, as defined by NIH. We gratefully acknowledge all data contributors, i.e., the Authors and their Originating laboratories responsible for obtaining the specimens, and their Submitting laboratories for generating the genetic sequence and metadata and sharing via the GISAID Initiative, on which this research is based. We also acknowledge Paul C. Roberts of the Division of Microbiology and Infectious Diseases, National Institute of Allergy and Infectious Diseases, National Institutes of Health for facilitating provision of COVAIL neutralizing antibody data, and Moderna for providing the COVE study data. We thank the COVAIL and COVE investigators and study participants.

## Author contributions

Conceptualization: N.N.M., B.Z., C.B., P.J.E., A.Tapley, S.D., N.M.M., T.S., Z.Chirenje, J.M., E.K., H.N.-B., J.G.K., L.-G.B., L.C., G.E.G., Y.H., P.K., N.G., J.H., G.F., E.A.-N., D.M., P.L.M., M.J.M., G.D.T., P.B.G. Methodology: N.N.M., B.Z., C.B., K.S., M.S.-K., J.P., A.K., Z.Chen, S.H. Software: B.Z., J.H., A.H., Y.J., C.A.M., J.P., A.K., Z.Chen, S.H., P.B.G. Validation: N.N.M., B.Z., C.B., B.T.N.D., D.J.S., J.H., S.S., K.S., M.S.-K., A.H., Y.J., J.A., A.K.R., L.H.F., J.J.K., C.A.M., J.P., A.K., D.M., P.L.M., G.D.T., P.B.G. Formal analysis: N.N.M., B.Z., C.B., J.H., B.T.N.D., D.J.S., J.H., S.S., K.S., M.S.-K., A.H., Y.J., J.A., A.K.R., L.H.F., J.J.K., C.A.M., J.P., A.K., Z.Chen, S.H., G.D.T., P.B.G. Investigation: N.N.M., C.B., P.J.E., A.Tapley, S.D., B.T.N.D., D.J.S., J.H., S.S., K.S., M.S.-K., S.B., H.K., P.K., T.M., N.M.M., M.V., A.Takalani, B.L.R., E.W., J.O., P.S., L.P., M.Y., T.S., Z.Chirenje, J.M., E.K., K.N., H.N.-B., A.B., S.B.-F., W.B., S.C., R.D., S.D.-M., A.H.D., S.F., K.G., A.M., Z.A.E.H., M.C.H., M.I., C.I., S.I., D.K., H.M., P.K., M.C.K., W.K., F.L., M.Malahleha, V.L.M., G.M., P.A.M., K.M., E.M., Y.D., P.M., M.Moerane, T.M., S.N., S.M., V.N., A.Nana, A.Nanvubya, B.K., M.N., W.O., E.L.P., D.P., Z.P., J.S., Y.S., S.K., D.v.d.V., M.S.T., Y.V., D.O.W., J.G.K., L.-G.B., L.C., G.E.G., P.K., N.G., G.D.T., P.B.G. Resources: N.N.M., C.B., P.J.E., A.Tapley, S.D., B.T.N.D., D.J.S., J.H., S.S., K.S., M.S.-K., S.B., H.K., P.K., T.M., N.M.M., M.V., A.Takalani, B.L.R., E.W., J.O., P.S., L.P., M.Y., T.S., Z.Chirenje, J.M., E.K., K.N., H.N.-B., A.B., S.B.-F., W.B., S.C., R.D., S.D.-M., A.H.D., S.F., K.G., A.M., Z.A.E.H., M.C.H., M.I., C.I., S.I., D.K., H.M., P.K., M.C.K., W.K., F.L., M.Malahleha, V.L.M., G.M., P.A.M., K.M., E.M., Y.D., P.M., M.Moerane, T.M., S.N., S.M., V.N., A.Nana, A.Nanvubya, B.K., M.N., W.O., E.L.P., D.P., Z.P., J.S., Y.S., S.K., D.v.d.V., M.S.T., Y.V., D.O.W., J.G.K., L.-G.B., L.C., G.E.G., Y.H., P.K., N.G., J.H., G.F., E.A.-N., D.M., P.L.M., M.J.M., G.D.T., P.B.G. Data curation: N.N.M., B.Z., C.B., J.H., B.T.N.D., D.J.S., J.H., S.S., K.S., M.S.-K., A.H., Y.J., J.A., A.K.R., L.H.F., J.J.K., C.A.M., J.P., A.K., G.D.T., P.B.G. Writing — original draft: B.Z., L.N.C., P.B.G. Writing — review and editing: all coauthors. Visualization: B.Z., J.H., J.J.K., C.A.M., J.P., A.K., L.N.C., G.D.T., P.B.G. Supervision: N.N.M., P.J.E., A.Tapley, S.D., N.M.M., T.S., Z.Chirenje, J.M., E.K., H.N.-B., J.G.K., L.-G.B., L.C., G.E.G., Y.H., P.K., N.G., J.H., G.F., E.A.-N., D.M., P.L.M., M.J.M., G.D.T., P.B.G. Project administration: N.N.M., B.Z., C.B., P.J.E., A.T., S.D., K.S., M.S.-K., J.A., A.K.R., T.S., Z.Chirenje, J.M., E.K., H.N.-B., J.G.K., L.-G.B., L.C., G.E.G., Y.H., P.K., N.G., J.H., G.F., E.A.-N., D.M., P.L.M., M.J.M., G.D.T., P.B.G. Funding acquisition: A.Tapley, L.C., G.E.G., Y.H., M.J.M., G.D.T., P.B.G.

## Competing interests

L.N.C. was compensated for work on this manuscript through a consulting agreement with Fred Hutchinson Cancer Center. L.C., F.L., S.B.-F., S.D.-M., M.C.H., C.A.M., E.A.-N., Y.H., M.J.M., G.M., N.M.M., P.L.M., and J.A. received funding from NIAID/NIH paid to their institutions. S.D. reports salary support from Johns Hopkins University. M.J.M. reports NIAID and Bill & Melinda Gates Foundation payments to her institution and payment from Stanford, CROI, NIH VRC, and the Ragon Institute. P.B.G. received funds from NIAID/NIH paid to his institution, consulting fees from Curevo Vaccine Company and MinervaX Vaccine Company, served unpaid on a Moderna Advisory Board for Zika vaccines, and received contracts to his institution for AstraZeneca Vaccine SAB, Sanofi SAB, and Vaccine Company SAB. All other authors have nothing to declare.

## Additional information

Nonhlanhla N. Mkhize [1,2,71], Bo Zhang [3,71], Caroline Brackett [4,71], Peter James Elyanu[5], Asa Tapley[3,6,7], Sufia Dadabhai[8], Jiani Hu[3], Bich T. N. Do[4], Daniel J. Schuster [4], Jack Heptinstall[4], Sheetal Sawant[4], Kelly Seaton [4], Marcella Sarzotti-Kelsoe[4], Aaron Hudson[3,9,10], Yutong Jin[3], Sinethemba Bhebhe[1,2], Haajira Kaldine[1,2], Prudence Kgagudi[1,2], Tandile Modise[1,2], Nyaradzo M. Mgodi [11], Jessica Andriesen[3], April K. Randhawa [3], Leigh H. Fisher[3], Jia Jin Kee[3], Craig A. Magaret [3], James Peng[9], Avi Kenny[12,13], Lindsay N. Carpp[3], Zhe Chen[14], Siyu Heng [15], Manuel Villaran[3], Azwidihwi Takalani[16], Bert Le Roux[16], Eduan Wilkinson [17,18], Jackline Odhiambo[3,16], Parth Shah [10], Laura Polakowski[19], Margaret Yacovone [19], Taraz Samandari [20], Zvavahera Chirenje[11], Joseph Makhema[21], Ethel Kamuti[22], Katanekwa Njekwa[22], Harriet Nuwagaba-Biribonwoha[23,24], Allan Baguma[5], Sharlaa Badal-Faesen [25], William Brumskine [26], Soritha Coetzer[27], Rodney Dawson[28], Sinead Delany-Moretlwe[29], Andreas Henri Diacon[30], Samantha Fry[31], Katherine Gill [32], Anda Madikida[32], Zaheer Ahmed Ebrahim Hoosain[33], Mina C. Hosseinipour[34,35], Mubiana Inambao[36], Craig Innes[37], Steve Innes [32], Dishiki Kalonji[38], Humphrey Mwape[39], Priya Kassim[40], Melvin C. Kamanga[41], William Kilembe[42], Fatima Laher[43], Mookho Malahleha[44], Vongane Louisa Maluleke[45], Grace Mboya[46], Philister Adhiambo Madiega[47], Kirsten McHarry[48], Essack Mitha[49], Yajna Duki[50], Pamela Mda[51], Moroesi Moerane[51], Tumelo Moloantoa[52], Simpson Nuwamanya[53], Sharana Mahomed [54], Vimla Naicker[55],

Anusha Nana[40], Annet Nanvubya[56], Barbarah Kawoozo[56], Maphoshane Nchabeleng[57], Walter Otieno[58,59], Elsje Louise Potgieter [60], Disebo Potloane[54], Zelda Punt[61], Jamil Said [62], Yashna Singh[32], Sheetal Kassim[32], Dorothie van der Vendt[32], Mohammed Siddique Tayob[63], Yacoob Vahed[64], Deo Ogema Wabwire[65], James G. Kublin [3], Linda-Gail Bekker [32], Lawrence Corey [3], Glenda E. Gray[66], Yunda Huang [3,67], Philip Kotze[68], Nigel Garrett [32,54,69], John Hural[3], Guido Ferrari [4], Erica Andersen-Nissen [3,70], David Montefiori [4], Penny L. Moore [1,2,54], M. Juliana McElrath [3], Georgia D. Tomaras [4,72] ✉, Peter B. Gilbert [3,9,10,72] ✉, CoVPN 3008 Study Team

[1]National Institute for Communicable Diseases, National Health Laboratory Service, Johannesburg, South Africa. [2]Faculty of Health Sciences, SAMRC Antibody Immunity Research Unit, University of the Witwatersrand, Johannesburg, South Africa. [3]Vaccine and Infectious Disease Division, Fred Hutchinson Cancer Center, Seattle, WA, USA. [4]Departments of Surgery and Integrative Immunobiology, Center for Human Systems Immunology, Duke University, Durham, NC, USA. [5]Baylor College of Medicine Children's Foundation-Uganda, Kampala, Uganda. [6]Division of Allergy and Infectious Diseases, Department of Medicine, University of Washington, Seattle, WA, USA. [7]Department of Medicine, University of Cape Town, Cape Town, South Africa. [8]Johns Hopkins Research Project, Blantyre, Malawi. [9]Department of Biostatistics, University of Washington, Seattle, WA, USA. [10]Public Health Sciences Division, Fred Hutchinson Cancer Center, Seattle, WA, USA. [11]Clinical Trials Research Centre, University of Zimbabwe, Harare, Zimbabwe. [12]Department of Biostatistics and Bioinformatics, Duke University, Durham, NC, USA. [13]Duke Global Health Institute, Duke University, Durham, NC, USA. [14]Department of Biostatistics, Epidemiology and Informatics, University of Pennsylvania, Pennsylvania, PA, USA. [15]Department of Biostatistics, School of Global Public Health, New York University, New York, NY, USA. [16]Hutchinson Center Research Institute of South Africa, Chris Hani Baragwanath Academic Hospital, Soweto, South Africa. [17]KwaZulu-Natal Research Innovation & Sequencing Platform, School of Laboratory Medicine and Medical Sciences, University of KwaZulu-Natal, Durban, South Africa. [18]Centre for Epidemic Response & Innovation, Stellenbosch, South Africa. [19]National Institute of Allergy and Infectious Diseases, National Institutes of Health, Bethesda, MD, USA. [20]COVID-19 Prevention Network, Seattle, WA, USA. [21]Botswana Harvard AIDS Institute, Gaborone, Botswana. [22]Centre for Infectious Disease Research in Zambia, Lusaka, Zambia. [23]ICAP at Columbia University, Eswatini Prevention Center, Mbabane, Eswatini. [24]Department of Epidemiology, Mailman School of Public Health, Columbia University, New York, NY, USA. [25]Clinical HIV Research Unit / Helen Joseph Clinical Research Site, Johannesburg, South Africa. [26]The Aurum Institute, Rustenburg Clinical Research Site, Rustenburg, South Africa. [27]Synexus Helderberg, Cape Town, South Africa. [28]University of Cape Town Lung Institute Clinical Research Site, Cape Town, South Africa. [29]Wits RHI University of the Witwatersrand, Johannesburg, South Africa. [30]TASK, Cape Town, South Africa. [31]FAMCRU Family Clinical Research Unit, Cape Town, South Africa. [32]Desmond Tutu HIV Centre, University of Cape Town, Cape Town, South Africa. [33]Josha Research Clinical Research Site, Bloemfontein, South Africa. [34]Malawi Clinical Research Site, Lilongwe, Malawi. [35]University of North Carolina at Chapel Hill School of Medicine, Chapel Hill, CA, USA. [36]CFHRZ - Ndola Clinical Research Site, Ndola, Zambia. [37]The Aurum Institute, Klerksdorp Clinical Research Site, Klerksdorp, South Africa. [38]South African Medical Research Council, Isipingo Clinical Research Site, KwaZulu-Natal, South Africa. [39]UNC Global Projects / Kamwala District Health Centre, Lusaka, Zambia. [40]Soweto - Kliptown Clinical Research Site, Soweto, South Africa. [41]Johns Hopkins Research Project-Kamuzu University of Health Sciences, Blantyre, Malawi. [42]CFHRZ Clinical Research Site, Lusaka, Zambia. [43]Perinatal HIV Research Unit, Faculty of Health Sciences, University of the Witwatersrand, Witwatersrand, South Africa. [44]Synergy Biomed Research Institute, East London, South Africa. [45]MERC Middelburg, Middelburg, South Africa. [46]Kisumu Clinical Research Site, Kisumu, Kenya. [47]Kenya Medical Research Institute, Nairobi, Kenya. [48]TASK Eden, Western Cape, South Africa. [49]Newtown Clinical Research, Johannesburg, South Africa. [50]Aurum Tembisa Clinic 4, Gauteng, South Africa. [51]Nelson Mandela Academic Clinical Research Unit, Walter Sisulu University, Mthatha, South Africa. [52]PHRU Matlosana Clinical Research Site, Klerksdorp, South Africa. [53]Joint Clinical Research Centre, Lubowa, Uganda. [54]Centre for the AIDS Programme of Research in South Africa, University of KwaZulu–Natal, Durban, South Africa. [55]South African Medical Research Council, Tongaat Clinical Research Site, KwaZulu-Natal, South Africa. [56]UVRI-IAVI HIV Vaccine Program Ltd. Clinical Research Site, Entebbe, Uganda. [57]MeCRU Clinical Research Site, Pretoria, South Africa. [58]Kombewa Clinical Research Site, Kisumu, Kenya. [59]Maseno University School of Medicine, Kisumu, Kenya. [60]Synexus Stanza Clinical Research Centre Clinical Research Site, Pretoria, South Africa. [61]PHOENIX Pharma (Pty) Ltd, Port Elizabeth, South Africa. [62]Moi University Clinical Research Centre, Eldoret, Kenya. [63]MERC Kempton, Kempton, South Africa. [64]MERC Welkom, Welkom, South Africa. [65]MU-JHU Research Collaboration Clinical Research Site, Kampala, Uganda. [66]South African Medical Research Council, Pretoria, South Africa. [67]Department of Global Health, University of Washington, Seattle, WA, USA. [68]Qhakaza Mbokodo Research Clinic, Ladysmith, South Africa. [69]Discipline of Public Health Medicine, School of Nursing and Public Health, University of KwaZulu-Natal, Durban, South Africa. [70]Cape Town HVTN Immunology Laboratory, Hutchinson Centre Research Institute of South Africa, Cape Town, South Africa. [71]These authors contributed equally: Nonhlanhla N. Mkhize, Bo Zhang, Caroline Brackett. [72]These authors jointly supervised this work: Georgia D. Tomaras, Peter B. Gilbert. *A list of authors and their affiliations appears at the end of the paper. ✉e-mail: gdt@duke.edu; pgilbert@fredhutch.org

## CoVPN 3008 Study Team

Eduan Wilkinson [17,18], Sharlaa Badal-Faesen [25] & Soritha Coetzer[27]

A full list of members and their affiliations appears in the Supplementary Information.

