## [Peer Review file · Nature Communications]

Neutralizing and Binding Antibodies are a Correlate of Risk of COVID-19 in the CoVPN 3008 Study in People with HIV

Corresponding Author: Professor Peter Gilbert

Version 0:

Reviewer comments:

Reviewer #1

(Remarks to the Author)

This study by Mkhize et al. provides an important and timely contribution to our understanding of immune correlates against SARS-CoV-2 among people with HIV (PWH). The authors leverage data from a large, multi-country trial during a period of significant SARS-CoV-2 viral evolution. The design comparing hybrid versus vaccine-only immunity is well-conceived and reflects real-world scenarios. The use of multiple immunological assays and sophisticated statistical approaches, including causal inference methods, strengthens the analyses and conclusions.

Major Comments

1. The introduction would benefit from a clearer explanation of the terms “correlate of risk” and “correlate of protection,” with citations of landmark studies that originally established their role in vaccine licensure and immunobridging. A short, plain-language paragraph could help orient readers unfamiliar with immunological endpoints.
2. Given that follow-up durations vary due to protocol amendments and early exits (notably around Month 6), the authors should more clearly discuss how this impacts the internal validity of correlates over time—particularly for the vaccine group. Consider elaborating on whether and how this may bias estimates of long-term protection.
3. The paper attributes unexpectedly higher peak antibody titers in PWH to “unmeasured confounding,” yet does not explore what confounders might plausibly account for this pattern. Potential explanations—such as differential timing of infection, ART adherence, or prior undetected infections—should be hypothesized or discussed.
4. CD4+ T-cell counts were available for PWH and could have been used to explore immunogenicity gradients, especially since prior studies have linked CD4 count with vaccine response. Including a scatter plot or stratified analysis by CD4 level (e.g., <200, 200–500, >500) could enhance the discussion on immune recovery and vaccine responsiveness.
5. The Statistical Analysis Plan (SAP) lists 281 hybrid and 121 vaccine participants for immunogenicity, but the manuscript states 287 and 115, respectively. This discrepancy should be addressed with a brief note confirming which sample sizes were used in each analysis.
6. The classification of participants who never developed symptomatic COVID-19 as “non-cases” could bias estimates if asymptomatic infections occurred and differed by group. Clarify whether asymptomatic RT-PCR positive individuals were censored or excluded. If they were censored, a competing risks framework may be more appropriate and should be acknowledged.
7. The discussion highlights the waning utility of peak titers over time, but it could more explicitly state that resting (post-peak) titers may better inform decisions around booster timing. This practical implication deserves emphasis for translational relevance.
8. The manuscript could include a brief caveat about how the case-cohort design might bias correlates estimates if sampling is imperfect or exposure misclassification occurs. A short paragraph on this methodological limitation would provide balance.

Minor Comments

1. The authors should use consistent phrasing for groups (e.g., “hybrid immunity group” vs. “Hybrid Group”) and antibody markers (e.g., nAb-ID50 vs. neutralizing antibody titer) throughout for clarity.
2. Some abbreviations like “RIS” (Random Immunogenicity Subset) and “IAS” (Immunogenicity Analysis Set) are introduced in the results with limited explanation. Brief reminders or footnotes would aid readability for non-specialist readers.
3. Figure legends could be more detailed, especially for key plots like Figure 3 and 4. It would help to explicitly note the

follow-up period and the interpretation of threshold-based risk analyses.

4. The discussion section is comprehensive but occasionally dense. Consider briefly summarizing key findings at the beginning of the Discussion for reader orientation before delving into comparisons with prior studies.

(Remarks on code availability)

Reviewer #2

(Remarks to the Author)

(Remarks on code availability)

Code was unavailable.

Reviewer #3

(Remarks to the Author)

Neutralizing and Binding Antibodies are a Correlate of Risk of COVID-19 in the CoVPN 3008 Study in People with HIV

Authors: Nonhlanhla N. Mkhize, Bo Zhang, Caroline Brackett, et al. for the CoVPN 3008 Study Team

Manuscript ID: 592966_0_art_0_svhf55

Review

In the manuscript “Neutralizing and Binding Antibodies are a Correlate of Risk of COVID-19 in the CoVPN 3008 Study in People with HIV” by Nonhlanhla N. Mkhize et al., the authors present an analysis of the Coronavirus Prevention Network (CoVPN) 3008 study. They estimate the effect of hybrid immunity through infection with SARS-CoV-2 followed by vaccination, and vaccine-only immunity on symptomatic COVID-19 in 11,681 individuals with and without HIV. They find that neutralizing antibodies are a good predictor of symptomatic COVID-19 and that hybrid immunity provides better protection than vaccine-only immunity from symptomatic COVID-19.

The manuscript addresses a relevant topic but requires substantial revisions for clarity and rigor. The introduction lacks a proper overview of the current literature and instead reads like a blend of methods and results, failing to establish the study’s rationale. Throughout the manuscript, the authors are often vague and imprecise in their wording. For example, the sentence “[...] pre-vaccination IgG binding antibodies [...], as well as neutralizing antibodies, all inversely correlated with COVID-19 [...]” (ll. 512–515) seems to suggest a reduced risk of symptomatic COVID-19, yet the term “symptomatic” is frequently omitted without clarification. Similarly, key terms like “inverse correlate” are used without previously defining the term.

The manuscript’s narrative is often difficult to follow, with unclear analytical goals. In several cases, analyses are not well motivated, and figures are missing some uniformity and quality. The authors also interpret large confidence intervals that include the null effect as indicative of trends, which is questionable and should be more cautiously framed (e.g., ll. 354–357).

Overall, the manuscript would benefit a lot from a clearer structure, more precise language, and greater attention to statistical interpretation and terminology. Only after this is done a comprehensive assessment of the scientific content is possible.

Issues:

Abstract:

=====

General: The abstract needs to be completely rewritten. It does not include an introduction to the field, nor information on a more detailed background, nor does it state the general problem clearly, nor does it put the results into the general context. It reads like a summary of the results.

L. 154: “reduce COVID-19”: A clearer description would be “reduce COVID-19 symptoms” or “reduce number of COVID-19 infections”.

Introduction:

=====

ll. 172-177: These are results and should not appear in the introduction.

Results:

=====

- Section “Sets of participants included in immunogenicity and immune correlates analyses selected through a case-cohort sampling design”:

L. 214: It should be described what M0 and Peak are at least once before.

ll. 214-218: It is very unclear what the Serum-IAS or the Serum-RIS are and whether the Serum-RIS is the same as the RIS. From the Supplementary Figure 1 it seems that they are sampled from the case-cohort subjects across the two groups PWH

and PWOH, which is not well described in this section. Additionally, it is unclear why they perform this additional random sampling. Since this is a crucial analysis step, this needs to be much clearer.

L. 220: The abbreviation NAAT should be spelled out here.

- Section: "Participant demographics"

Supplementary Table 1: Similar to a Supp Fig 12, the groups could be visualized better by grouping two columns together. Section "Immune response markers at M0 and Peak differ between the Hybrid and Vaccine Groups and are similar between people with and without HIV":

"Linear regression is used to provide covariate-adjusted point and 95% confidence interval estimation of geometric mean values." What do you mean by geometric mean here? The geometric mean is a sample estimate but they are estimating true quantities. I guess they are referring to estimating $E(\ln(\text{outcome}))$ but the term seems to be inappropriate.

Fig. 1: There is no indication of significance testing in panel B. There is also no mention of the exact test used that the p-values represent. Could also not find it in the "Statistical methods" section.

L. 715: As the bootstrap does not sample from H_0 , I suggest to rather use a permutation test instead of a bootstrap in order to examine the difference between hybrid and vaccine.

Fig. 1: Better grouping in the x-axis labels would help the visualization.

LI. 247-259: Since Figure 1 includes panel labels, I would suggest to reference them in the text.

- Section "Correlations of immune response markers at M0, at Peak, and across time points"

L. 266: For the correlation coefficient "r" is commonly used and not "rho".

p-values should be computed for the correlations.

LI. 264-266: Where can one see the correlations of antibody responses to different Spike antigens?

LI. 266-269: A correlation coefficient of 0.8 would be considered a high correlation. So, there is not really more independent information for Spike BA.4/5 and N Index.

LI. 273-276: It is unclear how vaccinations lead to less strong correlations. Also, from the data it does not look like "nearly all persons with hybrid immunity at enrolment had [...] M0 [IgG Spike BA.4/5] values between 100 and 10,000 AU/ml". It seems that it's less than half.

- Section "Higher neutralizing antibody titers to BA.4/5 at enrolment are an independent inverse correlate of post-vaccination COVID-19 risk among those with hybrid immunity":

Table 1: Lines would help to distinguish the different rows and columns.

L. 296f: Why not perform some statistical test on the measurements directly, instead of calculating geometric mean ratios?

L. 302: What covariates did the Cox model adjust for? Since the methods come after the results it would be good to mention them here. Maybe not too detailed, just something like "adjusted for age, sex and BMI".

L. 304: More precisely IgG N at M0 was not significantly associated with a risk reduction of symptomatic COVID-19.

LI. 307-310: How is non-significant risk reduction for the marker levels at Peak recapitulated by significant risk increase through antibody fold-rise? The fold-rise result seems counter-intuitive. See also issue in discussion.

Table 2: Panel B: Which variable do the p-values correspond to?

LI. 312f: Since Table 2 is separated into A and B it should be referenced properly here.

L. 314 "are the inverse correlate": It is unclear what the authors want to say here.

L. 316: Unclear what the authors want to say with "the independent correlate". Independent is a well-defined statistical term but, as far as I understand, not used in the statistical sense here. This is confusing to me.

L. 321: Controlled-risk causal analysis: It should be clarified in the main text that causality is assumed because of the inverse probability weighting. Maybe also mention against which type of misspecification the inverse probability weighting helps.

L. 324f: "COVID-19 risk [...] trends with increasing antibody level at Peak" is very inspecific. It should be made clearer what happens with higher antibody levels.

All Figures (e.g. 10 + 11) need to have clear labels of the used colors/curves within the figure. Reading the figure caption first to find out about the meaning makes it difficult to interpret the figure.

Section "In the Vaccine Group, the antibody markers IgG Spike and neutralization titers at Peak are higher in non-cases vs. vaccination-proximal cases and similar vs. vaccination-distal cases":

LI. 338-343: The first sentence is almost a direct copy of the title. This section is also very short, so why not include it somewhere else?

- Section "Vaccine Group Peak antibody levels inversely correlate with risk of COVID-19, but only in early vaccination-proximal follow-up":

LI. 371f: There seems to be no uncertainty displayed here. Why is that?

LI. 374-377: The PWH group was not mentioned in the paragraph before. How do these data support this claim then?

- Section "Antibody marker correlates of risk are similar for people with and without HIV":

Supplementary Figure 12: The visualization in the supplementary of the Cox model results is better than in the main text. Maybe adapt the main text figures according to this.

Why not estimate the effect size of PWH in the Cox model directly?

L. 385: What does it mean to perform an interaction test? It should probably be called the test of the interaction term.

L. 388: Is multiple testing corrected p-value meant here?

- Section "Correlates of protection analyses support that the Hybrid Group advantage over the Vaccine Group was partially mediated through the antibody markers":

Figure 5: There should be a legend for the figure such that it is possible to understand which curve belongs to which y-axis.

Section: "Circulating strains have similar Spike sequence antigenic distances to the mRNA-1273 vaccine strain in CoVPN 3008 compared to the COVE and COVAIL correlates studies"

This section seems very disconnected from the rest and it is unclear why this antigenic distance is even studied.

- Section "Quantitative comparison of CoVPN 3008 neutralizing antibody correlates results to those of previous mRNA-1273 vaccine immune correlates studies":

Figure 6: A-D: Using two different x-axes on one graph is too confusing. Some of the content should be moved to the supplementary. E-J: Missing x-axis labels. y-axis labels should be "density" and not the study name.

L. 451: If the subfigures E, G, and I are analyzed first, they should appear as A, B, and C for better reliability.

Discussion:

=====

Ll. 519f: What is multiplicity correction? Is multiple testing correction meant here?

Ll. 522: "Suggestive of correlates" because they were not significantly associated with a reduced risk of COVID-19? This should be stated more clearly.

Ll. 532-539: It seems counter intuitive that a larger fold-change between pre and post-vaccination would lead to an increased hazard. However, it is never explained what the reason for this could be. It seems that this is simply ignored. If it is not a good marker to use, then it simply should not appear in the model or if it does it should be discussed more in detail.

L. 580: Second time that the wording "mark [...] risk" was used. It is unclear what the authors mean. Is the meaning simply "antibodies reduce risk"? Later in the text (l. 594) it is also written "mark protection".

L. 588: The authors mention "both markers" but do not mention the specific markers.

Ll. 588-602: The comparison with COVE and COVAIL could be combined with the other section where a comparison was done.

L. 611: How is "variability" of the antibodies important and not just the level of the antibodies?

Methods:

=====

L. 646: What is "baseline"? Is that the same as enrolment M0 or something else?

L. 647: What is a "point-of-care" serology test? Should be explained more.

Ll. 665-667: This sounds like COVID-19 endpoints were only counted as COVID-19 endpoints if they occurred within 7 days after the Peak or after 230 days after the Peak.

L. 681: If there are measurements that are in BAU/mL, why are all results reported in AU/mL?

Ll. 682-685: It would be nice to see the different assay measurements; not just the three that are presented in the figures. This could become part of the supplementary. Or at least argue why they weren't used more clearly in the results.

Ll. 702f: Why are LOD values assigned LOD/2

L. 703: The abbreviation ULOQ was not explained.

Ll. 703-705: How were these thresholds chosen? What would be the effect of choosing them differently?

Ll. 714f: What are two-sided bootstrap p-values? What kind of test was performed?

L. 818: Since R versions or more specifically the packages are not always compatible, ideally one would only use a single version of R to ensure reproducibility.

(Remarks on code availability)

Version 1:

Reviewer comments:

Reviewer #1

(Remarks to the Author)

The authors have adequately addressed my concerns.

(Remarks on code availability)

Reviewer #2

(Remarks to the Author)

(Remarks on code availability)

Code still not available.

Reviewer #3

(Remarks to the Author)

The authors have addressed my main concerns, and the manuscript has improved substantially in both clarity and structure. I now recommend the manuscript for publication, pending minor revisions as outlined below. The abstract and introduction are clearer, and research goals are clearly outlined. The sampling procedure is now better explained, and the revisions have helped clarify the methodology. The coherence of the figures has improved overall, and I commend the authors for making their code available, which enhances the transparency and reproducibility of their work.

Some minor issues remain, which are mostly a result of the revisions (line numbers as in the revised manuscript):

- Paragraph following l. 190: The authors properly introduce CoRs and CoPs, however the description of the terms is very disconnected from the rest of the introduction. This could be improved.
- L. 232: Minor inconsistency. The group now referred to as "Serum-RIS" is still labeled simply as "RIS" in Supplementary Figure 1.
- Ll. 310f: This sentence seems disconnected from the rest and does not describe any results.
- L. 322: The "geometric mean" was already used in l. 273, but was not abbreviated here.
- Ll. 326f: The same as in ll. 310f.
- Paragraph following l. 451: The authors should consider adding the hazard ratios to the previous paragraph, as well.
- Ll. 589f: Missing reference to Omicron evolution.
- Ll. 617f: Missing reference to a study showing that B cell responses are important for protection.
- Ll. 621f: Missing reference to the limited cross-reactivity.
- L. 839: The authors have changed from the wording from estimation of the geometric mean to estimation of mean log-scale. This addresses the issue that I raised. However, they still talk about "geometric mean ratios" in the results (for example, l. 273 or 322). This needs to be changed, as it is now unclear where this geometric mean comes from.

(Remarks on code availability)

Note: Line numbers refer to the clean version of the manuscript.

REVIEWER COMMENTS

Reviewer #1 (Remarks to the Author):

This study by Mkhize et al. provides an important and timely contribution to our understanding of immune correlates against SARS-CoV-2 among people with HIV (PWH). The authors leverage data from a large, multi-country trial during a period of significant SARS-CoV-2 viral evolution. The design comparing hybrid versus vaccine-only immunity is well-conceived and reflects real-world scenarios. The use of multiple immunological assays and sophisticated statistical approaches, including causal inference methods, strengthens the analyses and conclusions.

Major Comments

1. The introduction would benefit from a clearer explanation of the terms “correlate of risk” and “correlate of protection,” with citations of landmark studies that originally established their role in vaccine licensure and immunobridging. A short, plain-language paragraph could help orient readers unfamiliar with immunological endpoints.

Response: We appreciate the feedback from the reviewer. We have now added the following paragraph:

“An immune correlate of risk (CoR) refers to an immune marker that is associated with an infectious disease outcome in a cohort such as a vaccine arm or pooled set of vaccine arms. When this association is inverse, i.e. higher immune marker levels are associated with lower risk of the infectious disease outcome, the correlate is referred to as an “inverse CoR.” On the other hand, an immune correlate of protection (CoP) refers to an immune marker that can be used to reliably predict a vaccine’s level of protection against an infectious disease outcome.¹⁻³ Immune CoPs are important for understanding basic immunology, providing endpoints for immunobridging regulatory approval of new vaccine formulations or extending indications to new populations, and modeling of vaccine efficacy and effectiveness. For example, CoPs have facilitated regulatory approval of modified (e.g., variant-adapted) vaccines, as well as approval for use in a different population (e.g., pediatric) other than the one in which the original phase 3 efficacy trial(s) was conducted, for pathogens including influenza virus⁴ and SARS-CoV-2.⁵” (lines 190-203)

2. Given that follow-up durations vary due to protocol amendments and early exits (notably around Month 6), the authors should more clearly discuss how this impacts the internal validity of correlates over time—particularly for the

vaccine group. Consider elaborating on whether and how this may bias estimates of long-term protection.

Response: We thank the reviewer for pointing out this great suggestion. The median follow-up time for the Vaccine Group is 112 days post last (second) study vaccination. The Cox regression-based estimates (HRs in the manuscript) are internally valid provided that the Cox regression model is correctly specified. On the other hand, for the cumulative incidence curve-based estimates, e.g., the covariate-adjusted, marker-specific risk curves, the methods we used adjusted for the right-censoring of times-to-COVID-19 using inverse probability of censoring weights (IPCW). The resulting estimates are valid when the censoring probability is correctly modeled as a function of the adjustment variables (region, HIV status, risk score, and period of enrollment). However, the estimates would be biased if the adjustment variables did not fully capture the censoring mechanism.

In the revised manuscript, we have added analyses for the Vaccine Group to Day 130 as Supplementary Figure 14. This contrasts with the analyses in the main article that follow for COVID-19 through Day 165. The value 130 days was selected because a substantial number of Vaccine Group participants had follow-up past this time point, whereas only a small minority reach Day 165. We found that the relationship between the marker level and Day 130 risk remained flat, similar to that between the marker level and Day 165 risk. We now state in the revision: “As a sensitivity analysis, Supplementary Figure 14 repeats the analyses through 130 days post Peak and shows similarly flat curves of COVID-19 risk for both Peak antibody markers.” (lines 402-405)

3. The paper attributes unexpectedly higher peak antibody titers in PWH to “unmeasured confounding,” yet does not explore what confounders might plausibly account for this pattern. Potential explanations—such as differential timing of infection, ART adherence, or prior undetected infections—should be hypothesized or discussed.

Response: Thank you for the comment. First, to clarify, we observed a higher Peak response in PWH compared to PWOH only in the Vaccine Group, not the Hybrid Group (“Surprisingly, in the Vaccine Group, Peak marker levels were significantly higher in PWH”, lines 549-550).

We have added the following (underlined): “We conjecture that this result may be caused by unmeasured confounding (such as differential antiretroviral therapy use, or differential rates of coinfections) that led to apparently higher responses in PWH in the Vaccine Group.” (lines 550-553)

4. CD4+ T-cell counts were available for PWH and could have been used to explore immunogenicity gradients, especially since prior studies have linked CD4 count with vaccine response. Including a scatter plot or stratified analysis

by CD4 level (e.g., <200, 200–500, >500) could enhance the discussion on immune recovery and vaccine responsiveness.

Response: Thank you for this good suggestion. Among the 103 PWH participants in the Serum-Random Immunogenicity Subset, only 12 had baseline CD4 counts less than 200 cells/mm³, and 30 less than 500 cells/mm³.

We have now added two scatter plots depicting the relationship between baseline CD4 counts and Baseline or Peak nAb-ID50 among PWH in the Serum-Random Immunogenicity Subsets as Supplementary Figure 2 in the revision.

We have also made the following revisions to the text:

“Within PWH, Peak nAb-ID50 BA.4/5 titer appears to trend higher in participants in the Hybrid Group with higher baseline CD4+ T-cell count; this finding is less apparent in the Vaccine Group (Supplementary Figure 2). In a post hoc linear regression analysis comparing Peak nAb-ID50 BA.4/5 titers among Serum-RIS PWH participants with baseline CD4+ T-cell count below vs. above 500 cells/mm³ and adjusting for baseline nAb-ID50 BA.4/5 titer and Hybrid vs. Vaccine Group, participants with baseline CD4+ T-cell count < 500 cells/mm³ had 0.23-log lower Peak nAb-ID50 BA.4/5 titer, although this finding is not significant (p=0.27).” (lines 276-283).

5. The Statistical Analysis Plan (SAP) lists 281 hybrid and 121 vaccine participants for immunogenicity, but the manuscript states 287 and 115, respectively. This discrepancy should be addressed with a brief note confirming which sample sizes were used in each analysis.

Response: Sorry for the confusion. Participants that contributed the marker data are listed in Table 5 of the SAP and reproduced below:

Study Group		1	2	3	4	Total
Analysis Group		1	2.1	3	4.1	
HIV		+	+	-	-	
SARS-CoV-2		-	+	-	+	
Serum	Non-cases	37	65	24	70	196
	Cases	49	130	5	22	206
	Total	86	195	29	92	402

The Hybrid Group analysis included Group 2.1 + 4.1, so N = 195 + 92 = 287. The Vaccine Group analysis included Group 1 + 3, so N = 86 + 29 = 115.

The sample size for PWH is 86 + 195 = 281. The sample size for PWoH = 29 + 92 = 121. With this added explanation, the set of numbers is compatible.

6. The classification of participants who never developed symptomatic COVID-19 as “non-cases” could bias estimates if asymptomatic infections occurred and differed by group. Clarify whether asymptomatic RT-PCR positive

individuals were censored or excluded. If they were censored, a competing risks framework may be more appropriate and should be acknowledged.

Response: We have clarified the definition of non-cases in the footnotes of Tables 1 and 3; the captions of Figures 2, 3, 4; and the captions of Supplementary Figures 9, 10, 11, and 16: "Non-cases did not have a positive RT-PCR result at the Peak visit and did not acquire a COVID-19 endpoint after M0 up to the date by which the last enrolled participant reached 230 days post Peak (March 31, 2023)." PCR tests were only administered routinely at M0, Peak, and the M6 visit. Therefore, undetected asymptomatic RT-PCR positive individuals may have been included among the non-cases in our analysis. We believe our analysis addresses an interesting question by studying time to COVID-19 ignoring asymptomatic SARS-CoV-2 infection, as a symptomatic disease endpoint is of interest in and of itself. Moreover, switching to a competing risks Cox model would change our scientific question, and also make the cumulative incidence based controlled risk and nonparametric thresholding methods give incorrect results, where significant methodological development (which we consider outside the scope of this manuscript) would be needed to extend these methods to handle competing risks.

Additional reasons for not implementing competing risk models include: 1) the competing risk framework requires that asymptomatic infection precludes the COVID-19 endpoint from happening, which may not be true; 2) surveillance in the study was symptom-triggered, so data on asymptomatic endpoints were collected through the Peak RT-PCR testing (where participants with positive results were excluded already) and through the Month 6 RT-PCR testing (where all participants are censored at the Month 6 visit anyway); and 3) in Garrett et al.,⁶ about 1% of participants in the entire study cohort developed asymptomatic SARS-CoV-2 infection. Therefore, among the ~200 non-cases, we would expect to see around 2 asymptomatic infections, which would be unlikely to affect our results in a material manner.

7. The discussion highlights the waning utility of peak titers over time, but it could more explicitly state that resting (post-peak) titers may better inform decisions around booster timing. This practical implication deserves emphasis for translational relevance.

Response: We have added "One implication of this waning utility of Peak titer as a correlate is that titers measured after Peak may better inform decisions around booster timing." (lines 592-593)

8. The manuscript could include a brief caveat about how the case-cohort design might bias correlates estimates if sampling is imperfect or exposure misclassification occurs. A short paragraph on this methodological limitation would provide balance.

Response: The cohort of the study was selected randomly; however, the reviewer is right that after applying all the exclusion criteria, we cannot completely rule out that the sampling becomes no longer completely random (within each analysis group). We have now added the following text to the paragraph discussing limitations in the discussion section:

“We also note that the study adopted a case-cohort design when sampling the immune markers. While the cohort was randomly selected, exclusion criteria were used to arrive at the final per-protocol serum correlates cohort. Estimates could be biased if the final per-protocol serum correlates cohort, after applying the exclusion criteria, did not remain a stratified random sample of the entire per-protocol cohort.” (lines 682-687)

Minor Comments

1. The authors should use consistent phrasing for groups (e.g., “hybrid immunity group” vs. “Hybrid Group”) and antibody markers (e.g., nAb-ID50 vs. neutralizing antibody titer) throughout for clarity.

Response: We have made these revisions, aided by Word’s search feature. Examples where we have made these changes include: line 218, lines 314-315, line 455, and line 498. At places where we are not referring to the specific Analysis Group analyzed in this work (Hybrid or Vaccine), but rather proposing how our conclusions might apply beyond our study, we have opted to retain the word “immunity” along with “population”. An example is “these antibody immune correlates can become weaker or shorter-lived in a naïve vaccine immunity population compared to a hybrid immunity population” (lines 536-537).

2. Some abbreviations like “RIS” (Random Immunogenicity Subset) and “IAS” (Immunogenicity Analysis Set) are introduced in the results with limited explanation. Brief reminders or footnotes would aid readability for non-specialist readers.

Response: We have made the following revisions (underlined) to the first paragraph of Results:

“The per-protocol Serum Immunogenicity Analysis Set (Serum-IAS) was based on sera stored from 11,697 participants in the per-protocol serum correlates cohort. According to the case-cohort sampling design, M0 and Peak marker levels were measured in all participants randomly sampled into the Serum Random Immunogenicity Subset (Serum-RIS), as well as in all evaluable breakthrough COVID-19 endpoints outside the Serum-RIS, together forming the Serum-IAS.” (lines 228-233)

3. Figure legends could be more detailed, especially for key plots like Figure 3 and 4. It would help to explicitly note the follow-up period and the interpretation of threshold-based risk analyses.

Response: We have revised these captions as follows (underlined):

Figure 3:

“(A, B) Covariate-adjusted hazard ratios (HRs) of COVID-19 [follow-up period was (A) 7 to 92 days post Peak or (B) 7 to 230 days post Peak] per 10-fold increase...” (C-G)...“The plots show the covariate-adjusted probability of COVID-19 over a follow up period of 165 days post Peak”

Figure 4:

“Covariate-adjusted hazard ratios (HRs) of COVID-19 [follow up period was: (A) 7 to 92 days post Peak, (B) 7 to 230 days post Peak]”
“Controlled risk plots for Vaccine Group (N = 115) antibody markers at Peak, over follow-up periods through 92 days or 165 days post Peak as noted.”

Figure 5:

“Each dot (threshold value) represents a point estimate of relative risk (compared to the entire Vaccine Group) of COVID-19 through 160 days post Peak for Hybrid Group participants if their marker levels were as high as or higher than that given threshold value.”

4. The discussion section is comprehensive but occasionally dense. Consider briefly summarizing key findings at the beginning of the Discussion for reader orientation before delving into comparisons with prior studies.

Results: Thank you for this excellent suggestion. We have rearranged the Discussion by moving the concluding paragraph, which summarizes the three main insights, to the beginning of the Discussion. We agree that this helps orient the reader to the most important conclusions of this work.

Reviewer #2 (Remarks to the Author):

Reviewer #2 (Remarks on code availability):

Code was unavailable.

Response: Sorry for the lack of clear communication about the code. As provided in our Code Availability statement, code is now available at Github via the following link:

https://github.com/CoVPN/CoVPN3008/tree/master/Antibody_manucript. The README file contains details on which RMD script reproduces which analyses in the manuscript. All analyses can be reproduced automatically using the provided code and the analysis-ready dataset.

Reviewer #3 (Remarks to the Author):

Neutralizing and Binding Antibodies are a Correlate of Risk of COVID-19 in the CoVPN 3008 Study in People with HIV

Authors: Nonhlanhla N. Mkhize, Bo Zhang, Caroline Brackett, et al. for the CoVPN 3008 Study Team

Manuscript ID: 592966_0_art_0_svhf55

Review

In the manuscript “Neutralizing and Binding Antibodies are a Correlate of Risk of COVID-19 in the CoVPN 3008 Study in People with HIV” by Nonhlanhla N. Mkhize et al., the authors present an analysis of the Coronavirus Prevention Network (CoVPN) 3008 study. They estimate the effect of hybrid immunity through infection with SARS-CoV-2 followed by vaccination, and vaccine-only immunity on symptomatic COVID-19 in 11,681 individuals with and without HIV. They find that neutralizing antibodies are a good predictor of symptomatic COVID-19 and that hybrid immunity provides better protection than vaccine-only immunity from symptomatic COVID-19.

The manuscript addresses a relevant topic but requires substantial revisions for clarity and rigor. The introduction lacks a proper overview of the current literature and instead reads like a blend of methods and results, failing to establish the study’s rationale. Throughout the manuscript, the authors are often vague and imprecise in their wording. For example, the sentence “[...] pre-vaccination IgG binding antibodies [...], as well as neutralizing antibodies, all inversely correlated with COVID-19 [...]” (ll. 512–515) seems to suggest a reduced risk of symptomatic COVID-19, yet the term “symptomatic” is frequently omitted without clarification. Similarly, key terms like “inverse correlate” are used without previously defining the term.

The manuscript’s narrative is often difficult to follow, with unclear analytical goals. In several cases, analyses are not well motivated, and figures are missing some uniformity and quality. The authors also interpret large confidence intervals that include the null effect as indicative of trends, which

is questionable and should be more cautiously framed (e.g., ll. 354–357).

Overall, the manuscript would benefit a lot from a clearer structure, more precise language, and greater attention to statistical interpretation and terminology. Only after this is done a comprehensive assessment of the scientific content is possible.

Response: Thank you for the thoughtful assessment of our manuscript. The issue of wording is addressed in several responses below, but additionally, we have clarified at the first instance in the main text that "...COVID-19" refers to "symptomatic COVID-19 [Centers for Disease Control and Prevention (CDC) criteria; hereafter referred to as "COVID-19"]..." (lines 172-173). We have also replaced all instances of "inverse correlate" with "inverse CoR"; we define "CoR" and "inverse CoR" in the introduction now: "An immune correlate of risk (CoR) refers to an immune marker that is associated with an infectious disease outcome in a cohort such as a vaccine arm or pooled set of vaccine arms. When this association is inverse, i.e. higher immune marker levels are associated with lower risk of the infectious disease outcome, the correlate is referred to as an "inverse CoR." "(lines 190-194)

Issues:

Abstract:

=====

General: The abstract needs to be completely rewritten. It does not include an introduction to the field, nor information on a more detailed background, nor does it state the general problem clearly, nor does it put the results into the general context. It reads like a summary of the results.

Response: Thank you for this suggestion. We have revised the abstract, with the revisions including: 1) beginning with a sentence briefly stating the motivation and context for the study ("*People with HIV (PWH) have been understudied in COVID-19 vaccine trials, raising questions as to whether COVID-19 vaccines are as immunogenic and efficacious in PWH, and if the identified immune correlates of protection also hold in PWH*", lines 133-135); and 2) removing some of the results quoted in the abstract [e.g.: "~~Relative risk of COVID-19 in the Hybrid group (vs. overall risk in the Vaccine group) was 0.46 (0.28, 0.64) when thresholding M0 nAb-ID50 BA.4/5 titer above 10 arbitrary units (AU)/ml (5th percentile) and decreased to 0.23 (0.10, 0.35) at titer above 1000 AU/ml (70th percentile)~~"] has been removed].

L. 154: "reduce COVID-19": A clearer description would be "reduce COVID-19 symptoms" or "reduce number of COVID-19 infections".

Response: We have revised to "to reduce COVID-19 incidence" (line 151).

Introduction:

=====

LI. 172-177: These are results and should not appear in the introduction.

Response: Thank you for the opportunity to clarify. These results are from the primary CoVPN 3008 paper (Garrett et al., eClinicalMedicine 2025⁶), which reported on the safety of the mRNA-1273 vaccine, the relative effectiveness of hybrid versus vaccine immunity, and SARS-CoV-2 viral persistence among PWH in East and Southern Africa during the Omicron outbreak. The present study analyzed the data from this study, and therefore we provide the Garrett et al. results as context for our current correlates analysis that addressed new objectives.

We have clarified (underlined) that the results in the text to which the reviewer refers are from a previous paper:

“Garrett et al.⁶ previously reported the results of the primary safety and relative risk (Hybrid vs. Vaccine Group) objectives of CoVPN 3008, with a major finding being that the Hybrid Group had lower risk of symptomatic COVID-19 [Centers for Disease Control and Prevention (CDC) criteria; hereafter referred to as “COVID-19”] and of severe COVID-19 between 1 day and 6 months post-enrolment compared to the Vaccine Group.” (lines 170-175)

Results:

=====

- Section “Sets of participants included in immunogenicity and immune correlates analyses selected through a case-cohort sampling design”:

L. 214: It should be described what M0 and Peak are at least once before.

Response: We have now clarified that M0 refers to “day of enrolment; before receiving any study vaccination,” and Peak refers to “one month after the last study vaccination.” (lines 226-227)

LI. 214-218: It is very unclear what the Serum-IAS or the Serum-RIS are and whether the Serum-RIS is the same as the RIS. From the Supplementary Figure 1 it seems that they are sampled from the case-cohort subjects across the two groups PWH and PWOH, which is not well described in this section.

Additionally, it is unclear why they perform this additional random sampling. Since this is a crucial analysis step, this needs to be much clearer.

Response: A common strategy to study the association between an immune marker and clinical endpoints is to conduct a ‘case-cohort’ or ‘case-control’ sampling design, the rationale of adopting a sampling design is that it is often not realistic or cost-effective to measure the immune marker on all study participants. Instead of measuring the immune marker from all study participants, marker levels measured from a random cohort plus participants with the infectious disease endpoints of

interest are used to estimate the association using statistical methods that account for the sampling design.

In this project, we adopted a 'case-cohort' sampling design. RIS refers to the random cohort that contributed the immune marker measurements, while IAS refers to the union of RIS and all symptomatic cases that also contributed the marker measurements. When forming a random cohort, we adopted a stratified random sampling scheme, stratified by each analysis group (defined by baseline HIV status and baseline SARS-CoV-2 status).

In this manuscript, the RIS used is always the Serum-RIS. We appreciate the opportunity to clarify this and every instance of "RIS" has now been revised to "Serum-RIS".

L. 220: The abbreviation NAAT should be spelled out here.

Response: We have clarified that NAAT stands for Nucleic Acid Amplification Test: "one Nucleic Acid Amplification Test (NAAT) positive nasal swab" (lines 234-235).

- Section: "Participant demographics"

Supplementary Table 1: Similar to a Supp Fig 12, the groups could be visualized better by grouping two columns together.

Response: Within the Vaccine and Hybrid groups, we believe it is important to present the demographics in PWH and PWOH separately. The addition of two more columns (PWH + PWOH pooled, within each of the Hybrid and Vaccine groups) would potentially hinder the readability of this table by making it too wide; therefore, we have opted not to take this suggestion.

Section "Immune response markers at M0 and Peak differ between the Hybrid and Vaccine Groups and are similar between people with and without HIV": "Linear regression is used to provide covariate-adjusted point and 95% confidence interval estimation of geometric mean values." What do you mean by geometric mean here? The geometric mean is a sample estimate but they are estimating true quantities. I guess they are referring to estimating $E(\ln(\text{outcome}))$ but the term seems to be inappropriate.

Response: Thank you. We have modified the sentence as follows:

"Linear regression is used to provide covariate-adjusted point and 95% confidence interval estimation of the mean log-scale antibody marker levels at M0 and Peak for the four subgroups defined by (Hybrid, Vaccine) \times (PWH, PWOH)." (lines 838-840)

Fig. 1: There is no indication of significance testing in panel B. There is also no mention of the exact test used that the p-values represent. Could also not

find it in the “Statistical methods” section.

Response: Because the mRNA-1273 vaccine does not contain the Nucleocapsid (N) protein, we expect M0 and Peak IgG N measurements be nearly identical (up to measurement error). Hence, we only analyzed the M0 N protein but not Peak N protein throughout the article. Panel B of Figure 1 is intended to document the expectation that Peak IgG N measurements are similar to those measured at M0 in Panel A.

The null hypothesis being tested here is the equality of the covariate-adjusted mean marker level in two comparison groups, e.g., Hybrid PWH vs Vaccine PWH. For instance, when the goal is to compare the Hybrid PWH vs Vaccine PWH, then the null hypothesis is testing $E[Y(0)] = E[Y(1)]$, where $E[Y(0)]$ is the counterfactual mean of Y if hypothetically all participants were in the $a=0$ Hybrid PWH group and $E[Y(1)]$ is the counterfactual mean of Y if hypothetically all participants were in the $a=1$ Vaccine PWH group. The mean outcome $E[Y | \text{Group} = a]$ is estimated using g-computation as $E[E[Y | \text{Group} = a, X]]$, for each $a = 0, 1$. Importantly, when estimating $E[Y | \text{Group} = a]$, $a = \text{Hybrid PWH or Vaccine PWH}$, we used the same marginal distribution of X (empirical distribution of X in Hybrid PWH union Vaccine PWH). This is a standard approach to causal inference for the evaluation of an average treatment effect.

To obtain the p-value, the observed difference in the covariate-adjusted marginalized mean was compared to the bootstrapped distribution of the difference in the covariate-adjusted marginalized mean. This is stated in the caption of Fig. 1: “Bootstrapped 2-sided p-values compare Hybrid vs. Vaccine and Hybrid PWH vs. Hybrid PWoH for markers at M0, and compare Hybrid vs. Vaccine, Hybrid PWH vs. Hybrid PWoH, and Vaccine PWH vs. Vaccine PWoH for markers at Peak,” as well as in “Comparison of antibody markers between key demographic groups” in “Statistical Methods”: “Two-sided bootstrap p-values are reported for the test of whether marginalized mean \log_{10} -scale M0 levels significantly differ by Hybrid vs. Vaccine (combining across PWH and PWoH) and by Hybrid PWH vs. Hybrid PWoH, and for whether marginalized mean Peak levels significantly differ by Hybrid vs. Vaccine, by Hybrid PWH vs. Hybrid PWoH, and by Vaccine PWH vs. Vaccine PWoH. The marginalized mean outcome was calculated using g-computation. The bootstrap p-value was obtained by comparing the observed difference in the marginalized mean outcome in each group to its bootstrapped distribution.” (lines 840-848)

L. 715: As the bootstrap does not sample from H0, I suggest to rather use a permutation test instead of a bootstrap in order to examine the difference between hybrid and vaccine.

Response: Thank you for the suggestion. We chose to still use the g-computation and bootstrap-based method because the goal is to test the difference in the covariate-adjusted mean outcome. We also seek to limit changes to the Statistical

Analysis Plan after conducting the analysis.

Fig. 1: Better grouping in the x-axis labels would help the visualization.

Response: Thank you for this suggestion, which we have implemented.

LI. 247-259: Since Figure 1 includes panel labels, I would suggest to reference them in the text.

Response: We have added references to panel labels in Figure 1 throughout this paragraph (lines 268-275).

- Section “Correlations of immune response markers at M0, at Peak, and across time points”

L. 266: For the correlation coefficient “r” is commonly used and not “rho”. p-values should be computed for the correlations.

Response: We have replaced all “rho” with “ ρ ”. P-values have also been added to the captions of the relevant supplementary figures, where the p-values can be found either as insets in the plots (Supplementary Figures 3, 7) or detailed in the captions (Supplementary Figures 4-6).

LI. 264-266: Where can one see the correlations of antibody responses to different Spike antigens?

Response: We have generated three new correlation plots for IgG Spike BA.4/5 vs IgG Spike Ancestral, for marker levels at (1) M0 in the Hybrid Group, as well as at (2) Peak in the Hybrid Group and (3) Peak in the Vaccine Group. We have also generated a correlation plot for IgG Spike BA.4/5 vs IgG Spike BA.2.12.1 for marker levels at Peak in the Hybrid Group (this was the lowest correlation observed across the different Spike markers). These four correlation plots have been added to the revision as Supplementary Figure 3.

In addition, correlations in the Hybrid Group of M0 levels of IgG Spike BA.4/5 with each of the other measured IgG Spike variants all exceeded 0.95, and at Peak they all exceeded 0.66. Results were similar in the Vaccine Group, with the Peak levels of IgG Spike BA.4/5 with each of the other measured IgG Spike variants all exceeding 0.92.

LI. 266-269: A correlation coefficient of 0.8 would be considered a high correlation. So, there is not really more independent information for Spike BA.4/5 and N Index.

Response: We agree a correlation of 0.8 is borderline high, and two markers with a 0.8 correlation could exhibit similar association with the outcome. However, we still

opted to analyze the N Index marker because previous work by other groups showed interesting results regarding serum anti-N binding antibody levels (i.e., Dowell et al. analyzed data from a prospective cohort study in 9-13 year-olds during 2022 in the United Kingdom and found that serum anti-N IgG concentration was a stronger correlate of reinfection by SARS-CoV-2 than serum anti-Spike IgG concentration,⁷ and Miyamoto et al. analyzed data from a prospective cohort study in Japanese adults from late 2022 to early 2023 and found that serum anti-N antibody concentration induced by prior infection in a hybrid immunity population was a stronger inverse correlate of BA.5 COVID-19 than serum anti-Spike antibody concentration⁸). We summarize the results of Dowell et al. in our Discussion (lines 637-640).

LI. 273-276: It is unclear how vaccinations lead to less strong correlations. Also, from the data it does not look like “nearly all persons with hybrid immunity at enrolment had [...] M0 [IgG Spike BA.4/5] values between 100 and 10,000 AU/ml”. It seems that it’s less than half.

Response: We have revised this sentence (underlined) as follows:

“Lower correlations at Peak could potentially be explained by the observation that vaccination appeared to narrow the range of inter-individual variation in IgG Spike BA.4/5 levels (compare the Hybrid Group plots in Figure 1C vs. those in Figure 1D), and the fact that the majority of the participants in the Hybrid Group had post first vaccination values > 100,000 AU/ml vs. M0 values between 100 and 100,000 AU/ml.” (lines 300-304)

- Section “Higher neutralizing antibody titers to BA.4/5 at enrolment are an independent inverse correlate of post-vaccination COVID-19 risk among those with hybrid immunity”:

Table 1: Lines would help to distinguish the different rows and columns.

Response: We have made formatting changes, including shading, bolding, and horizontal lines, to Tables 1 and 3 to improve their readability.

L. 296f: Why not perform some statistical test on the measurements directly, instead of calculating geometric mean ratios?

Response: Thank you for the chance to clarify. Titers are routinely analyzed in its geometric mean and geometric mean fold increase/ratio because they are often right-skewed in their raw measurements; see, e.g., refs.^{9,10} There is no loss of information because testing for a geometric mean ratio departing from unity is equivalent to testing for a mean difference of log scale marker data departing from zero.

L. 302: What covariates did the Cox model adjust for? Since the methods come after the results it would be good to mention them here. Maybe not too detailed, just something like “adjusted for age, sex and BMI”.

Response: We now state:

“First, based on proportional hazards models adjusting for region, PWH status, TB status, enrolment period, and baseline risk score, IgG Spike BA.4/5 and nAb-ID50 BA.4/5 levels at M0 inversely correlated with COVID-19 over 92 days follow-up post Peak ($p=0.017$, $p=0.003$).” (lines 330-333)

L. 304: More precisely IgG N at M0 was not significantly associated with a risk reduction of symptomatic COVID-19.

Response: We have revised (underlined) to “IgG N at M0 trended toward inverse correlation, but this association was not significant ($p=0.079$)”. (lines 333-334)

LI. 307-310: How is non-significant risk reduction for the marker levels at Peak recapitulated by significant risk increase through antibody fold-rise? The fold-rise result seems counter-intuitive. See also issue in discussion.

Response: Thank you for the chance to clarify. Participants with higher M0 antibody level tended to have lower fold-rise; therefore, the positive association between the M0 antibody level and COVID-19 translated to a negative association between the fold-rise marker and COVID-19. We have now included a scatterplot of M0 nAb-ID50 Spike BA.4/5 against the fold-rise nAb-ID50 Spike BA.4/5 as Supplementary Figure 12, and modified the sentence as follows (revisions underlined):

“These results are recapitulated by the fold-rise in antibody levels being significantly directly correlated with COVID-19 (Figure 3B), as the fold-rise inversely correlated with the antibody level at M0 (Supplementary Figure 12).” (lines 338-340)

Table 2: Panel B: Which variable do the p-values correspond to?

Response: We have now clarified in one of Table 2’s footnotes (footnote #4) that the p-values are the usual goodness-of-fit test testing of the null hypothesis that the coefficients of the added variables are 0.

“The P-values tested the null hypothesis that the coefficients of added variables (one or more) all equal 0.”

LI. 312f: Since Table 2 is separated into A and B it should be referenced properly here.

Response: We have revised the Results section reporting on Table 2, including adding references to the relevant part of the table. Please see the detailed revisions in the response to the reviewer’s query pertaining to L. 316 (below).

L. 314 “are the inverse correlate”: It is unclear what the authors want to say here.

Response: We have revised the Results section reporting on Table 2, including clarifying the phrase to which the reviewer refers. Please see the detailed revisions in the response to the reviewer’s query pertaining to L. 316 (below).

L. 316: Unclear what the authors want to say with “the independent correlate”. Independent is a well-defined statistical term but, as far as I understand, not used in the statistical sense here. This is confusing to me.

Response: Thank you for the chance to clarify. We meant in a statistical sense: nAb-ID50 BA.4/5 is still statistically associated with COVID-19 after the model controls for another marker (Table 2A). For instance, in a bivariate model with M0 IgG Spike BA.4/5 and M0 nAb-ID50 Spike BA.4/5, M0 nAb-ID50 Spike BA.4/5 is still associated with COVID-19 ($P = 0.004$) (model 3, Table 2A).

We have revised the relevant paragraph in Results as follows (underlined):

“To address the question of whether the identified antibody correlates still associate with COVID-19 risk in the Hybrid Group even after controlling for the other antibody correlates that were identified, i.e. which antibody markers are independent correlates, multivariable Cox models including pairs of antibody markers were analyzed. Results from a Cox model including IgG Spike BA.4/5 at both M0 and Peak (Table 2A, model 4) and from a Cox model including nAb-ID50 BA.4/5 at both M0 and Peak (Table 2A, model 5) showed that the M0 markers are inverse correlates of COVID-19 risk through 230 days post Peak (HRs = 0.44, 0.46; p-values 0.029, 0.002), while the Peak markers are not (HRs = 1.86, 1.13; p-values 0.21, 0.45). Models with any pair of the three antibody markers at M0 (Table 2A, models 1, 2, and 3) showed that nAb-ID50 BA.4/5 is the independent correlate (models 2 and 3: HRs = 0.50, 0.36; p-values 0.018, 0.004), i.e., is still statistically associated with COVID-19 risk even after the model controls for any of the other markers considered in the analysis. Moreover, comparing to the univariable Cox model results (HR = 0.46, p-value 0.002, Figure 3B), the strength of the inverse correlation of M0 nAb-ID50 BA.4/5 with COVID-19 risk remains similar whether or not M0 IgG Spike or M0 N levels are accounted for. Given that N levels are a proxy for recency or for the severity of prior illness, these results support robustness of the neutralization correlate to this aspect of hybrid immunity heterogeneity.” (lines 342-359)

L. 321: Controlled-risk causal analysis: It should be clarified in the main text that causality is assumed because of the inverse probability weighting. Maybe also mention against which type of misspecification the inverse probability weighting helps.

Response: A key assumption for the marginalized risk to have a causal interpretation is sequential ignorability, namely that there is no unmeasured confounding between the marker to be studied and COVID-19 after controlling for the observed covariates.^{11,12}

Nevertheless, for simplicity we have revised this to “controlled-risk analyses” in the text (lines 361 and 367).

L. 324f: “COVID-19 risk [...] trends with increasing antibody level at Peak” is very inspecific. It should be made clearer what happens with higher antibody levels.

Response: We have revised (underlined) to: “COVID-19 risk decreases with increasing antibody level at M0 (Figure 3C-3E), but remains generally flat with increasing antibody level at Peak (Figure 3F-3G), with a decrease in risk only observed at the highest antibody levels.” (lines 364-366)

All Figures (e.g. 10 + 11) need to have clear labels of the used colors/curves within the figure. Reading the figure caption first to find out about the meaning makes it difficult to interpret the figure.

Response: Thank you for this suggestion. All the thresholded antibody marker figures [Figure 5, Supplementary Figure 13, and Supplementary 15 (the last two were originally numbered Supplementary Figures 10 and 11 but have been renumbered in the revision)] now include an in-figure key that details the meaning of the used colors/curves within the figure.

We have also made additional revisions to each figure caption for clarification. In the caption of Figure 5, for example, we have added the following (underlined):

“Each black dot (threshold value) represents a point estimate of relative risk (compared to the entire Vaccine Group) of COVID-19 through 160 days post Peak for Hybrid Group participants if their marker levels were as high as or higher than that given threshold value. The grid of thresholds was created by segmenting the marker values at COVID-19 endpoints into increments of 0.1. This grid spans from the minimum marker value to the highest value for which there are at least 3 COVID-19 endpoints with a marker value at or above that value. The solid black lines linearly interpolate the grid points. The blue dots represent the marker values of Hybrid Group participants who had a COVID-19 endpoint through 160 days post Peak, overlaid on the black line. The vertical red dashed line is the antibody marker threshold above which no COVID-19 endpoints occurred in the Hybrid Group through 160 days post Peak.” (lines 1321-1330)

Similar revisions have been made to the captions of Supplementary Figures 13 and 15.

Section “In the Vaccine Group, the antibody markers IgG Spike and neutralization titers at Peak are higher in non-cases vs. vaccination-proximal cases and similar vs. vaccination-distal cases”:

LI. 338-343: The first sentence is almost a direct copy of the title. This section is also very short, so why not include it somewhere else?

Response: We have removed this subsection heading and integrated the results into the following results section, “*Vaccine Group Peak antibody levels inversely correlate with risk of COVID-19, but only in early vaccination-proximal follow-up*” (lines 379-380).

- Section “Vaccine Group Peak antibody levels inversely correlate with risk of COVID-19, but only in early vaccination-proximal follow-up”:

LI. 371f: There seems to be no uncertainty displayed here. Why is that?

Response: Uncertainty in Supplementary Figure 15C and 15D (the figure referenced in the sentence in the reviewer’s comment) is shown via the grey shaded area in each plot, which indicates pointwise 95% confidence intervals. This is stated in the figure caption: “The grey shaded area indicates pointwise 95% CIs.”

LI. 374-377: The PWH group was not mentioned in the paragraph before. How do these data support this claim then?

Response: We appreciate this opportunity to improve our manuscript. We have moved this information to the Discussion and revised (underlined) as follows: “The finding that Peak nAb-ID50 BA.4/5 titer inversely correlated with risk of COVID-19 in the Vaccine Group through early vaccination-proximal follow-up, coupled with the statistical interaction test result finding no evidence that HIV status modifies this inverse correlation, together support that among PWH, those with high post vaccination BA.4/5 neutralization titers — especially those with > 1000 AU/ml — develop significant short-term protection after receiving 2 doses of ancestral strain vaccine, but that this protection wanes precipitously over time.” (lines 568-574)

- Section “Antibody marker correlates of risk are similar for people with and without HIV”:

Supplementary Figure 12: The visualization in the supplementary of the Cox model results is better than in the main text. Maybe adapt the main text figures according to this.

Response: We have more fully harmonized the aesthetic appearance of the Cox model results forest plots in Figures 3 and 4 with that of the ones in Supplementary Figure 16 (numbered Supplementary Figure 12 in the original submission).

Why not estimate the effect size of PWH in the Cox model directly?

Response: Thank you for the chance to clarify. We are interested in whether the association between the immune marker and COVID-19 is modified by whether the person is PWH or PWOH. The rationale is that knowledge of this could shed insight on whether immune correlates generalized to PWH (most previous immune correlates work did not enroll PWH).

If we understand your question correctly, you are asking for the association between PWH status and COVID-19 hazard. This question was previously studied in the efficacy paper of the CoVPN 3008 study.⁶ For example, exploratory analyses comparing the cumulative incidence estimates of COVID-19 in different subgroups found comparable estimates between PWH and PWOH (see Figures S19 to S33⁶): in baseline SARS-CoV-2 negative recipients of two mRNA-1273 doses, the hazard ratio of CDC COVID-19 was 0.89 (95% CI: 0.40, 1.94; P=0.762)] for PWOH vs. PWH. Similarly, in baseline SARS-CoV-2 positive recipients of one mRNA-1273 dose, the hazard ratio of CDC COVID-19 was 0.88 (95% CI: 0.57, 1.36; P=0.572)] for PWOH vs. PWH (Figure S19).⁶

L. 385: What does it mean to perform an interaction test? It should probably be called the test of the interaction term.

Response: We now state:

“Tests for whether HIV status modifies the correlate provided no evidence of effect modification, with one possible exception: M0 IgG N (vaccine-matched) had p-value...” (lines 420-421)

L. 388: Is multiple testing corrected p-value meant here?

Response: Thank you for the chance to clarify. Because we conducted a test of the interaction between the marker and PWH status for multiple markers, multiplicity adjusted P-values are reported.

- Section “Correlates of protection analyses support that the Hybrid Group advantage over the Vaccine Group was partially mediated through the antibody markers”:

Figure 5: There should be a legend for the figure such that it is possible to understand which curve belongs to which y-axis.

Response: We have added this.

Section: “Circulating strains have similar Spike sequence antigenic distances to the mRNA-1273 vaccine strain in CoVPN 3008 compared to the COVE and COVAIL correlates studies”

This section seems very disconnected from the rest and it is unclear why this antigenic distance is even studied.

Response: Thank you for the suggestion. We have rearranged the Results section so that the “Quantitative comparison of CoVPN 3008 nAb-ID50 correlates results to those of previous mRNA-1273 vaccine immune correlates studies” section, which provides context/background on the COVE and COVAIL studies, precedes the “Circulating strains have similar Spike sequence antigenic distances to the mRNA-1273 vaccine strain in CoVPN 3008 compared to the COVE and COVAIL correlates studies” now. Additionally, we have added more context:

“To provide context for interpreting the results above, we compared how well the vaccine strain matched the circulating strains in the three studies, given that CoVPN 3008 was conducted over a different period of the COVID-19 pandemic compared to COVE and COVAIL.” (lines 509-512)

- Section “Quantitative comparison of CoVPN 3008 neutralizing antibody correlates results to those of previous mRNA-1273 vaccine immune correlates studies”:

Figure 6: A-D: Using two different x-axes on one graph is too confusing. Some of the content should be moved to the supplementary. E-J: Missing x-axis labels. y-axis labels should be “density” and not the study name.

Response: Thank you for the chance to clarify.

On the comment on panels A-D: The distribution of the time to the clinical endpoint is critical for understanding and harmonizing the results from multiple studies; therefore, we kept it in the main figure and use two different x-axes.

On the comment on panels E-J: We have updated the panels by adding x-axis labels as well as adding “Density” to the y-axis labels.

L. 451: If the subfigures E, G, and I are analyzed first, they should appear as A, B, and C for better reliability.

Response: Thanks for this suggestion, which we have carefully considered.

However, given the fact that there is no journal requirement to cite figure panels in order, it is our view that rearranging the figure so that the overlaid risk curves for the three trials – the most important part of the figure – are the bottom 4 panels would somewhat bury the message of the figure, and we prefer to place those panels more prominently at the top of the figure.

Discussion:

=====

LI. 519f: What is multiplicity correction? Is multiple testing correction meant here?

Response: Thank you for the chance to clarify. Yes, multiplicity correction means multiple comparisons correction. Because we conducted a test of the interaction between PWH/PWoH and the marker level for multiple markers, we report both the raw P-values and the FWER-corrected P-values in Supplementary Table 3.

LI. 522: “Suggestive of correlates” because they were not significantly associated with a reduced risk of COVID-19? This should be stated more clearly.

Response: We have revised this portion as follows (underlined):

“From the Cox regression models, none of the Peak marker HRs had a P-value below the chosen threshold for defining statistical significance, 0.05. However, Peak nAb-ID50 BA.4/5 trended towards an inverse correlate, with this conclusion based on the Cox model result (HR 0.72; 95% CI: 0.52, 1.01; P=0.059) combined with the marker-thresholded correlates analysis that showed a precipitous drop in COVID-19 risk through 160 days post Peak at high Peak nAb-ID50 BA.4/5 titer.” (lines 595-601)

LI. 532-539: It seems counter intuitive that a larger fold-change between pre and post-vaccination would lead to an increased hazard. However, it is never explained what the reason for this could be. It seems that this is simply ignored. If it is not a good marker to use, then it simply should not appear in the model or if it does it should be discussed more in detail.

Response: Thank you for the chance to clarify. Please see our response to your previous question. Reproduced below:

Participants with higher M0 antibody level tended to have lower fold-rise; therefore, the positive association between the M0 antibody level and COVID-19 translated to a negative association between the fold-rise marker and COVID-19. We have now included a scatterplot of M0 nAb-ID50 Spike BA.4/5 against the fold-rise nAb-ID50 Spike BA.4/5 as Supplementary Figure 12, and modified the sentence as follows (revisions underlined):

“These results are recapitulated by the fold-rise in antibody levels being significantly directly correlated with COVID-19 (Figure 3B), as the fold-rise inversely correlated with the antibody level at M0 (Supplementary Figure 12).” (lines 338-340)

L. 580: Second time that the wording “mark [...] risk” was used. It is unclear what the authors mean. Is the meaning simply “antibodies reduce risk”? Later in the text (l. 594) it is also written “mark protection”.

Response: We have revised to

“were too low ~150-230 days post Peak for an association with reduced COVID-19 risk to be observed” (lines 585-586) and “higher levels of these antibodies may be

needed for an inverse association with risk to be observed.” (lines 624-625)

L. 588: The authors mention “both markers” but do not mention the specific markers.

Response: Thank you for the chance to clarify. We now state: “Peak nAb-ID50 in both studies” (lines 644-645).

LI. 588-602: The comparison with COVE and COVAIL could be combined with the other section where a comparison was done.

Response: We have reorganized portions of the Discussion to help consolidate comparison sections, although we maintain our previous structure of discussing results pertaining to the Hybrid Group in one paragraph (“For the Hybrid Group...”, lines 628-641) followed by discussion of results pertaining to the Vaccine Group in a following paragraph (“For the Vaccine Group...”, lines 642-658). Within each of these paragraphs, we begin with comparison to literature including COVE and COVAIL.

L. 611: How is “variability” of the antibodies important and not just the level of the antibodies?

Response: Revised to “the range of nAb-ID50 titers across participants remained high enough to provide protection for 8 months.” (lines 671-672)

Methods:

=====

L. 646: What is “baseline”? Is that the same as enrolment M0 or something else?

Response: “Baseline” in the manuscript refers to the screening or enrollment visit, which were often but not always the same visit. Appendix B from the study protocol (provided with Garrett et al.) illustrates how some baseline procedures (outlined in red) were done at screening versus at the enrolment. Screening may have occurred over the course of several contacts/visits up to and including Day 1 prior to study product administration.

Appendix B Laboratory procedures table for Groups 1 and 3 (SARS-CoV-2 seronegative participants, main study)

Procedure	Ship to ¹	Assay Location ²	Tube ⁴	Tube size (vol. capacity) ⁴	Visit:	01	02	03	04	05	106	107	108	109	110	111	112	113	114	115	116	117
					Day:	-56 to 1	D1	D29	D57	D169	D197	D231	D265	D299	D333	D365	D396	D426	D456	D486	D516	D547
					Week:	W0	W1	W4	W8	W24	W28	W33	W38	W43	W48	W52	W56	W61	W65	W69	W74	W78
					Month:	Screening visit ³	M0	M1	M2	M6	M7	M8	M9	M10	M11	M12	M13	M14	M15	M16	M17	M18
						Vac 1	Vac 2	Vac 3														
BLOOD COLLECTION																						
Screening/Diagnostic																						
HIV test ⁵	Local Lab	Local Lab	EDTA	5mL	5	—	—	—	—	—	—	—	—	—	—	—	—	—	—	—	—	—
HIV Viral Load ⁶	Local Lab	Local Lab	EDTA	5mL	5 ⁶	5 ⁶	—	—	5 ⁶	—	—	—	—	—	—	—	—	—	—	—	—	—
CD4+ T Cell Count ⁶	Local Lab	Local Lab	EDTA	5mL	5 ⁶	5 ⁶	—	—	5 ⁶	—	—	—	—	—	—	—	—	—	—	—	—	—
SARS-CoV-2 POC Serology	Local Lab	Local Lab	EDTA	5mL	5	—	—	—	—	—	—	—	—	—	—	—	—	—	—	—	—	—
SARS-CoV-2 Serology	BARC	HVTN Labs	SST	5mL	—	5	—	5	5	5	—	—	—	—	—	5	—	—	—	—	—	5
Immunogenicity assays																						
Humoral assays	BARC	HVTN Labs	SST	8.5mL	—	8.5	—	8.5	8.5	8.5	—	—	—	—	—	8.5	—	—	—	—	—	8.5
Gene Expression	BARC	HVTN Labs	Tempus	3mL	—	8	—	8	8	8	—	—	—	—	—	8	—	—	—	—	—	8
Visit total					20.0	29.5	0.0	19.5	29.5	19.5	0.0	0.0	0.0	0.0	0.0	19.5	—	—	—	—	—	19.5
56-Day total					20.0	49.5	49.5	69.0	29.5	49.0	19.5	19.5	0.0	0.0	0.0	19.5	—	—	—	—	—	19.5
NASAL SWAB SAMPLE COLLECTION⁷																						
SARS-CoV-2 PCR	Local Lab	Local Lab			—	X	X	—	X	X	X	X	X	X	X	X	X	X	X	X	X	X
SARS-CoV-2 Viral Sequencing ⁷	BARC	HVTN Labs			—	X'	X'	—	X'	X'	X'	X'	X'	X'	X'	X'	X'	X'	X'	X'	X'	X'
URINE COLLECTION																						
Pregnancy Test ⁸	Local Lab	Local Lab			X	—	—	—	—	—	—	—	—	—	—	—	—	—	—	—	—	—

¹ BARC = Bio Analytical Research Corporation South Africa (Pty) Ltd (Johannesburg, South Africa).

² HVTN Laboratories include: Bio Analytical Research Corporation South Africa (Pty) Ltd (Johannesburg, South Africa); Duke University Medical Center (Durham, North Carolina, USA); KwaZulu-Natal Research and Innovation Sequencing Platform (KRISP, Durban, South Africa).

³ Screening may occur over the course of several contacts/visits up to and including Day 1 prior to study product administration.

⁴ Local labs may assign appropriate alternative specimen type, tube and volume for locally performed tests.

⁵ HIV diagnostic test will be performed for all participants at screening and as clinically indicated.

⁶ HIV Viral Load and CD4+ T Cell Count will be performed at screening or at any visit for participants with newly diagnosed HIV infection, at enrollment for known PLWH, approximately 4-6 weeks after any change in antiretroviral therapy (other than just dosage) for PLWH, at visit 5 for PLWH, and as clinically indicated. (see Sections 9.2, 9.3, 9.4 and 9.9). HIV Viral Load and CD4+ T Cell Count do not need to be recollected if they were collected and the results available within the past 14 days.

⁷ Nasal swab sample will only be sent to site processing lab for processing and storage if local testing results at the same visit return as SARS-CoV-2 infected.

⁸ Pregnancy test will be performed at screening for all participants who were assigned female sex at birth and are of reproductive potential. A pregnancy test will be performed at other visits if the participant reports a new pregnancy and / or as clinically indicated.

L. 647: What is a “point-of-care” serology test? Should be explained more.

Response: “Point-of-care” refers to laboratory testing that is conducted close to (or at) the site of patient care.¹³

The point-of-care serology testing was part of screening because it determined group assignment and was conducted before the first vaccination, which occurred at enrollment.

We have revised as follows (underlined): “The baseline point-of-care anti-Spike serology test result was used to determine whether participants were given a second mRNA-1273 dose at Month 1, where only participants with a negative result were given a second mRNA-1273 dose at Month 1.” (lines 759-762)

LI. 665-667: This sounds like COVID-19 endpoints were only counted as COVID-19 endpoints if they occurred within 7 days after the Peak or after 230 days after the Peak.

Response: Revised (underlined) to “except COVID-19 endpoints began to be counted only at 7 days after the Peak visit up until 230 days post Peak (or up until 92 days post Peak, as noted), including endpoints up to March 31, 2023. (lines 775-777)

L. 681: If there are measurements that are in BAU/mL, why are all results reported in AU/mL?

Response: Readouts to the other, non-Index strains cannot be converted to BAU/ml because the anti-SARS-CoV-2 immunoglobulin international standard has not been assayed against these strains for conversion. We have revised as follows:

“Readouts of the Index D614 strain (Spike) and Index D614 strain (N) in AU/ml units may be multiplied by 0.009 to convert them to Binding Antibody Units (BAU/ml), for comparison with previous correlates analyses.^{19,23} Readouts to the other, non-Index strains cannot be converted to BAU/ml because the anti-SARS-CoV-2 immunoglobulin international standard has not been assayed against these strains for conversion.” (lines 797-802)

LI. 682-685: It would be nice to see the different assay measurements; not just the three that are presented in the figures. This could become part of the supplementary. Or at least argue why they weren't used more clearly in the results.

Response: Correlations in the Hybrid Group of M0 levels of IgG Spike BA.4/5 with each of the other measured IgG Spike variants all exceeded 0.95, and at Peak they all exceeded 0.66. Results were similar in the Vaccine Group, with the Peak levels of IgG Spike BA.4/5 with each of the other measured IgG Spike variants all exceeding 0.92. These high correlations are the basis for why we focus the analyses on IgG Spike BA.4/5 and IgG Spike Index and do not present results for the other Spike antigens.

LI. 702f: Why are LOD values assigned LOD/2

Response: To clarify, values below the LOD were assigned LOD/2: “The limit of detection (LOD) was 10 AU/ml, and values below the LOD were assigned LOD/2” (lines 827-828).

Thanks for the good question, as at some level the value LOD/2 is arbitrary and not reality as to the actual readout value. Over the years our group has considered a variety of ways to deal with detection limits, ranging from likelihood based that use the information that a value < LOD but does not impute to LOD/2, to the simple approach that assigns value LOD/2.

The imputation of values < LOD to LOD/2 was specified in the SAP (Section 5.1.3) and is a convenient and conventional approach for handling values that are below an assay's LOD. Some statistical methods require actual values, not only knowledge that the response is below the LOD, such that a numerical value is needed.

(Likelihood-based methods are an exception that can express the knowledge that a marker is < LOD without imputing a value.) Because a variety of statistical methods are used developed by various people, setting of values below the LOD to LOD/2 affords the flexibility to allow all of these methods to be applied. Some analyses treat an immune marker as 'mixed binary and continuous' which consider the subgroup with undetectable level (defined by value < LOD) and analyzes the continuous readout among those with values > LOD. These methods do not use the imputation

to the specific value LOD/2 and hence are attractive from the perspective of not using an arbitrary value.

In sum, we ended up electing for LOD/2 simple imputation because it enables many statistical methods to be applied without requiring all of them to be restricted to a certain subtype of methods; this facilitates many statisticians participating and speeds up the analysis work. In parallel, we also encourage use of methods that use a better approach than LOD/2 assignment when we can.

For the set of COVID-19 vaccine immune correlates analyses, we have in general used the LOD/2 imputations.

L. 703: The abbreviation ULOQ was not explained.

Response: Fixed (see line 829).

LI. 703-705: How were these thresholds chosen? What would be the effect of choosing them differently?

Response: The choice, which was guided by commonly used rules by the FDA for a positive seroresponse (i.e., 4-fold increase), was prespecified in the SAP. We considered performing a sensitivity analysis to discern the possible effects of choosing these thresholds differently but decided that such an analysis is outside the scope of this work given how commonly used these thresholds are.

LI. 714f: What are two-sided bootstrap p-values? What kind of test was performed?

Response: Please see our response to the same question you raised previously.

L. 818: Since R versions or more specifically the packages are not always compatible, ideally one would only use a single version of R to ensure reproducibility.

Response: We agree that this would be the ideal scenario; however, analyses were run by different statisticians at different periods of time over the course of the project, and in practice this was not feasible.

At our openly available code folder

(https://github.com/CoVPN/CoVPN3008/tree/master/Antibody_manucript), we

include a full README file that includes all information necessary to ensure reproducibility.

** See Nature Portfolio's author and referees' website at www.nature.com/authors for information about policies, services and author benefits.

References

- 1 Plotkin, S. A. & Gilbert, P. B. Nomenclature for immune correlates of protection after vaccination. *Clin Infect Dis* **54**, 1615-1617, doi:10.1093/cid/cis238 (2012).
- 2 Plotkin, S. A. Recent updates on correlates of vaccine-induced protection. *Front Immunol* **13**, 1081107, doi:10.3389/fimmu.2022.1081107 (2022).
- 3 Plotkin, S. A. & Gilbert, P. B. Correlates of Protection. In: Vaccines, Eighth Edition, Editors Walter Orenstein, Paul Offit, Kathryn Edwards, Stanley Plotkin. Pages 35-40. Elsevier Inc., New York. (2022).
- 4 Trombetta, C. M. & Montomoli, E. Influenza immunology evaluation and correlates of protection: a focus on vaccines. *Expert Rev Vaccines* **15**, 967-976, doi:10.1586/14760584.2016.1164046 (2016).
- 5 Gilbert, P. B. *et al.* A Covid-19 Milestone Attained — A Correlate of Protection for Vaccines. *New England Journal of Medicine* **387**, 2203-2206 (2022).
- 6 Garrett, N. *et al.* Hybrid versus vaccine immunity of mRNA-1273 among people living with HIV in East and Southern Africa: a prospective cohort analysis from the multicentre CoVPN 3008 (Ubuntu) study. *eClinicalMedicine* **80**, 103054, doi:<https://doi.org/10.1016/j.eclinm.2024.103054> (2025).
- 7 Dowell, A. C. *et al.* Nucleocapsid-specific antibodies as a correlate of protection against SARS-CoV-2 reinfection in children. *J Infect* **87**, 267-269, doi:10.1016/j.jinf.2023.06.018 (2023).
- 8 Miyamoto, S. *et al.* Serum anti-nucleocapsid antibody correlates of protection from SARS-CoV-2 re-infection regardless of symptoms or immune history. *Commun Med (Lond)* **5**, 172, doi:10.1038/s43856-025-00894-8 (2025).
- 9 Nauta, J. *Statistics in Clinical Vaccine Trials*. (Springer Science & Business Media, 2010).
- 10 Gilbert, P. B. *et al.* Immune correlates analysis of the mRNA-1273 COVID-19 vaccine efficacy clinical trial. *Science* **375**, 43-50, doi:10.1126/science.abm3425 (2022).
- 11 Gilbert, P. B., Fong, Y., Kenny, A. & Carone, M. A Controlled Effects Approach to Assessing Immune Correlates of Protection. *Biostatistics* **24**, 850–865 (2023).
- 12 Imai, K., Keele, L. & Yamamoto, T. Identification, inference and sensitivity analysis for causal mediation effects. *Statist. Sci.* 25(1): 51-71 (February 2010). DOI: 10.1214/10-STS321 (2010).
- 13 Larkins, M. C. & Thombare, A. in *StatPearls* (2025).

Reviewer #1 (Remarks to the Author):

The authors have adequately addressed my concerns.

Response: Thank you for the positive comment.

Reviewer #2 (Remarks to the Author):

Reviewer #2 (Remarks on code availability):

Code still not available.

Response: The link to the website where our code is available is contained in our Data Availability statement (copied below, lines 1011-1014 in the revision). Multiple people have tested our Github link on multiple browsers and the link appears to be functioning properly. Moreover, as recommended, we have obtained a DOI for the Github repository in order to provide a permanent reference to the version of the code used in this study and improve reproducibility.

“Code Availability

All analyses were done reproducibly based on publicly available R scripts hosted on the GitHub collaborative platform at https://github.com/CoVPN/CoVPN3008/tree/master/Antibody_manucript.⁶³”

Reference #63:

63 Zhang, B. R code for reproducing the results in "Neutralizing and Binding Antibodies are a Correlate of Risk of COVID-19 in the CoVPN 3008 Study in People with HIV" (1.0). Zenodo. <https://doi.org/10.5281/zenodo.16729966>. (2025).

If the reviewer continues to have issues accessing the code, could they please provide us with the specific error message(s) that they are getting when they navigate to the github site or the Zenodo doi above?

Reviewer #3 (Remarks to the Author):

The authors have addressed my main concerns, and the manuscript has improved substantially in both clarity and structure. I now recommend the manuscript for

publication, pending minor revisions as outlined below. The abstract and introduction are clearer, and research goals are clearly outlined. The sampling procedure is now better explained, and the revisions have helped clarify the methodology. The coherence of the figures has improved overall, and I commend the authors for making their code available, which enhances the transparency and reproducibility of their work.

Response: Thank you for the positive comments.

Some minor issues remain, which are mostly a result of the revisions (line numbers as in the revised manuscript):

- Paragraph following l. 190: The authors properly introduce CoRs and CoPs, however the description of the terms is very disconnected from the rest of the introduction. This could be improved.

Response: We have revised to include two transition sentences: “This analysis investigates antibody markers as correlates of risk (CoRs) of COVID-19, seeking insights into hybrid and vaccine immunity and whether antibody correlates previously defined in immunocompetent persons are qualitatively and quantitatively similar among PWH. A CoR refers to....” (lines 187-190) as well as “A large body of work has studied immune CoRs and CoPs against COVID-19.¹¹⁻¹³” (line 204).

- 11 Gilbert, P. B. *et al.* A Covid-19 Milestone Attained — A Correlate of Protection for Vaccines. *New England Journal of Medicine* 387, 2203-2206 (2022).
- 12 Mahrokhian, S. H., Tostanoski, L. H., Vidal, S. J. & Barouch, D. H. COVID-19 vaccines: Immune correlates and clinical outcomes. *Hum Vaccin Immunother* 20, 2324549, doi:10.1080/21645515.2024.2324549 (2024).
- 13 Goldblatt, D., Alter, G., Crotty, S. & Plotkin, S. A. Correlates of protection against SARS-CoV-2 infection and COVID-19 disease. *Immunol Rev* 310, 6-26, doi:10.1111/imr.13091 (2022).

- L. 232: Minor inconsistency. The group now referred to as “Serum-RIS” is still labeled simply as “RIS” in Supplementary Figure 1.

Response: We have changed all “RIS” labels in Supplementary Figure 1 to “Serum-RIS”.

- Ll. 310f: This sentence seems disconnected from the rest and does not describe any results.

Response: We have added the following to describe the spaghetti plot:

“Vaccination boosts anti-Spike responses in the majority of study participants in both the Hybrid and Vaccine groups.” (lines 312-313)

- L. 322: The "geometric mean" was already used in l. 273, but was not abbreviated here.

Response: "Geometric mean" has been removed here due to revisions made in response to the comment pertaining to line 839 (below).

- Ll. 326f: The same as in ll. 310f.

Response: We have better connected and described Supplementary Figure 9 through the addition of the following sentences:

"Supplementary Figure 9 shows the distributions of the two fold-rise (M0 to Peak) antibody markers for IgG Spike BA.4/5 and nAb-ID50 BA.4/5, in cases and in non-cases in the Hybrid Group. Among cases and among non-cases, the median fold-rise was higher for nAb-ID50 BA.4/5 than for IgG Spike BA.4/5, and for each antibody marker the fold-rise was higher in cases than in non-cases. Specifically, the median fold-rise of IgG Spike BA.4/5 on the log₁₀ scale was 1.35 (22-fold increase) for cases and 0.97 (9-fold increase) for non-cases, and for nAb-ID50 BA.4/5 it was 1.54 (35-fold increase) for cases and 1.08 (12-fold increase) for non-cases. This result is consistent with the observation above that the difference in antibody marker levels between non-cases and cases is smaller at Peak compared to at M0." (lines 328-337)

- Paragraph following l. 451: The authors should consider adding the hazard ratios to the previous paragraph, as well.

Response: We have added the following sentences (underlined) in the revision:

"The controlled risk curves show consistency of the correlates results across the trials over the 92-day follow-up period (Fig. 6A, 6B) in that over the range of overlapping titers, COVID-19 risk decreases with increasing Peak nAb-ID50 titer, in both the Hybrid and Vaccine groups. [This finding is consistent with the previously reported HRs (95% CIs) of COVID-19 per 10-fold increase in Peak nAb-ID50 BA.1 titer: 0.31 (0.10, 0.96) in the naïve (Vaccine) cohort and 0.28 (0.07, 1.08) in the non-naïve (Hybrid) cohort over 92 days post-dost 3 in the COVE booster study (Fig. 2e in Zhang et al.³⁹), as well as the HR of COVID-19 per 10-fold increase in Peak nAb-ID50 BA.4/5 titer in the non-naïve (Hybrid) cohort over 92 days post-second booster for one-dose mRNA-1273 second boost recipients in COVAIL study, 0.35 (0.20, 0.62). The corresponding HR for the naïve (Vaccine) cohort was 0.88 (0.65, 1.19)]."(lines 502-513)

39 Zhang, B. *et al.* Omicron COVID-19 immune correlates analysis of a third dose of mRNA-1273 in the COVE trial. *Nat Commun* 15, 7954, doi:10.1038/s41467-024-52348-9 (2024).

- LI. 589f: Missing reference to Omicron evolution.

Response: Based on Fig. 4 in Garrett et al. and the fact that no JN isolates were identified in Part A of CoVPN 3008, yet a number of “unknown” and “Other” lineages caused COVID-19 throughout the study, we conjecture that these later unknown lineages may represent pre-JN lineages, such as BA.2.12.1 and XBB.1.5. We have revised and added two references to papers describing circulation of Omicron sublineages in early 2023:

“Two other possible contributing explanations are the continued evolution of the Omicron epidemic from BA.4/5 to Omicron-descendent isolates^{44,45} ...” (lines 617-619)

- 44 Ma, K. C. *et al.* Genomic Surveillance for SARS-CoV-2 Variants: Circulation of Omicron XBB and JN.1 Lineages - United States, May 2023-September 2024. *MMWR Morb Mortal Wkly Rep* 73, 938-945, doi:10.15585/mmwr.mm7342a1 (2024).
- 45 Dor, G. *et al.* Tracing the spatial origins and spread of SARS-CoV-2 Omicron lineages in South Africa. *Nat Commun* 16, 4937, doi:10.1038/s41467-025-60081-0 (2025).

- LI. 617f: Missing reference to a study showing that B cell responses are important for protection.

Response: We have added four references in this sentence:

“...which may be correlated with the salient memory/recall B-cell responses^{46,47} that appear to be critical for protection against future exposures to Omicron viruses.^{48,49}” (lines 649-651)

- 46 Goel, R. R. *et al.* Efficient recall of Omicron-reactive B cell memory after a third dose of SARS-CoV-2 mRNA vaccine. *Cell* 185, 1875-1887 e1878, doi:10.1016/j.cell.2022.04.009 (2022).
- 47 Goel, R. R. *et al.* Distinct antibody and memory B cell responses in SARS-CoV-2 naive and recovered individuals following mRNA vaccination. *Sci Immunol* 6, doi:10.1126/sciimmunol.abi6950 (2021).
- 48 Zhong, Y. *et al.* Correlates of protection against symptomatic SARS-CoV-2 in vaccinated children. *Nat Med* 30, 1373-1383, doi:10.1038/s41591-024-02962-3 (2024).
- 49 Byrne, J. *et al.* Robust and persistent B-cell responses following SARS-CoV-2 vaccine determine protection from SARS-CoV-2 infection. *Front Immunol* 15, 1445653, doi:10.3389/fimmu.2024.1445653 (2024).

- LI. 621f: Missing reference to the limited cross-reactivity.

Response: We show this in the present work. “Peak antibody levels in both the Hybrid and Vaccine groups were also about 6-fold lower against Reference (Supplementary Figure 11, where for Vaccine Group non-cases GM nAb-ID50 against Reference and BA.4/5 was 5279 and 757 AU/ml, respectively).” (lines 465-468)

To address this comment, we have made the following revision (underlined) in the Discussion:

“In contrast, antibodies elicited by vaccination to the Ancestral vaccine strain have limited cross-reactivity against Omicron viruses (~6-fold reduced neutralization titer, Supplementary Figure 11 in the present work)...” (lines 651-653)

- L. 839: The authors have changed from the wording from estimation of the geometric mean to estimation of mean log-scale. This addresses the issue that I raised. However, they still talk about "geometric mean ratios" in the results (for example, l. 273 or 322). This needs to be changed, as it is now unclear where this geometric mean comes from.

Response: We appreciate the feedback from the reviewer. Yes, the reviewer is right that we should have updated our reference to ‘geometric mean ratio,’ which is not accurate. We have now presented the immunogenicity comparisons based on the log₁₀-scale, which is how the estimation and g-computation was conducted. We have updated Supplementary Table 2 and the corresponding numbers in Figure 1 and the text. Revisions in the text are underlined below:

“At Peak, antibody responses are also higher in the Hybrid Group (p-values < 0.001; Figure 1D, 1F and Supplementary Table 2). In the Hybrid Group, antibody responses are similar between PWH and PWOH at both time points (p-values > 0.27; Figure 1E, 1F and Supplementary Table 2), whereas in the Vaccine Group, Peak responses are higher in PWH than PWOH, with a 0.23-log difference (95% CI 0.08, 0.36) for IgG Spike BA.5 and 0.48-log difference (0.17, 0.81) for 50% inhibitory serum dilution neutralizing antibody (nAb-ID50) BA.4/5 titer (Figure 1F and Supplementary Table 2).” (lines 269-276)

The following sentence has also been added to the Figure 1 caption:

“Covariate-adjusted, mean titer level (in the log₁₀-scale) with 95% confidence intervals and two-sided P-values based on non-parametric bootstrap (see Methods for details of how the bootstrapped p-values were obtained based on g-computation) are given above violin boxplots.”